# Theoretical Analysis of Inductive Biases in Deep Convolutional Networks

**Zihao Wang**
Peking University
zihaowang@stu.pku.edu.cn

**Lei Wu**
Peking University
leiwu@math.pku.edu.cn

## Abstract

In this paper, we provide a theoretical analysis of the inductive biases in convolutional neural networks (CNNs). We start by examining the universality of CNNs, i.e., the ability to approximate any continuous functions. We prove that a depth of $\mathcal{O}(\log d)$ suffices for deep CNNs to achieve this universality, where $d$ in the input dimension. Additionally, we establish that learning sparse functions with CNNs requires only $\widetilde{\mathcal{O}}(\log^2 d)$ samples, indicating that deep CNNs can efficiently capture *long-range* sparse correlations. These results are made possible through a novel combination of the multichanneling and downsampling when increasing the network depth. We also delve into the distinct roles of weight sharing and locality in CNNs. To this end, we compare the performance of CNNs, locally-connected networks (LCNs), and fully-connected networks (FCNs) on a simple regression task, where LCNs can be viewed as CNNs without weight sharing. On the one hand, we prove that LCNs require $\Omega(d)$ samples while CNNs need only $\widetilde{\mathcal{O}}(\log^2 d)$ samples, highlighting the critical role of weight sharing. On the other hand, we prove that FCNs require $\Omega(d^2)$ samples, whereas LCNs need only $\widetilde{\mathcal{O}}(d)$ samples, underscoring the importance of locality. These provable separations quantify the difference between the two biases, and the major observation behind our proof is that weight sharing and locality break different symmetries in the learning process.

## 1 Introduction

Convolutional neural networks (CNNs) (Fukushima, 1988; LeCun et al., 1998) are a fundamental model in deep learning, known for their exceptional performance in many tasks. In particular, CNNs consistently outperform the fully-connected neural network (FCN) counterparts in vision-related tasks (Krizhevsky et al., 2012; He et al., 2016; Huang et al., 2017). Uncovering the underlying mechanism behind the success of CNNs is thus of paramount importance in deep learning.

Zhou (2020a;b); Feng et al. (2022); He et al. (2022) studied the approximation capabilities of CNNs for target functions in spaces such as continuous functions and Sobolev spaces. Although these results are important, they cannot explain why CNNs perform better than FCNs. The primary reason for this limitation is that these works did not take into account the specific structures of CNNs, including *multichanneling, downsampling, weight sharing, and locality*. The locality refers to the use of small filters, e.g., filter sizes can be as small as 3 in popular VGG nets (Simonyan and Zisserman, 2014) and ResNets (He et al., 2016). Comprehending the inductive biases of these architecture choices is critical to understand the exceptional performance of CNNs.

Li et al. (2020) designed a simple classification task to demonstrate the superiority of CNNs over FCNs. The authors prove that for this task, FCNs need $\Omega(d^2)$ samples while CNNs need only $\mathcal{O}(1)$ samples, thereby providing theoretical support for the superiority of using convolutions. However, this study neither examined the individual impact of weight sharing and locality nor considered the

37th Conference on Neural Information Processing Systems (NeurIPS 2023).

inductive biases of multichanneling and downsampling. Additionally, the analysis was limited to shallow CNNs and did not examine the interaction between these structures and network depth.

## 1.1 Our Results

In this work, we conduct a systematic analysis of the inductive biases associated with the specific structures of CNNs. Our main contributions are summarized as follows.

**Universality.** We establish the universality of deep CNNs with a depth of $\mathcal{O}(\log d)$. This is in contrast to existing works (Zhou, 2020a;b; He et al., 2022), where the universality requires a depth of at least $\Omega(d)$. The key to our improvement is an effective leveraging of the inductive biases of multichanneling and downsampling:

- *Downsampling* amplifies the size of the receptive field exponentially, thus explaining the need for logarithmic depth. Furthermore, we prove that if downsampling is not used, CNNs require a depth of at least $\Omega(d)$, demonstrating the cruciality of downsampling.

- *Multichanneling* serves as a mechanism for storing extracted information. By increasing the number of channels whenever the spatial dimension is reduced by downsampling, we ensure that no information is lost. This combination of multichanneling and downsampling is widely employed in practical CNNs, ranging from classical LeNet (LeCun et al., 1998) to modern VGG nets (Simonyan and Zisserman, 2014) and ResNets (He et al., 2016).

It is worth mentioning that while studies like Poggio et al. (2017) and Cagnetta et al. (2022) have examined similar CNN architectures with $O(\log d)$ depth, they did not explicitly establish universality as our work does. Specifically, Cagnetta et al. (2022) focused on the kernel associated with deep CNNs having infinitely many channels. In retrospect, one could potentially show that the deep-CNN kernel is universal by verifying that kernel has no zero eigenvalues.

**Learning sparse functions.** A function $f : \mathbb{R}^d \mapsto \mathbb{R}$ is said to be sparse if it only depends on a few coordinates of the input, e.g., $f(\mathbf{x}) := g(x_1, x_d)$ for some $g : \mathbb{R}^2 \mapsto \mathbb{R}$. We prove that learning sparse functions using CNNs requires only $\widetilde{\mathcal{O}}(\log^2 d)$ samples, which is nearly optimal as the information-theoretic lower bound of learning such functions is $\Omega(\log d)$ (Han and Yuan, 2020).

This result is surprising because it has been widely believed that CNNs struggle to capture long-range correlations. However, our findings suggest that CNNs can efficiently learn long-range sparse ones, which is a valuable attribute for many applications.

In addition, it is important to note that the near-optimal sample complexity of learning sparse functions using CNNs is achieved with only $\mathcal{O}(\log d)$ depth and $\mathcal{O}(k^2 \log d)$ total parameters, where $k$ is the number of critical coordinates in sparse functions. A lower bound is established to demonstrate the optimality of the depth requirement. The ability of CNNs to select any $k$ coordinate with only $\mathcal{O}(k^2 \log d)$ total parameters is remarkable, especially considering that even in the linear case, LASSO (Tibshirani, 1996) requires $\Omega(d)$ parameters. It is the synergy of increased depth, weight sharing, and multichanneling that gives CNNs this exceptional capability.

**Disentangling the weight sharing and locality.** We next study the inductive biases of locality and weight sharing by comparing the performance of CNNs, LCNs, and FCNs on a synthetic regression task. This allows us to separate the effects of weight sharing and locality.

- CNNs vs. LCNs. We prove that CNNs requires only $\widetilde{\mathcal{O}}(\log^2 d)$ samples to learn, while LCNs trained by SGD or Adam with standard initialization need $\Omega(d)$ samples. This provides a separation between CNNs and LCNs and demonstrates the crucial role of weight sharing.

- LCNs vs. FCNs. We prove that LCNs requires only $\widetilde{\mathcal{O}}(d)$ samples to learn, while FCNs trained by SGD with Gaussian initialization need $\Omega(d^2)$ samples. This provably separates LCNs from FCNs and demonstrates the benefit of locality.

The difference in sample complexity can be attributed to the different symmetries encoded in the architecture biases. For instance, stochastic gradient descent (SGD) exhibits different symmetries for these models: a lack of equivariance for CNNs, a local permutation equivariance for LCNs, and a global orthogonal equivariance for FCNs. The size of the equivariance groups determines the

minimax sample complexity and distinguishes between different architectures. Similar ideas have been previously explored in Li et al. (2020); Xiao and Pennington (2022) to understand CNNs; but these works did not differentiate the roles of weight sharing and locality. For a detailed comparison with these works, see the related work section in Appendix A.

**Technical contribution.** The lower bounds for LCNs and FCNs are established using Fano's method from minimax theory (Wainwright, 2019, Section 15). However, different from the traditional statistical setup where the estimator is deterministic, the estimators produced by stochastic optimizers are random. To address this issue, we develop a variant of Fano's method for random estimators, which might be of independent interest. Further details can be found in Appendix B.3.

**Related work.** We refer to Appendix A for a detailed comparison with other related works.

## 1.2 Notations

We use $\mathrm{poly}(z_1, \ldots, z_p)$ to denote a quantity that depends on $z_1, \ldots, z_p$ polynomially. We use $X \lesssim Y$ to denote $X \leqslant CY$ for some absolute constant $C$ and $X \gtrsim Y$ is defined analogously. We also use the standard big-O notations: $\Theta(\cdot)$, $\mathcal{O}(\cdot)$ and $\Omega(\cdot)$. In addition, we use $\widetilde{\mathcal{O}}$ and $\widetilde{\Omega}$ to hide higher-order terms, e.g., $\mathcal{O}((\log d)(\log\log d)^2) = \widetilde{\mathcal{O}}(\log d)$ and $\mathcal{O}(d\log d) = \widetilde{\mathcal{O}}(d)$. In addition, we use $G_{\mathrm{ort}}(d)$ to denote the orthogonal group in dimension $d$:

$$G_{\mathrm{ort}}(d) = \{Q \in \mathbb{R}^{d \times d} : QQ^\top = Q^\top Q = I_d\}.$$

Let $\mathcal{P}(\Omega)$ be the set of probability distributions over $\Omega$ and $\mathcal{M}(\Omega)$ be the set of random variables taking values in $\Omega$. Given two functions $f, g$ over $\mathcal{X}$ and $\{\mathbf{x}_i\}_{i=1}^n$, let $\hat{\rho}_n(f, g) = \sqrt{\frac{1}{n}\sum_{i=1}^n |f(\mathbf{x}_i) - g(\mathbf{x}_i)|^2}$. Let $\mathbb{S}^{d-1} = \{x \in \mathbb{R}^d : \|x\|_2 = 1\}, r\mathbb{S}^{d-1} = \{x : x/r \in \mathbb{S}^{d-1}\}$. Let $a \wedge b = \min(a, b)$, $[k] = \{1, 2, \ldots, k\}$ for $k \in \mathbb{N}$, and $[a, b] = \{a, a+1, \ldots, b\}$ for $b, a \in \mathbb{N}$ and $b > a$. For a vector $\mathbf{v}$, denote by $\|\mathbf{v}\|_p := (\sum_i |v_i|^p)^{1/p}$ the $\ell^p$ norm. For a matrix $A$, let $\|A\|$ and $\|A\|_F$ be the spectral norm and Frobenius norm, respectively. Moreover, denote by $A_{i,:}$ and $A_{:,j}$ the $i$-th row and $j$-th column, respectively, and similar notations are also defined for tensors. Given $\mathcal{I} = (i_1, i_2, \ldots, i_k)$, let $\mathbf{x}_{\mathcal{I}} = (x_{i_1}, \ldots, x_{i_k})$. We use $\sigma$ to denote both the activation function and standard deviation of label noise. To avoid ambiguity, we shall use $\sigma(\cdot)$ and $\sigma$ to distinguish them. When applying $\sigma(\cdot)$ to a vector/matrix/tensor, it should be understood in an element-wise manner.

## 2 Preliminaries

We consider the standard setup of supervised learning. Let $\mathcal{X} \subset \mathbb{R}^{4d}$ and $\mathcal{Y} \subset \mathbb{R}$ be the input and output domain, respectively. We are given the training set $S_n = \{(x_i, y_i)\}_{i=1}^n$ with $y_i = h^*(x_i) + \xi_i, x_i \overset{iid}{\sim} P$, and $\xi_i \overset{iid}{\sim} \mathcal{N}(0, \sigma^2)$. Here $P$ and $h^*$ denote the input distribution and target function, respectively. We assume $\log_2 d \in \mathbb{N}^+$ for simplicity. Let $h : \mathcal{X} \times \Theta \mapsto \mathbb{R}$ be our parametric model with $\Theta = \mathbb{R}^p$, where $p$ denotes the number of parameters; we often write $h_\theta = h(\cdot; \theta)$ for short. Our task is to recover $h^*$ from $S_n$ by using $h_\theta$.

Given a threshold $A > 0$, we also consider the truncated model $\pi_A \circ h_\theta$, where the truncation operator $\pi_A$ is defined by $\pi_A \circ h(\mathbf{x}) = \max(\min(h(\mathbf{x}), A), -A)$. In addition, we consider both the square loss $\ell(y, y') = (y - y')^2/2$ and its truncated version $\ell_B(y, y') = \frac{1}{2}(y - y')^2 \wedge \frac{1}{2}B^2$. This loss truncation is applied to handle the noise unboundedness; see Appendix B.2 for more details.

### 2.1 Network Architectures

A $L$-layer neural network is given by $h_\theta(\mathbf{x}) = \mathcal{M}_o \circ \sigma_L \circ \mathcal{T}_L \circ \cdots \circ \sigma_1 \circ \mathcal{T}_1(\mathbf{x})$, where $\sigma_l$ and $\mathcal{T}_l$ denote the activation function and linear transform with bias at the $l$-th layer, respectively. Let $\mathbf{z}^{(l)}$ denote the hidden state of $l$-th layer: $\mathbf{z}^{(l)}(\mathbf{x}) = \sigma_l \circ \mathcal{T}_l \circ \cdots \circ \sigma_1 \circ \mathcal{T}_1(\mathbf{x})$ for $l \in [L]$ and $\mathbf{z}^{(0)}(\mathbf{x}) = \mathbf{x}$. When it is clear from context, we will write $\mathbf{z}^{(l)} = \mathbf{z}^{(l)}(\mathbf{x})$ and $\mathcal{T}_l \mathbf{z} = \mathcal{T}_l(\mathbf{z})$ for short. In different architectures, $\{\mathcal{T}_l\}_{l=1}^L$ are parameterized in different ways. $\mathcal{M}_o$ denotes the output layer, which performs a linear combination of the output features: $\mathcal{M}_o \circ \mathbf{z}^{(L)}(\mathbf{x}) = W_o \mathrm{vec}(\mathbf{z}^{(L)}(\mathbf{x}))$, where $W_o$ is the weight used to parameterize $\mathcal{M}_o$.

**FCNs.** $\mathcal{T}_l : \mathbb{R}^{C_l} \mapsto \mathbb{R}^{C_{l-1}}$ is a fully-connected transform parameterized by $\mathcal{T}_l(\mathbf{z}) = W^{(l)}\mathbf{z} + \mathbf{b}^{(l)}$ with $W^{(l)} \in \mathbb{R}^{C_l \times C_{l-1}}$ and $\mathbf{b}^{(l)} \in \mathbb{R}^{C_l}$. Here $C_l$ denotes the width of $l$-th layer.

**CNNs.** The $l$-th hidden state is a feature matrix: $\mathbf{z}^{(l)}(\mathbf{x}) \in \mathbb{R}^{D_l \times C_l}$ with $D_l$ and $C_l$ denoting the spatial dimension and number of channels, respectively. $\mathcal{T}_l : \mathbb{R}^{D_{l-1} \times C_{l-1}} \mapsto \mathbb{R}^{D_l \times C_l}$ is parameterized by a kernel $W^{(l)} \in \mathbb{R}^{C_l \times C_{l-1} \times s}$ and bias $\mathbf{b}^{(l)} \in \mathbb{R}^{C_l}$ as follows

$$(\mathcal{T}_l(\mathbf{z}))_{:,j} = \sum_{i=1}^{C_{l-1}} \mathbf{z}_{:,i} *_s W_{j,i,:}^{(l)} + b_j^{(l)}\mathbf{1}, \quad \text{for } j = 1, \ldots, C_l \tag{1}$$

where $*_s : \mathbb{R}^{sk} \times \mathbb{R}^s \mapsto \mathbb{R}^k$ denotes the *convolution with stride* given by

$$\mathbf{v} *_s \mathbf{w} = (\mathbf{v}_{I_1}^\top \mathbf{w}, \mathbf{v}_{I_2}^\top \mathbf{w}, \ldots, \mathbf{v}_{I_k}^\top \mathbf{w}) \in \mathbb{R}^k, \tag{2}$$

where $I_j := [(j-1)s + 1, js]$ denotes the $j$-th patch, and $\mathbf{v}$ and $\mathbf{w}$ denote the signal and filter, respectively. As a comparison, we also consider the convolution without stride $* : \mathbb{R}^k \times \mathbb{R}^s \mapsto \mathbb{R}^{k-s+1}$ given by $(\mathbf{v} * \mathbf{w})_i = \sum_{j=1}^{s} v_{i+j-1} w_j$. Note that the stride plays a role of downsampling and in practice, it is also common to use other downsampling schemes such as max pooling and average pooling. All results in this paper hold regardless of which one is used and thus, we will focus on the stride case without loss of generality.

**LCNs.** A LCN has the same architecture as its CNN counterpart but lacks weight sharing. Consequently, LCNs have more parameters. Specifically, the linear transform $\mathcal{T}_l : \mathbb{R}^{D_{l-1} \times C_{l-1}} \mapsto \mathbb{R}^{D_l \times C_l}$ is parameterized by $W^{(l)} \in \mathbb{R}^{C_l \times C_{l-1} \times D_{l-1}}$ and $\mathbf{b}^{(l)} \in \mathbb{R}^{D_l \times C_l}$ as follows

$$(\mathcal{T}_l(\mathbf{z}))_{:,j} = \sum_{i=1}^{C_{l-1}} \mathbf{z}_{:,i} \star_s W_{j,i,:}^{(l)} + \mathbf{b}_{:,j}^{(l)}, \quad \text{for } j = 1, \ldots, C_l$$

where the local linear operater $\star_s : \mathbb{R}^{ks} \times \mathbb{R}^{ks} \mapsto \mathbb{R}^k$ is defined by $\mathbf{v} \star_s \mathbf{w} = (\mathbf{v}_{I_1}^\top \mathbf{w}_{I_1}, \mathbf{v}_{I_2}^\top \mathbf{w}_{I_2}, \ldots, \mathbf{v}_{I_k}^\top \mathbf{w}_{I_k})$, where $I_j = [(j-1)s + 1, js]$ denotes the indices of $j$-th patch.

Throughout this paper, we always assume the filter size $s = 2$ for technical simplicity and thus $D_l = 4d/2^l = D_{l-1}/2$ for both CNNs and LCNs.

**Regularizer.** To regularize CNNs and LCNs, we consider following $\ell_2$-type norm:

$$\|\theta\|_{\mathcal{P}} := \|W_o\|_2 + \sum_{l=1}^{L} (\|W^{(l)}\|_F + \alpha_l \|\mathbf{b}^{(l)}\|_F), \tag{3}$$

where $\alpha_l = \sqrt{D_l}$ for CNNs and $\alpha_l = 1$ for LCNs. The factor $\alpha_l$ is introduced such that $\|\theta\|_{\mathcal{P}}$ can control the Lipschitz norm of $h_\theta$, thereby yielding effective capacity controls for CNNs and LCNs. See Appendix C for more details.

## 3 Universal Approximation

The following theorem shows that deep ReLU CNNs are *universal* as long as they are logarithmically deep with respect to the input dimension and the proof is deferred to Appendix D.1.

**Theorem 3.1** (Universality). *Consider CNNs with all activation functions to be* ReLU. *Suppose $L = \log_2(4d)$ and $C_l = 2^{l+1}$ for $l \in [L-1]$ to be fixed and allow the number of channels of the last layer $C_L$ to increase. Then, the CNNs are universal: for any $\epsilon > 0$, any compact set $\Omega \subset \mathbb{R}^{4d}$, and any $h^* \in C(\Omega)$, there exists a CNN $h_\theta$ such that $\sup_{\mathbf{x} \in \Omega} |h_\theta(\mathbf{x}) - h^*(\mathbf{x})| \leqslant \epsilon$.*

**Proof idea.** Write $h_\theta = \mathcal{M}_o \circ \sigma \circ \mathcal{T}_L \circ \mathbf{z}^{(L-1)}$. First, we show there exists parameters (independent of $h^*$) such that

$$\mathbf{z}^{(L-1)}(\mathbf{x}) = \begin{pmatrix} \sigma(x_1) & \sigma(-x_1) & \sigma(x_2) & \sigma(-x_2) & \cdots & \sigma(x_{2d}) & \sigma(-x_{2d}) \\ \sigma(x_{2d+1}) & \sigma(-x_{2d+1}) & \sigma(x_{2d+2}) & \sigma(-x_{2d+2}) & \cdots & \sigma(x_{4d}) & \sigma(-x_{4d}) \end{pmatrix} \in \mathbb{R}^{2 \times (4d)}.$$

This implies that after $L - 1$ layers, the spatial dimension is reduced to 2 but the spatial information is stored to different channels in the form of $\{\sigma(x_i), \sigma(-x_i)\}$. Comparing $\mathbf{z}^{(L-1)}(\mathbf{x})$ with the input $\mathbf{x}$, there is no information loss since $x_i = \sigma(x_i) - \sigma(-x_i)$ for any $i \in [4d]$. Then, the universality can be established by simply showing that $\mathcal{M}_o \circ \sigma \circ \mathcal{T}_L$ can simulate any two-layer ReLU networks.

**The synergy between multichanneling and downsampling.** The key to achieving universality with a depth of $\mathcal{O}(\log d)$ lies in a unique synergy between multichanneling and downsampling. Downsampling can expand the receptive field at an exponential rate, enabling CNNs to capture long-range correlations with only $\mathcal{O}(\log d)$ depth. Meanwhile, multichanneling prevents information loss whenever downsampling operations reduce the spatial dimensions. It is worth noting that this specific collaboration between multichanneling and downsampling has been adopted in most practical CNNs, such as VGG Nets (Simonyan and Zisserman, 2014) and ResNets (He et al., 2016). Our universality analysis provides theoretical support for this widespread architectural choice.

The following proposition further shows that the depth requirement in Theorem 3.1 is optimal, whose proof can be found in Appendix D.2.

**Proposition 3.2.** *Let $\mathcal{X} = [0, 1]^{4d}$ and consider the target function $h^*(\mathbf{x}) = x_1 x_{2d+1}$. If $L \leqslant \log_2(4d) - 1$, then for any $C_l \in \mathbb{N}$ for $l \in [L]$ and any activation functions $\{\sigma_l\}_{l=1}^L$, we have $\inf_\theta \sup_{\mathbf{x} \in \mathcal{X}} |h_\theta(\mathbf{x}) - h^*(\mathbf{x})| \geqslant \frac{1}{8}$.*

The intuition behind this is straightforward: If the depth of the CNN is less than $\log_2(4d)$, the size of the receptive field will not exceed $2d$. Consequently, functions that encode longer-range correlations cannot be accurately approximated.

**The cruciality of downsampling.** The following proposition shows that without downsampling, a minimum depth of $\Omega(d)$ is necessary for achieving universality. This highlights the importance of downsampling as it enables universality with logarithmic depth (Theorem 3.1).

**Proposition 3.3.** *We temporarily use $h_\theta$ to denote the CNN without downsampling. Let $\mathcal{X} = [0, 1]^{4d}$ and $h^*(\mathbf{x}) = x_1 x_{4d}$. If $L \leqslant 4d - 2$, then for any $C_l \in \mathbb{N}$ for $l \in [L]$ and any activation functions $\{\sigma_l\}_{l=1}^L$, we have $\inf_\theta \sup_{\mathbf{x} \in \mathcal{X}} |h_\theta(\mathbf{x}) - h^*(\mathbf{x})| \geqslant \frac{1}{8}$.*

The proof is presented in Appendix D.3. The reason behind is simple: vanilla convolution (without stride) can only capture local correlations of length 2 (since our filter size is $s = 2$). Stacking $L$ layers of vanilla convolutions without downsampling will only allow the network to capture correlations of length $L + 1$.

## 4 Efficient Learning of Sparse Functions

**Definition 4.1** (Sparse function). A function $f : \mathbb{R}^d \mapsto \mathbb{R}$ is said to be sparse if $f$ only depends on a few coordinates. Given $k \in \mathbb{N}$ with $k \ll d$, $f$ is said to be $k$-sparse if there exist an index set $\mathcal{I} = \{i_1, \ldots, i_k\} \subset [d]$ and $g : \mathbb{R}^k \mapsto \mathbb{R}$ such that $f(\mathbf{x}) = g(\mathbf{x}_\mathcal{I})$, where $\mathbf{x}_\mathcal{I} = (x_{i_1}, \ldots, x_{i_k}) \in \mathbb{R}^k$.

The class of sparse functions includes functions of both functions with short-range correlations, such as $f(\mathbf{x}) = g(x_1, x_2)$, and those with long-range correlations, like $f(\mathbf{x}) = g(x_1, x_d)$. It is widely held that it is difficult for CNNs to capture long-range correlations due to locality bias. Consequently, it might seem that CNNs are not well-suited to learning sparse functions like $\mathbf{x} \mapsto g(x_1, x_d)$. However, for CNNs with downsampling, increasing depth can expand the receptive field exponentially, providing the opportunity to learn long-range correlations. Indeed this has been proven in the universality analysis.

In this section, we further show that deep CNNs are not only capable of, but also efficient at learning long-range sparse functions. The term "efficient" refers that the sample complexity depends on the input dimension logarithmically. This is because deep CNNs can effectively identify any $k$ critical coordinates using only $\mathcal{O}(k^2 \log d)$ parameters as demonstrated by the following lemma.

**Lemma 4.2** (Adaptive coordinate selection for Linear CNNs). *Let $\mathcal{I} = (i_1, i_2, \ldots, i_k) \subset [d]$. For the linear CNN model $h_\theta$ with $C_l = k$ for $l \in [L]$ and $L = \log_2(d)$. Then, there exist parameters (depending on $\mathcal{I}$) such that $\mathbf{z}^{(L)}(\mathbf{x}) = (x_{i_1}, x_{i_2}, \ldots, x_{i_k})$.*

*Proof.* First, consider the case of $k = 1$ where $\mathcal{I} = (i)$. Let $i - 1 = \sum_{l=0}^{L-1} a_l 2^l$ be the binary representation of $i$. Set $W_{l,l,:} = (1 - a_l, a_l)$ for $l = [L]$ and all other parameters (including the bias) to zero. Then, it is easy to verify that $\mathbf{z}^{(L)}(\mathbf{x}) = (x_i)$; see Figure 1 for a diagram illustration.

For the case of $k > 1$, set the cross-channel weights to zero; for each channel, follow the case of $k = 1$ to set weights and bias. Under this setup, different channels have no interaction and proceed in

a completely independent way. As a result, each channel selects one critical coordinate, and thus, $\mathbf{z}^{(L)}(\mathbf{x}) = (x_{i_1}, x_{i_2}, \ldots, x_{i_k})$. □

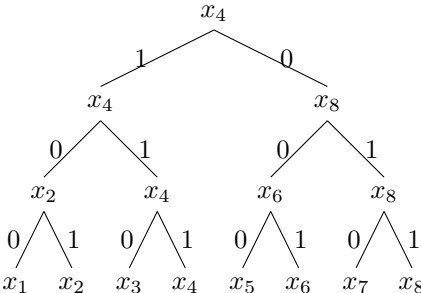

Figure 1: A diagram illustration of how CNNs select coordinates adaptively. In this case $d = 8, L = 3$. The nonzero coordinate is $i = 4$, for which $a_0 = 1, a_1 = 1, a_2 = 0$. The values on edges represent the weights, which are set according to the proof of Lemma 4.2.

**Remark 4.3.** *Lemma 4.2 indicates that deep CNNs are able to effectively identify any $k$ critical coordinates with $\mathcal{O}(\log d)$ depth and $\mathcal{O}(k^2 \log d)$ parameters, which is significantly smaller than $d$. The key to this achievement is the adaptivity of neural networks, in combination with weight sharing, multichanneling, downsampling, and depth, as demonstrated in the proof. Specifically, downsampling and increased depth allow for capturing long-range sparse correlations with only $\mathcal{O}(\log d)$ depth, multichanneling facilitates the storage of information regarding different critical coordinates, and weight sharing ensures the number of parameters of each layer to be independent of $d$.*

We now proceed to consider the learning of nonlinear sparse functions. We first need the following feature selection result for ReLU CNNs. The proof is similar to the linear case (Lemma 4.2) and deferred to Appendix E.1.

**Lemma 4.4.** *Given $k, m \in \mathbb{N}$, consider a* ReLU *CNN with depth $L = \log_2(4d)$ and the channel numbers $C_l = 2k$ for all $l \in [L-1]$ and $C_L = m$. Then for any $\mathcal{I} = (i_1, \ldots, i_k) \subset [4d]$, $\mathbf{u}_1, \ldots, \mathbf{u}_m \in \mathbb{R}^k$, and $c_1, \ldots, c_m \in \mathbb{R}$, there exists $\theta \in \Theta$ such that the $L$-th layer outputs:*

$$\mathbf{z}^{(L)}(\mathbf{x}) = \left( \sigma(\mathbf{u}_1^\top \mathbf{x}_\mathcal{I} + c_1), \ldots, \sigma(\mathbf{u}_m^\top \mathbf{x}_\mathcal{I} + c_m) \right) \in \mathbb{R}^{1 \times m}. \tag{4}$$

*Furthermore, this CNN has $\mathcal{O}(k^2 \log d + km)$ parameters.*

According to this lemma, deep CNNs are capable of generating adaptive features of the form: $\mathbf{x} \mapsto \sigma(\mathbf{u}^\top \mathbf{x}_\mathcal{I} + c)$ with only $\mathcal{O}(\log d)$ depth and $\mathcal{O}(k^2 \log d)$ parameters, where the features depend only on the $k$ critical coordinates. Note that linear combinations of this type of features give two-layer networks. Therefore, functions of the form $f(\mathbf{x}) := g(\mathbf{x}_\mathcal{I})$ with $g : \mathbb{R}^k \mapsto \mathbb{R}$ can be well approximated by deep CNNs, as long as $g$ can be wll approximated by two-layer ReLU networks. To measure the learnability of $g$, we adopt the Barron regularity as proposed in E et al. (2022).

**Definition 4.5** (Barron space). Consider functions admitting the following integral representation $g_\rho(\mathbf{x}) = \int_\Omega a\sigma\left(\mathbf{u}^\top \mathbf{x} + c\right) \rho(\mathrm{d}a, \mathrm{d}\mathbf{u}, \mathrm{d}c)$ for all $\mathbf{x} \in \mathcal{X}$, where $\sigma = \mathrm{ReLU}$, $\Omega = \mathbb{R}^1 \times \mathbb{R}^k \times \mathbb{R}^1$ and $\rho \in \mathcal{P}(\Omega)$. For a function $g : \mathcal{X} \mapsto \mathbb{R}$, denote by $A_g = \{\rho : g_\rho(\mathbf{x}) = g(\mathbf{x}) \text{ for all } \mathbf{x} \in \mathcal{X}\}$. Then, we define

$$\|g\|_\mathcal{B} = \inf_{\rho \in A_f} \mathbb{E}_{(a, \mathbf{u}, c) \sim \rho}\left[|a|\left(\|\mathbf{u}\|_1 + |c|\right)\right], \qquad \mathcal{B} = \{g : \|g\|_\mathcal{B} < \infty\}.$$

More explicitly, the Fourier-analytic characterization (Barron, 1993; Klusowski and Barron, 2016) showed that $\|g\|_\mathcal{B} \lesssim \int (1 + \|\xi\|_1)^2 |\hat{g}(\xi)| \, \mathrm{d}\xi$, where $\hat{g}$ denotes the Fourier transform of $g$. The analysis with Barron regularity is often much simpler (E et al., 2019; 2022) and can yield a better dependence on $k$ than using traditional smoothness measures. Specifically, the Barron regularity allows obtaining generalization bounds with dimension-independent rates like $\mathcal{O}(n^{-1/2})$. In contrast, for traditional smoothness measures such as assuming $g \in C^s([0,1]^k)$, the resulting error rate would scale like $\mathcal{O}(n^{-s/k})$, which depends on $k$ exponentially.

Recall $\ell_B(y, y') = \frac{1}{2}(y - y')^2 \wedge \frac{1}{2}B^2$. Consider the regularized estimator given by $\hat{\theta}_n = \mathrm{argmin}_\theta \left( \frac{1}{n} \sum_{i=1}^n \ell_B(\pi_A \circ h_\theta(\mathbf{x}_i), y_i) + \lambda \|\theta\|_\mathcal{P} \right)$, where $A, B, \lambda$ are hyperparameters to be tuned.

**Theorem 4.6.** *Let* $\mathcal{X} = [0,1]^{4d}$ *and* $h^*(\mathbf{x}) := g(\mathbf{x}_{\mathcal{I}})$ *for some* $\mathcal{I} \subset [4d]$ *be the target function. Let* $k = |\mathcal{I}|$ *and suppose* $g \in \mathcal{B}$, $\sup_z |g(z)| \leqslant 1$, *and* $d \geqslant 3$. *For any* $\delta \in (0, 1/2), \epsilon \in (0, 1/2)$, *there is a choice of* $A, B$, *and* $\lambda$ *such that w.p. at least* $1 - 2\delta$ *over the sampling of training set we have* $\left\| \pi_A \circ h_{\hat{\theta}_n} - h^* \right\|_{L^2(P)}^2 \leqslant \epsilon$, *whenever*

$$n \geqslant \mathrm{poly}(\|g^*\|_{\mathcal{B}}, k, \sigma, \log\frac{1}{\delta}, \log\frac{1}{\epsilon}) \left( \frac{\log(d)(\log\log d)^3}{\epsilon^3} + \frac{\log^2(d)(\log\log d)^3}{\epsilon^2} \right).$$

The proof is presented in Appendix E.2. This theorem shows that learning sparse functions with deep CNNs requires only $\widetilde{\mathcal{O}}(\log^2 d)$ samples, which is nearly optimal from an information-theoretic perspective. This is because Han and Yuan (2020) proved that learning sparse functions requires at least $\Omega(\log d)$ samples. It is also important to mention that Cagnetta et al. (2022) showed that using deep convolutional kernels to learn long-range sparse functions suffers the curse of dimensionality. The comparison with our results also highlights the significance of *adaptivity* in neural networks.

**Experimental validation.** In this experiment, both short-range and long-range sparse target functions are considered. We set the input dimension $d = 4096$, the sample size $n = 400$ and the noise level $\sigma$ to be zero. For the CNN architecture, the filter size is $s = 4$, resulting in a depth $L = \log_4(d) = 6$; the number of channels is set to $C = 4$ across all layers. The Adam optimizer is employed to our models, and importantly, no regularization is applied. As a comparison, we also examine two-layer fully-connected networks (FCNs) with a width 10, as well as the ordinary least linear regression (OLS). The results are shown in Figure 2. One can see clearly that even without any explicit sparsity regularization, CNN can still learn sparse interactions efficiently in both short-range and long-range scenarios. In contrast, FCN and OLS overfit the data and fail to generalize to test data.

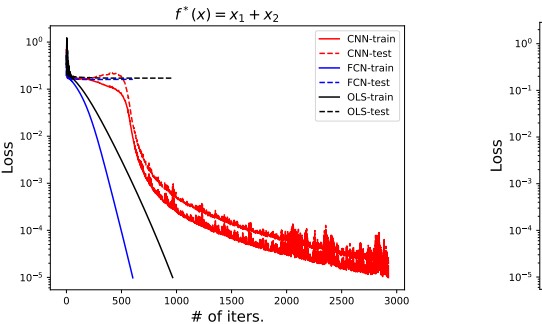 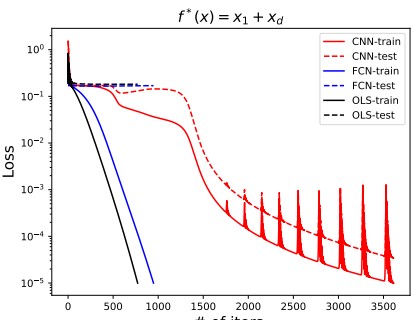

Figure 2: CNN can learn sparse functions efficiently. Both short-range (**left**) and long-range (**right**) sparse target functions are considered in this experiment. The training is stopped when the training loss drops below $10^{-5}$.

# 5 Disentangle the Inductive Biases of Weight Sharing and Locality

To facilitate our statement, in this section, we denote by $h_\theta^{\mathrm{cnn}}, h_\theta^{\mathrm{lcn}}, h_\theta^{\mathrm{fcn}}$ the CNN, LCN, and FCN model, respectively. We shall consider the following task.

**Separation Task.** Suppose the input distribution $P = \mathcal{N}(0, I_{4d})$ and the target function $\bar{h}^*(\mathbf{x}) = \pi_{A_0} \circ h^*(\mathbf{x})$ where $A_0$ is a universal absolute constant to be specified later and

$$h^*(\mathbf{x}) = \frac{1}{d} \left( \sum_{i=1}^d (x_{2i-1}^2 - x_{2i}^2) \right) \left( \sum_{i=1}^d (x_{2d+2i-1}^2 - x_{2d+2i}^2) \right). \tag{5}$$

The truncation is employed to ensure boundedness of the output. However, we believe that a more refined analysis could eliminate the need for this constraint. This target function possesses the structures of both "weight sharing" and "locality" (the global sum can be computed hierarchically through local additions). We then consider the following comparisons.

- CNNs vs. LCNs. This comparison allows us to isolate the bias of weight sharing as both models have the same structures, but LCNs do not have weight sharing.

- LCNs vs. FCNs. This comparison allows us to isolate the bias of locality since the only difference between FCNs and LCNs is the presence or absence of locality.

By these comparisons, we can evaluate the effectiveness of each bias in contributing to the improved performance of CNNs over FCNs, and gain insights into how these biases interact with other structures such as depth, multichanneling, and downsampling.

To establish provable separations, we need both upper and lower bounds of the sample complexity of learning $\bar{h}^*$. For the upper bounds, we consider the regularized estimator given by

$$\hat{\theta}_n = \underset{\theta}{\operatorname{argmin}} \left( \frac{1}{n} \sum_{i=1}^{n} [\ell_B(\pi_A \circ h_\theta(\mathbf{x}_i), y_i)] + \lambda \|\theta\|_\mathcal{P} \right), \tag{6}$$

where the model $h_\theta$ and hyperparameters $A$, $B$, and $\lambda$ will be specified for each comparison. We will use the covering number-based technique (see Appendix B.2) to upper bound the generalization error. See also Appendix C for how to bound the covering numbers of deep CNNs and deep LCNs.

Establishing lower bounds is often much more challenging. We will adopt a similar approach to Ng (2004); Li et al. (2020); Abbe and Boix-Adserà (2022), which involves understanding the learning hardness through the equivariance group of the learning algorithm.

### 5.1 Learning Algorithm, Group Equivariance, and Lower Bounds

A learning algorithm $\mathbb{A} : (\mathcal{X} \times \mathcal{Y})^n \mapsto \mathcal{M}(\Theta)$ generates a random variable taking values in $\Theta$ by using the training set $S_n$. This definition includes the situation where our models are returned by stochastic optimizers such as SGD and Adam. We will denote $X \overset{\mathrm{d}}{=} Y$ as $\mathrm{Law}(X) = \mathrm{Law}(Y)$.

**$G$-equivariant/-invariant.** Let $G$ be a group acting on $\mathcal{X}$. A learning algorithm $\mathbb{A}$ is said to be $G$-equivariant[1] if $\forall \{\mathbf{x}_i, y_i\}_{i=1}^n \in (\mathcal{X} \times \mathcal{Y})^n$ and $\forall \tau \in G, \mathbf{x} \in \mathcal{X}$:

$$h_{\mathbb{A}(\{\tau(\mathbf{x}_i), y_i\}_{i=1}^n)} \circ \tau \overset{\mathrm{d}}{=} h_{\mathbb{A}(\{\mathbf{x}_i, y_i\}_{i=1}^n)}. \tag{7}$$

A distribution $\mu$ is said to be $G$-invariant if for any $\tau \in G$, $\tau(X) \sim \mu$ if $X \sim \mu$.

**Sample Complexity.** Given a target function class $\mathcal{F}$ and a learning algorithm $\mathbb{A}$, for a target accuracy $\epsilon > 0$, the sample complexity $\mathcal{C}(\mathbb{A}, \mathcal{F}, \epsilon)$ is defined as the smallest integer $n$ such that

$$\sup_{h^* \in \mathcal{F}} \mathbb{E} \left\| h_{\mathbb{A}(S_n)} - h^* \right\|_{L^2(P)}^2 \leqslant \epsilon, \tag{8}$$

where the expectation is taken over both the sampling of $S_n$ and the randomness in $\mathbb{A}$. In addition, we define the *minimax sample complexity* by $\bar{\mathcal{C}}(\mathcal{F}, \epsilon) := \inf_{\mathbb{A}} \mathcal{C}(\mathbb{A}, \mathcal{F}, \epsilon)$, where the infimum is taken over all learning algorithms, i.e., all the mappings: $(\mathcal{X} \times \mathcal{Y})^n \mapsto \mathcal{M}(\Theta)$.

For a function class $\mathcal{F}$, denote by $\mathcal{F} \circ G := \{f \circ \tau : f \in \mathcal{F}, \tau \in G\}$ the $G$-enlarged class.

**Lemma 5.1.** *(A restatement of (Li et al., 2020, Lemma D.1)) Let $\mathcal{A}_G$ be the set of all $G$-equivariant algorithms. For any function class $\mathcal{F}$ and $\epsilon > 0$, it holds that $\inf_{\mathbb{A} \in \mathcal{A}_G} \mathcal{C}(\mathbb{A}, \mathcal{F}, \epsilon) \geqslant \bar{\mathcal{C}}(\mathcal{F} \circ G, \epsilon)$.*

This lemma shows that the sample complexity of learning $\mathcal{F}$ with a $G$-equivalent algorithm can be lower bounded by the minimax complexity of learning *the enlarged class*: $\mathcal{F} \circ G$. The latter can be handled using standard approaches from minimax theory (see (Wainwright, 2019, Section 15)). Specifically, we will adopt Fano's method to lower-bound the minimax sample complexity, which reduces the problem to find a proper packing of the enlarged class $\mathcal{F} \circ G$ (see Appendix B.3). In particular, when $\mathcal{F} = \{f^*\}$, the packing number of $\mathcal{F} \circ G$ can be completely determined by the packing number of the symmetry group $G$ if $f^*$ is chosen to satisfy

$$\|f^* \circ \tau_1 - f^* \circ \tau_2\|_{L^2(\mu)}^2 \sim \|\tau_1 - \tau_2\|_G^2, \tag{9}$$

where $\|\cdot\|_G$ is a proper norm defined on $G$. Moreover, estimating the packing number of a symmetry group is often much easier.

---

[1]The definition here is intuitive but not fully rigorous. We refer to Appendix F.1 for a rigorous definition.

Note that the specific target function (5) in our separation task is designed to satisfies (9) for the symmetry groups associated with SGD algorithms and thus, the sample complexity can be completely governed by the size of symmetry groups. Specifically, for SGD algorithms, we have

- The symmetry group of FCNs is the orthogonal group $G_{\mathrm{ort}}(4d)$, whose degree of freedom is $\Theta(d^2)$. This explains the lower bound $\Omega(d^2)$ for FCNs.
- The symmetry group of LCNs is the local permutation group $G_{\mathrm{loc}}$, whose degree of freedom is $\Theta(d)$. This explains the lower bound $\Omega(d)$ for LCNs.

See the sections below for more details.

## 5.2 CNNs vs. LCNs

In this section, we will derive an upper bound of sample complexity for learning $\bar{h}^*$ with CNNs and a lower bound for learning $\bar{h}^*$ with LCNs. The two bounds together provide a quantification of the effect of weight sharing.

Consider the deep CNN $h_\theta^{\mathrm{cnn}}$ with $L = \log_2(4d)$, $C_1 = C_L = 4$, $C_l = 2$ for $l \in [2, L-1]$, and $\sigma_1(x) = \sigma_L(x) = \mathrm{ReLU}^2(x), \sigma_l(x) = \mathrm{ReLU}(x)$ for $l \in [2, L-1]$.

**Theorem 5.2** (Upper bound of CNNs). *Suppose $d \geqslant 3$ and $A_0 \geqslant 0$ and consider the estimator $\hat{\theta}_n$ given by (6). For any $\epsilon \in (0, 1/2)$ and $\delta \in (0, 1/2)$, there is a choice of $A, B$, and $\lambda$ such that $\left\| \pi_A \circ h_{\hat{\theta}_n}^{\mathrm{cnn}} - \bar{h}^* \right\|_{L^2(P)}^2 \leqslant \epsilon$ holds w.p. at least $1 - 2\delta$, whenever*

$$n \geqslant \mathrm{poly}(A_0, \sigma, \log(1/\delta), \log(1/\epsilon))\epsilon^{-2}(\log^2 d)(\log\log d)^3 .$$

The proof is deferred to Appendix G.2. It is shown that CNNs need only $\widetilde{\mathcal{O}}(\log^2 d)$ samples to learn $\bar{h}^*$. The reason behind this upper bound is that $\bar{h}^*$ can be accurately represented by deep CNNs with only $\mathcal{O}(\log d)$ parameters and well-controlled norms due to the well matching between the weight sharing and locality in CNNs and the unique characteristics of $\bar{h}^*$.

Next we turn to establish a lower bound for LCNs. Let $\ell(\cdot, \cdot)$ be a general differentiable loss function and $\hat{L}^{\mathrm{lcn}}(\theta) := \frac{1}{n}\sum_{i=1}^n \ell(h_\theta^{\mathrm{lcn}}(\mathbf{x}_i), y_i) + r\left(\|\theta\|_p\right)$, where $p \geqslant 1$ and $r : [0, +\infty) \to [0, +\infty)$. Denote by $\mathbb{A}_T^{\mathrm{lcn}}$ the algorithm that returns solutions by optimizing $\hat{L}^{\mathrm{lcn}}(\cdot)$ for $T$ steps using SGD or Adam. Under mild conditions, it can be shown that $\mathbb{A}_T^{\mathrm{lcn}}$ is equivariant under the local permutation group:

$$G_{\mathrm{loc}} = \left\{ U \in \mathbb{R}^{4d \times 4d} : U = \mathrm{diag}\left(U_1, U_2, \ldots, U_{2d}\right), U_i \in \left\{ \begin{pmatrix} 1 & 0 \\ 0 & 1 \end{pmatrix}, \begin{pmatrix} 0 & 1 \\ 1 & 0 \end{pmatrix} \right\} \right\}. \quad (10)$$

Obviously, $G_{\mathrm{loc}}$ is a group under matrix product. To ensure the $G_{\mathrm{loc}}$-equivariance of $\mathbb{A}_T^{\mathrm{lcn}}$, we need the following assumption, which is satisfied by all popular initialization schemes used in practice.

**Assumption 1.** *At initialization, the fist layer weight satisfies: $W_{j,1,2k-1}^{(1)}$ and $W_{j,1,2k}^{(1)}$ have the same marginal distribution for any $j \in [C_1], k \in [2d]$.*

**Lemma 5.3.** *Suppose that the input distribution $P$ is $G_{\mathrm{loc}}$-invariant and Assumption 1 is satisfied. Then, for any $T \in \mathbb{N}$, $\mathbb{A}_T^{\mathrm{lcn}}$ is $G_{\mathrm{loc}}$-equivariant for both SGD and Adam.*

The proof is deferred to Appendix F.2. Xiao and Pennington (2022) showed that under Gaussian initialization, the equivariance group of $\mathbb{A}_T^{\mathrm{lcn}}$ is $O(2) \otimes I_{2d}$, which is slightly larger than $G_{\mathrm{loc}}$ obtained in the current work. It should be stressed that $G_{\mathrm{loc}}$ is obtained under a much milder assumption of initialization which holds for the popular uniform initialization scheme. Moreover, we show below that the $G_{\mathrm{loc}}$ equviariance is sufficient for separating LCNs from CNNs.

**Theorem 5.4** (Lower bound of LCNs). *Under Assumption 1, there exists absolute constants $C, c > 0$ such that $\forall T \in \mathbb{N}, \epsilon_0 \in (0, c], d, A_0 \geqslant C$:*

$$\mathcal{C}(\mathbb{A}_T^{\mathrm{lcn}}, \{\bar{h}^*\}, \epsilon_0) = \Omega(\sigma^2 d)$$

The proof is deferred to Appendix G.1. By comparing Theorem 5.2 and 5.4, we see that the weight sharing in CNNs yields an $\widetilde{\Theta}(d)$ improvement on the sample complexity for learning $\bar{h}^*$. This highlights the importance of exploiting the "weight sharing" in $\bar{h}^*$ for efficient learning.

## 5.3 LCNs vs. FCNs

We will derive an upper bound of sample complexity for learning $\bar{h}^*$ with LCNs and a lower bound for learning $\bar{h}^*$ with FCNs. The two bounds together provide a quantification of the effect of locality.

We consider the deep LCN $h_\theta^{\text{lcn}}$ with $L = \log_2(4d)$, $C_1 = C_L = 4$, $C_l = 2$ for $l \in [2, L-1]$, and $\sigma_1(x) = \sigma_L(x) = \text{ReLU}^2(x)$, $\sigma_l(x) = \text{ReLU}(x)$ for $l \in [2, L-1]$.

**Theorem 5.5** (Upper bound of LCNs). *Suppose $d \geqslant 3$ and $A_0 \geqslant 0$ and consider the estimator $\hat{\theta}_n$ given by (6). Then, for any $\epsilon \in (0, 1/2)$ and $\delta \in (0, 1/2)$, there is a choice of $A, B,$ and $\lambda$ such that $\left\| \pi_A \circ h_{\hat{\theta}_n}^{\text{lcn}} - \bar{h}^* \right\|_{L^2(P)}^2 \leqslant \epsilon$ holds w.p. at least $1 - 2\delta$, whenever*

$$n \geqslant \text{poly}(A_0, \sigma, \log(1/\delta), \log(1/\epsilon))\epsilon^{-2} d \log^4 d$$

The proof is deferred to Appendix H.2. It is shown that LCNs needs $\widetilde{O}(d)$ samples to learn $\bar{h}^*$. The reason behind this upper bound is that $\bar{h}^*$ can be well approximated by LCNs with $\mathcal{O}(d)$ parameters and well-controlled norm. However, as opposed to CNNs, LCNs lack weight sharing and can only imitate the local features of $\bar{h}^*$ with the "weight sharing" structure completely ignored.

Next we turn to establish a lower bound for FCNs. Let $\ell(\cdot, \cdot)$ be a general differentiable loss function. Denote by $\mathbb{A}_T^{\text{fcn}}$ the algorithm: Run SGD for $T$ steps by optimizing $\hat{L}^{\text{fcn}}(\theta) := \frac{1}{n} \sum_{i=1}^n \ell(h_\theta^{\text{fcn}}(\mathbf{x}_i), y_i) + r\left(\|\theta\|_2\right)$, where $r : [0, +\infty) \to [0, +\infty)$ is a general penalty function. We refer to Appendix F.1 for a rigorous definition.

**Lemma 5.6.** *Suppose that the input distribution $P$ is $G_{\text{ort}}(4d)$-invariant and entries of $W^{(1)}$ are i.i.d. initialized from $\mathcal{N}(0, \beta^2)$ for some $\beta > 0$. Then, for any $T \in \mathbb{N}$, $\mathbb{A}_T^{\text{fcn}}$ is $G_{\text{ort}}(4d)$-equivariant.*

This lemma was proved in (Li et al., 2020, Corollary C.2) and we state it here for completeness. It implies that training FCNs with SGD induces a larger equivariance group than that of LCNs since $G_{\text{loc}} \subset G_{\text{ort}}(4d)$. It is important to note that this lemma only holds for SGD, as Adam is only permutation invariant. In contrast, the result in Lemma 5.3 applies to both SGD and Adam.

**Theorem 5.7** (Lower bound of FCNs). *Suppose that the input distribution $P$ is $G_{\text{ort}}(4d)$-invariant and entries of $W^{(1)}$ are i.i.d. initialized from $\mathcal{N}(0, \beta^2)$ for some $\beta > 0$. Then there exists absolute constants $C, c > 0$ such that $\forall T \in \mathbb{N}, \epsilon_0 \in (0, c], d, A_0 \geqslant C$:*

$$\mathcal{C}(\mathbb{A}_T^{\text{fcn}}, \{\bar{h}^*\}, \epsilon_0) = \Omega(\sigma^2 d^2)$$

The proof is deferred to Appendix H.1. By comparing Theorem 5.5 and 5.7, we see that the locality itself yields another $\widetilde{\Theta}(d)$ improvement on the sample complexity for learning $\bar{h}^*$. This highlights the importance of exploiting the locality in $\bar{h}^*$ for efficient learning.

## 6   Conclusion

In this paper, we delve into the theoretical analysis of the inductive biases associated with the multichanneling, downsampling, weight sharing, and locality in deep CNN architectures with a focus on understanding the interplay between network depth and these biases. Our results highlight the critical role of multichanneling and downsampling in enabling deep CNNs to effectively capture long-range correlations. We also analyze the effects of weight sharing and locality on breaking the learning algorithm's symmetry, through which we make a clear distinction between the two biases: the global orthogonal equivariance vs. the local permutation equivariance. By leveraging these symmetries, we establish provable separations for FCNs vs. LCNs and LCNs vs. CNNs, providing a strong theoretical support for the unique nature of weight sharing and locality.

## Acknowledgements

Lei Wu is supported in part by the National Key R&D Program of China (No. 2022YFA1008200).

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

# Appendix

# A  Related Works

**Approximation and estimation.**  Zhou (2020b) established the first universality theorem for CNNs, which, however, requires the depth to increase as approximation accuracy improved. Later, He et al. (2022) showed a depth of $\mathcal{O}(d)$ is sufficient when multichanneling is utilized. In this paper, we further reduce the required depth to $\mathcal{O}(\log d)$ by combining multichanneling with downsampling. We also prove that without downsampling, CNNs require a minimum depth of $\Omega(d)$ to achieve universality, highlighting the necessity of downsampling for deep CNNs.

Deza et al. (2020) designed some synthetic tasks to show the benefit of depth in CNNs. Okumoto and Suzuki (2022) analyzed dilated CNNs (Yu and Koltun, 2016) by considering target functions with mixed and anisotropic smoothness. They showed that dilated CNNs can capture long-range sparse signals if the dilation rate is on the order of $\Omega(d)$. In contrast, we show that for a similar task, vanilla CNNs can capture the long-range sparse correlations with only $\mathcal{O}(\log d)$ depth and $\widetilde{\mathcal{O}}(\log^2 d)$ samples if downsampling is appropriately combined with multichanneling.

Long and Sedghi (2019); Lin and Zhang (2019) established the upper bounds of the covering number and Rademacher complexity of deep CNNs, where the effect of weight sharing was incorporated. We design concrete tasks to instantiate these bounds to demonstrate the superiority of CNNs over other architectures such as FCNs. Beyond that, we also improve the covering number bound of Long and Sedghi (2019) in various aspects, including improved dependence on parameter norms (from exponential to polynomial). See Appendix C.3 for a detailed discussion.

**Understanding the inductive bias.**  Mei et al. (2021); Bietti et al. (2021) studied the superiority of convolutional kernels by exploiting the translation invariance of convolution. Later, under similar setups, Misiakiewicz and Mei (2022) studied the benefit of using local kernels in convolutions and the effect of downsampling; Bietti (2021); Cagnetta et al. (2022) studied the inductive bias of kernel composition (mimicking the effect of depth in neural networks). However, these studies are limited to kernel methods and it is generally unclear how the results are relevant for understanding CNNs. In addition, Gunasekar et al. (2018); Jagadeesan et al. (2022); Razin et al. (2022); Jiang et al. (2021); Du et al. (2018) studied the inductive biases in linear CNNs. By contrast, we examine vanilla nonlinear CNNs and show that the nonlinearity of activation functions and the network adaptivity are critical for reaping the benefits of the CNN-specific structures.

We highlight the work Li et al. (2020), where a provable separation between CNNs and FCNs was established by exploiting the equivariance groups of the learning algorithm such as SGD and Adam. However, Li et al. (2020) neither disentangled the effects of locality and weight sharing nor studied the effects of increasing network depth, multichanneling, and downsampling. Xiao and Pennington (2022) took a similar idea to explain the inductive bias of CNNs and other architectures but the authors focused on empirical investigations without providing theoretical guarantees. It is worth noting that the idea of studying a learning algorithm through its equivariance group was first introduced in Ng (2004) for linear regression and was also applied to study other hardness problems in deep learning (Abbe and Boix-Adserà, 2022).

# B  Technical Background

## B.1  Toolbox for Concentrations

**Theorem B.1.**  *(Hoeffding's inequality, (Vershynin, 2018, Theorem 2.2.6)) Let $X_1, X_2, \ldots, X_n$ be i.i.d. random variables with mean $\mu$ and satisfy $X_i \in [a, b]$. Then,*

$$\mathbb{P}\left\{\left|\frac{1}{n}\sum_{i=1}^{n} X_i - \mu\right| \geqslant t\right\} \leqslant 2e^{-\frac{2nt^2}{(b-a)^2}}.$$

**Definition B.2** (Sub-exponential Random Variable).  For a random variable $X$, define

$$\|X\|_{\psi_1} = \inf\{t > 0 : \mathbb{E}\exp(|X|/t) \leqslant 2\} \tag{11}$$

to be its sub-exponential norm. A random variable $X$ that satisfies $\|X\|_{\psi_1} < +\infty$ is called a sub-exponential random variable.

**Lemma B.3.** *(Bernstein's inequality, ([Vershynin](), 2018, Corollary 2.8.3)) Let $X_1, \ldots, X_N$ be independent, mean zero, sub-exponential random variables. Then, for every $t \geqslant 0$, we have*

$$\mathbb{P}\left\{\left|\frac{1}{N}\sum_{i=1}^{N}X_i\right| \geqslant t\right\} \leqslant 2\exp\left[-c_2\min\left(\frac{t^2}{K^2}, \frac{t}{K}\right)N\right] \tag{12}$$

*where $K = \max_i \|X_i\|_{\psi_1}$ and $c_2 > 0$ is an absolute constant.*

We will use covering number, packing number, and Rademacher complexity to measure the complexity of a function class.

**Definition B.4** (Covering number). Let $(T, \rho)$ be a metric space. Consider a subset $K \subset T$ and let $\varepsilon > 0$. A subset $K_\varepsilon \subseteq K$ is called an $\varepsilon$-covering of $K$ if every point in $K$ is within distance $\varepsilon$ of some point of $K_\varepsilon$, i.e. $\forall x \in K, \exists x_0 \in K_\varepsilon : \rho(x, x_0) \leqslant \varepsilon$. The smallest possible cardinality of an $\varepsilon$-covering of $K$ is called the covering number of $K$ and is denoted by $\mathcal{N}(K, \rho, \varepsilon)$.

**Definition B.5** (Packing number). Let $(T, \rho)$ be a metric space and $K \subset T$. Given $\varepsilon > 0$, $K_\varepsilon \subset K$ is said to be an $\varepsilon$-packing of $K$ if $\rho(x, y) > \varepsilon$ for all distinct points $x, y \in K_\varepsilon$. The largest possible cardinality of an $\varepsilon$-packing of $K$ is called the packing number of $K$ and is denoted by $\mathcal{P}(K, \rho, \varepsilon)$.

**Definition B.6** (Rademacher complexity). The empirical Rademacher complexity of a function class $\mathcal{F}$ on finite samples is defined as

$$\widehat{\mathrm{Rad}}_n(\mathcal{F}) = \mathbb{E}_\xi\left[\sup_{f \in \mathcal{F}} \frac{1}{n}\sum_{i=1}^{n}\xi_i f(X_i)\right] \tag{13}$$

where $\xi_1, \xi_2, \ldots, \xi_n$ are i.i.d. Rademacher random variables: $\mathbb{P}(\xi_i = 1) = \mathbb{P}(\xi_i = -1) = \frac{1}{2}$. Let $\mathrm{Rad}_n(\mathcal{F}) = \mathbb{E}[\widehat{\mathrm{Rad}}(\mathcal{F})]$ be the population Rademacher complexity.

The following lemma provides an upper bound for the covering number of a general ball.

**Lemma B.7.** *([Long and Sedghi](), 2019, Lemma A.8.) Let $p$ be a positive integer and $\|\cdot\|$ be a norm in $\mathbb{R}^p$. Denote by $B_r = \{\theta \in \mathbb{R}^p : \|\theta\| \leqslant r\}$. Then, $\mathcal{N}(B_r, \|\cdot\|, t) \leqslant \left(\frac{3r}{t}\right)^p$ provided $t \leqslant r$.*

**Lemma B.8** (Covering number of Lipschitz class). *Let $g : \mathcal{X} \times \Theta \mapsto \mathbb{R}$ with $\Theta = \mathbb{R}^p$ be a general parametric model and $\|\cdot\|$ be a norm over $\Theta$. Define $\Theta_r = \{\theta : \|\theta\| \leqslant r\}$ and $\mathcal{G}_r = \{g_\theta : \|\theta\| \leqslant r\}$. Suppose for any $\theta, \theta' \in \Theta_r$, we have $|g_\theta(z) - g_{\theta'}(z)| \leqslant B(z)\|\theta - \theta'\|$. Given $z_1, z_2, \ldots, z_n \in \mathcal{X}$, let $\hat{B}_n = \sqrt{\frac{1}{n}\sum_{i=1}^{n}B^2(z_i)}$ and $\hat{\rho}_n(f, g) = \sqrt{\frac{1}{n}\sum_{i=1}^{n}(f(z_i) - g(z_i))^2}$ for any $f, g \in \mathcal{F}$. Then for $t \leqslant r\hat{B}_n$, we have*

$$\mathcal{N}(\mathcal{G}_r, \hat{\rho}_n, t) \leqslant \left(\frac{3\hat{B}_n r}{t}\right)^p. \tag{14}$$

*Proof.* Note that

$$\hat{\rho}_n(g_\theta, g_{\theta'}) = \sqrt{\frac{1}{n}\sum_{i=1}^{n}(g_\theta(z_i) - g_{\theta'}(z_i))^2} \leqslant \sqrt{\frac{1}{n}\sum_{i=1}^{n}B(z_i)^2}\|\theta - \theta'\| = \hat{B}_n\|\theta - \theta'\|.$$

This implies that for any $t > 0$, an $t/\hat{B}_n$-cover of $\Theta_r$ w.r.t. $\|\cdot\|$ induces a $t$-cover of $\mathcal{G}_r$ w.r.t. $\hat{\rho}_n$. Therefore,

$$\mathcal{N}(\mathcal{G}_r, \hat{\rho}_n, t) \leqslant \mathcal{N}(\Theta_r, \|\cdot\|, t/\hat{B}_n) \leqslant \left(\frac{3r}{t/\hat{B}_n}\right)^p,$$

where the last step follows from Lemma B.7. Thus, we complete the proof. $\qquad\square$

Then we recall the uniform law of large number via Rademacher complexity, which can be found in ([Wainwright](), 2019, Theorem 4.10).

**Lemma B.9.** *Assume that $f$ ranges in $[0, R]$ for all $f \in \mathcal{F}$. For any $n \geqslant 1$, for any $\delta \in (0, 1)$, w.p. at least $1 - \delta$ over the choice of the i.i.d. training set $S = \{X_1, \ldots, X_n\}$, we have*

$$\sup_{f \in \mathcal{F}}\left|\frac{1}{n}\sum_{i=1}^{n}f(X_i) - \mathbb{E}f(X)\right| \leqslant 2\mathrm{Rad}_n(\mathcal{F}) + R\sqrt{\frac{\log(4/\delta)}{n}} \tag{15}$$

Next, the Dudley's integral entropy bound given below shows that the Rademacher complexity can be further upper bounded by using the covering number.

**Lemma B.10.** *Given $n$ samples $X_1, \ldots, X_n$, let $\hat{\rho}_n(f, g) := \sqrt{\frac{1}{n} \sum_{i=1}^{n} (f(X_i) - g(X_i))^2}$ for $f, g \in \mathcal{F}$ and $D = \sup_{f,g \in \mathcal{F}} \hat{\rho}_n(f, g)$ be the diameter of $\mathcal{F}$ with respect to $\hat{\rho}_n$. Then,*

$$\widehat{\mathrm{Rad}}_n(\mathcal{F}) \leqslant 12 \int_0^D \sqrt{\frac{\log \mathcal{N}(\mathcal{F}, \hat{\rho}_n, t)}{n}} \, \mathrm{d}t \tag{16}$$

**Corollary B.11.** *Given $R \geqslant 1$, let $\mathcal{F}$ be a set of functions from $\mathcal{X}$ to $[0, R]$ and $P$ be an arbitrary probability distribution over $\mathcal{X}$. Given $n \in \mathbb{N}$, let $X_1, X_2, \ldots, X_n$ be i.i.d. samples drawn from $P$. Assume that there exists a $\hat{K}_n$, which may depend on the training data $X_1, X_2, \ldots, X_n$, such that $\mathcal{N}(\mathcal{F}, \hat{\rho}_n, t) \leqslant \left( \frac{3\hat{K}_n}{t} \right)^p$ for all $0 < t \leqslant \hat{K}_n$. Then, for any $\delta \in (0, 1)$, w.p. at least $1 - \delta$ we have*

$$\sup_{f \in \mathcal{F}} \left| \frac{1}{n} \sum_{i=1}^{n} f(X_i) - \mathbb{E}f(X) \right| \lesssim R\sqrt{\frac{p \log(\mathbb{E}[\hat{K}_n] + 3)}{n}} + R\sqrt{\frac{\log(4/\delta)}{n}} \tag{17}$$

**Proof.** When $2R \leqslant \hat{K}_n$,

$$
\begin{aligned}
\int_0^D \sqrt{\frac{\log \mathcal{N}(\mathcal{F}, \hat{\rho}_n, t)}{n}} \, \mathrm{d}t &\leqslant \sqrt{\frac{p}{n}} \int_0^{2R} \sqrt{\log(3\hat{K}_n/t)} \, \mathrm{d}t \\
&\leqslant \sqrt{\frac{p}{n}} \int_0^{2R} \left( \sqrt{\log(3\hat{K}_n + 1)} \right) \mathrm{d}t + \sqrt{\frac{p}{n}} \int_0^{2R} \sqrt{\log(1/t)} \, \mathrm{d}t \\
&\leqslant \left( 2R\sqrt{\log(3\hat{K}_n + 1)} + 4R\log(3) \right) \sqrt{\frac{p}{n}} \\
&\lesssim R\sqrt{\frac{p \log(\hat{K}_n + 3)}{n}}
\end{aligned}
\tag{18}
$$

When $2R \geqslant \hat{K}_n$,

$$
\begin{aligned}
\int_0^D \sqrt{\frac{\log \mathcal{N}(\mathcal{F}, \hat{\rho}_n, t)}{n}} \, \mathrm{d}t &\leqslant \int_0^{\hat{K}_n} \sqrt{\frac{\log \mathcal{N}(\mathcal{F}, \hat{\rho}_n, t)}{n}} \, \mathrm{d}t + \int_{\hat{K}_n}^{2R} \sqrt{\frac{\log \mathcal{N}(\mathcal{F}, \hat{\rho}_n, t)}{n}} \, \mathrm{d}t \\
&\leqslant \left( \hat{K}_n \sqrt{\log(3\hat{K}_n + 1)} + 2 \right) \sqrt{\frac{p}{n}} + \int_{\hat{K}_n}^{2R} \sqrt{\frac{p \log(3)}{n}} \, \mathrm{d}t \\
&\leqslant \left( 2R\sqrt{\log(3\hat{K}_n + 1)} + 4R\log(3) \right) \sqrt{\frac{p}{n}} \\
&\lesssim R\sqrt{\frac{p \log(\hat{K}_n + 3)}{n}}
\end{aligned}
\tag{19}
$$

By applying Lemma B.10, we have

$$\mathrm{Rad}_n(\mathcal{F}) = \mathbb{E}[\widehat{\mathrm{Rad}}_n(\mathcal{F})] \lesssim R\,\mathbb{E}\left[ \sqrt{\frac{p \log(\hat{K}_n + 3)}{n}} \right] \leqslant R\sqrt{\frac{p \log(\mathbb{E}[\hat{K}_n + 3])}{n}}, \tag{20}$$

where the last step follows from the Jensen's inequality and the fact that both $\sqrt{z}$ and $\log(z)$ are concave with respect to $z$. By plugging (20) into (15), we complete the proof. $\square$

## B.2 A Generalization Bound for Learning with Unbounded Noise

We first recall our setup: $y_i = h^*(\mathbf{x}_i) + \xi_i$ with $\mathbf{x}_i \overset{iid}{\sim} P$ and $\xi_i \overset{iid}{\sim} \mathcal{N}(0, \sigma^2)$. We assume $\sup_{\mathbf{x} \in \mathcal{X}} |h^*(\mathbf{x})| \leqslant A$ and denote by $h_\theta$ our parametric model.

**Truncation.** Due to $\sup_{\mathbf{x}\in\mathcal{X}}|h^*(\mathbf{x})| \leqslant A$, we can consider the truncated model $\pi_A \circ h_\theta$, which does not lose the expressivity but can make the mathematical analysis much simpler. In addition, due to the noise, the labels are unbounded but the generalization bound in Corollary B.11 requires boundedness. To handle this challenge, we define a truncated loss $\ell_B(y, y') := \frac{1}{2}(y - y')^2 \wedge \frac{1}{2}B^2$. The value of $B$ will be specified later. For the truncated loss, we have

$$|\partial_y \ell_B(y, y')| \leqslant B, \quad |\ell_B(y, y')| \leqslant \frac{B^2}{2}. \tag{21}$$

To facilitate our statements, we define the following truncated risks

$$L_A(\theta) = \mathbb{E}_{\mathbf{x},y}\left[\ell(\pi_A \circ h_\theta(\mathbf{x}), y)\right]$$
$$L_{A,B}(\theta) = \mathbb{E}_{\mathbf{x},y}\left[\ell_B(\pi_A \circ h_\theta(\mathbf{x}), y)\right]$$
$$\hat{L}_{A,B}(\theta) = \frac{1}{n}\sum_{i=1}^n \left[\ell_B(\pi_A \circ h_\theta(\mathbf{x}_i), y_i)\right].$$

In particular,

$$L_A(\theta) = \frac{1}{2}\|\pi_A \circ h_\theta - h^*\|_{L^2(P)}^2 + \frac{1}{2}\sigma^2. \tag{22}$$

**Assumption 2** (Model capacity). *Let $\|\cdot\|$ be a norm over the parameter space $\Theta$ used to control the model capacity. Define $\mathcal{H}_J = \{x \mapsto h_\theta(x) : \|\theta\| \leqslant J\}$. Denote by $p_\mathcal{H}$ the number of parameters. Let $\|\cdot\|_U$ be a norm satisfying that there exists a constant $\alpha_\mathcal{H}$ such that*

$$\|\theta\| \leqslant \alpha_\mathcal{H}\|\theta\|_U, \ \forall \theta \in \Theta. \tag{23}$$

*We make the following assumptions.*

- *Covering number. Suppose that there exists a $\hat{\gamma}_n : \mathbb{R}_{\geqslant 0} \to \mathbb{R}_{\geqslant 0}$ such that*

$$\mathcal{N}(\mathcal{H}_J, \hat{\rho}_n, t) \leqslant \left(\frac{3\hat{\gamma}_n(J)}{t}\right)^{p_\mathcal{H}} \text{ for } t \leqslant \hat{\gamma}_n(J). \tag{24}$$

  *Note that $\hat{\gamma}_n(\cdot)$ may depend on the training data $\mathbf{x}_1, \mathbf{x}_2, \ldots, \mathbf{x}_n$ and we let $\gamma(J) = \mathbb{E}[\hat{\gamma}_n(J)]$*

- *Approximation error. We assume that there is $\theta^* \in \Theta$ such that*

$$\|\pi_A \circ h_{\theta^*} - h^*\|_{L^2(P)}^2 \leqslant \epsilon_* \text{ and } \|\theta^*\|_U \leqslant M_* \tag{25}$$

  *Thus, $L_A(\theta^*) \leqslant (\epsilon^* + \sigma^2)/2$.*

The assumption (25) essentially means that $h^*$ can be well approximated by our model $h_\theta$ with the parameter norm $\|\theta^*\|_U$ well controlled.

Consider the regularized estimator given by

$$\hat{\theta}_n \in \underset{\theta}{\operatorname{argmin}}\left(\hat{L}_{A,B}(\theta) + \lambda\|\theta\|_U\right), \tag{26}$$

where $A, B$, and $\lambda$ are hyperparameters to determined later.

**Remark B.12.** *Note that penalizing the upper bound $\|\theta\|_U$ instead of $\|\theta\|$ is very common in practice. First, if the constant $\alpha_\mathcal{H}$ is not too big, then this choice does not hurt the performance too much. Second, the upper bound $\|\theta\|_U$ is often more interpretable and easier to compute than $\|\theta\|$.*

**Lemma B.13** (Properties of $\hat{\theta}_n$). *For any $\delta \in (0, 1)$, let $U_\lambda = \frac{1}{2\lambda}\left(\epsilon_* + \sigma^2 + B^2\sqrt{\frac{2\log(2/\delta)}{n}}\right) + M_*$. With probability at least $1 - \delta$, we have*

$$\hat{L}_{A,B}(\hat{\theta}_n) \leqslant \frac{1}{2}(\epsilon_* + \sigma^2) + \frac{1}{2}B^2\sqrt{\frac{2\log(2/\delta)}{n}} + \lambda M_*$$
$$\|\hat{\theta}_n\|_U \leqslant U_\lambda. \tag{27}$$

*Proof.* Noting that $\theta^*$ is independent of the training data, we have for any $\delta \in (0,1)$, it holds *w.p.* at least $1 - \delta$ that

$$\hat{L}_{A,B}(\theta^*) \overset{(a)}{\leqslant} L_{A,B}(\theta^*) + \frac{1}{2}B^2\sqrt{\frac{2\log(2/\delta)}{n}} \leqslant L_A(\theta^*) + \frac{1}{2}B^2\sqrt{\frac{2\log(2/\delta)}{n}}$$

$$\overset{(b)}{\leqslant} \frac{1}{2}(\epsilon_* + \sigma^2) + \frac{1}{2}B^2\sqrt{\frac{2\log(2/\delta)}{n}},$$

where $(a)$ and $(b)$ follow from the Hoeffding's inequality (Theorem B.1) and the approximation assumption (25), respectively.

Since $\hat{\theta}_n$ is one solution of the minimization problem, we have

$$\hat{L}_{A,B}(\hat{\theta}_n) + \lambda\left\|\hat{\theta}_n\right\|_U \leqslant \hat{L}_{A,B}(\theta^*) + \lambda\|\theta^*\|_U$$

$$\leqslant \frac{1}{2}(\epsilon_* + \sigma^2) + \frac{1}{2}B^2\sqrt{\frac{2\log(2/\delta)}{n}} + \lambda M_*.$$

Then, the conclusions follow trivially from the above inequality. $\qquad\square$

**Proposition B.14** (Excess risk). *Denote by $\hat{\theta}_n$ the estimator defined in* (26). *Suppose Assumption 2 holds and $B > 2A$. For any $\delta \in (0, 1/2)$, we have w.p. at least $1 - 2\delta$ that*

$$\left\|\pi_A \circ h_{\hat{\theta}_n} - h^*\right\|^2_{L^2(P)} - \epsilon_* \lesssim \frac{\sigma^3 B}{(B-2A)^2}e^{-\frac{(B-2A)^2}{2\sigma^2}} + \lambda M_*$$

$$+ B^2\sqrt{\frac{p_{\mathcal{H}}\log(B\gamma(U_\lambda/\alpha_{\mathcal{H}}) + 3)}{n}} + B^2\sqrt{\frac{\log(4/\delta)}{n}}.$$

**Remark B.15.** *If $\gamma(J) \sim J^{\beta_{\mathcal{H}}}$, then by taking $B = C(2A + \sigma\sqrt{\log n})$ with $C \geqslant 1$ and $\lambda \sim 1/\sqrt{n}$, we have*

$$\left\|\pi_A \circ h_{\hat{\theta}_n} - h^*\right\|^2_{L^2(P)} - \epsilon_* \lesssim \frac{M_* + \mathrm{poly}(\sigma, A, \log n, \log(4/\delta), \log M_*, \log(1/\alpha_{\mathcal{H}}))\sqrt{p_{\mathcal{H}}\beta_{\mathcal{H}}}}{\sqrt{n}}.$$
(28)

*Thus, we show how to set the hyperparameters $B, \lambda$ to yield a standard $\widetilde{\mathcal{O}}(1/\sqrt{n})$ rate. Note that on the right hand side of* (28), *$\alpha_{\mathcal{H}}$, $\beta_{\mathcal{H}}$ and $p_{\mathcal{H}}$ depend on the model $h_\theta$ and we will specify them for CNNs and LCNs separately in this paper. However, it should be stressed that Proposition B.14 is generally applicable, which might be of independent interest.*

*Proof.* We will prove this theorem by utilizing the following decomposition:

$$\begin{aligned}L_A(\hat{\theta}_n) = {} & L_A(\hat{\theta}_n) - L_{A,B}(\hat{\theta}_n) && \textbf{(Step 1)}\\ & + L_{A,B}(\hat{\theta}_n) - \hat{L}_{A,B}(\hat{\theta}_n) && \textbf{(Step 2)}\\ & + \hat{L}_{A,B}(\hat{\theta}_n) && \textbf{(Step 3)}.\end{aligned}$$

**Step 1.** We first need the following tail bound of $\mathcal{N}(0, \sigma^2)$ (Vershynin, 2018, Proposition 2.1.2): For $X \sim \mathcal{N}(0, \sigma^2)$, it holds that

$$\mathbb{P}\{X \geqslant t\} \leqslant \frac{\sigma}{t} \cdot \frac{1}{\sqrt{2\pi}}e^{-t^2/(2\sigma^2)} \qquad \forall t > 0.$$
(29)

For any $\theta$, denote $Z(\mathbf{x}) = \pi_A \circ h_\theta(\mathbf{x}) - h^*(\mathbf{x}) + \xi$ where $\xi \sim \mathcal{N}(0, \sigma^2)$. Then, we have

$$
\begin{aligned}
|L_A(\theta) - L_{A,B}(\theta)| &= \frac{1}{2}\mathbb{E}_Z\left[\left(Z^2 - B^2\right)\mathbf{1}_{|Z|\geqslant B}\right] \\
&= \frac{1}{2}\int_0^{+\infty} \mathbb{P}\left(Z^2 - B^2 \geqslant t\right)\mathrm{d}t \\
&\leqslant \frac{1}{2}\int_0^{+\infty} \mathbb{P}\left(|\xi| \geqslant \sqrt{B^2 + t} - 2A\right)\mathrm{d}t \\
&\stackrel{(a)}{\leqslant} \frac{1}{\sqrt{2\pi}}\int_0^{+\infty} \frac{\sigma}{\sqrt{B^2 + t} - 2A}e^{-\frac{(\sqrt{B^2+t}-2A)^2}{2\sigma^2}}\,\mathrm{d}t \\
&= \int_B^{+\infty} \frac{2\sigma}{\sqrt{2\pi}}\frac{s}{s - 2A}e^{-\frac{(s-2A)^2}{2\sigma^2}}\,\mathrm{d}s \qquad (s = \sqrt{B^2 + t}) \\
&\leqslant \frac{2\sigma^2 B}{B - 2A}\int_B^{+\infty} \frac{1}{\sigma\sqrt{2\pi}}e^{-\frac{(s-2A)^2}{2\sigma^2}}\,\mathrm{d}s \\
&= \frac{2\sigma^2 B}{B - 2A}\mathbb{P}\left(\xi \geqslant B - 2A\right) \\
&\stackrel{(b)}{\leqslant} \frac{2\sigma^3 B}{(B - 2A)^2\sqrt{2\pi}}e^{-\frac{(B-2A)^2}{2\sigma^2}},
\end{aligned}
$$

where $(a)$ and $(b)$ follow from (29). By taking $\theta = \hat{\theta}_n$, we have

$$
L_A(\hat{\theta}_n) - L_{A,B}(\hat{\theta}_n) \leqslant \frac{2\sigma^3 B}{(B - 2A)^2\sqrt{2\pi}}e^{-\frac{(B-2A)^2}{2\sigma^2}}. \tag{30}
$$

**Step 2.** Define $\mathcal{F}_J = \{(x, y) \mapsto \ell_B(\pi_A \circ h_\theta(x), y) : \|\theta\| \leqslant J\}$. Due to

$$
\begin{aligned}
|\ell_B(\pi_A \circ h_\theta(\mathbf{x}), y) - \ell_B(\pi_A \circ h_{\theta'}(\mathbf{x}), y)| &\leqslant B\,|\pi_A \circ h_\theta(\mathbf{x}) - \pi_A \circ h_{\theta'}(\mathbf{x})| \\
&\leqslant B\,|h_\theta(\mathbf{x}) - h_{\theta'}(\mathbf{x})|,
\end{aligned}
$$

we have $\mathcal{N}(\mathcal{F}_J, \hat{\rho}_n, t) \leqslant \mathcal{N}(\mathcal{H}_J, \hat{\rho}_n, t/B) \leqslant (3B\hat{\gamma}_n(J)/t)^{p_\mathcal{H}}$ for $t \leqslant B\hat{\gamma}_n(J)$, where the last inequality comes from Assumption 2. For any $J > 0$, by Corollary B.11 and noting $\gamma(J) = \mathbb{E}[\hat{\gamma}_n(J)]$, It is straightforward that the following holds *w.p.* $1 - \delta$

$$
\sup_{\|\theta\|_\mathcal{P}\leqslant J} \left|L_{A,B}(\theta) - \hat{L}_{A,B}(\theta)\right| \lesssim B^2\sqrt{\frac{p_\mathcal{H}\log(B\gamma(J) + 3)}{n}} + B^2\sqrt{\frac{\log(4/\delta)}{n}}.
$$

By Lemma B.13, it holds *w.p.* $1 - \delta$ that $\|\hat{\theta}_n\|_U \leqslant U_\lambda$, which leads to $\|\hat{\theta}_n\| \leqslant U_\lambda/\alpha_\mathcal{H}$. Plugging it into the above bound gives: *w.p.* $1 - 2\delta$ it holds that

$$
L_{A,B}(\hat{\theta}_n) - \hat{L}_{A,B}(\hat{\theta}_n) \lesssim B^2\sqrt{\frac{\log(4/\delta)}{n}} + B^2\sqrt{\frac{p_\mathcal{H}\log(B\gamma(U_\lambda/\alpha_\mathcal{H}) + 3)}{n}}. \tag{31}
$$

**Step 3.** By Lemma B.13,

$$
\hat{L}_{A,B}(\hat{\theta}_n) \leqslant \frac{1}{2}(\epsilon_* + \sigma^2) + \frac{1}{2}B^2\sqrt{\frac{2\log(2/\delta)}{n}} + \lambda M_*. \tag{32}
$$

Combining (30), (31) and (32) gives

$$
L_{A,B}(\hat{\theta}_n) - \frac{1}{2}(\epsilon_* + \sigma^2) \lesssim B^2\sqrt{\frac{\log(4/\delta)}{n}} + B^2\sqrt{\frac{p_\mathcal{H}\log(B\gamma(U_\lambda/\alpha_\mathcal{H}) + 3)}{n}} + \lambda M_*. \tag{33}
$$

Then we complete the proof by noting $L_{A,B}(\hat{\theta}_n) = \frac{1}{2}\left\|\pi_A \circ h_{\hat{\theta}_n} - h^*\right\|^2_{L^2(P)} + \frac{1}{2}\sigma^2$. $\qquad\square$

## B.3 Fano's Method for Random Estimators

In this paper, the lower bounds for LCNs and FCNs are established using Fano's method from minimax theory (Wainwright, 2019, Section 15). However, different from the traditional statistical setup where the estimator is deterministic, the estimators produced by stochastic optimizers are random. To address this issue, we develop a variant of Fano's method for random estimators, which might be of independent interest.

We first recall the definition of Kullback–Leibler (KL) divergence.

**Definition B.16** (KL Divergence). Given two distributions $Q$ and $P$, the KL divergence between them is given by

$$D_{\mathrm{KL}}(Q\|P) := \begin{cases} \mathbb{E}_Q\left[\log \frac{\mathrm{d}Q}{\mathrm{d}P}\right] & \text{when } Q \text{ is absolutely continuous with respect to } P, \\ +\infty & \text{otherwise.} \end{cases}$$

If the distributions have densities with respect to some underlying measure $\nu$: $\mathrm{d}Q = q\,\mathrm{d}\nu, \mathrm{d}P = p\,\mathrm{d}\nu$, then then the KL divergence can be written in the form

$$D_{\mathrm{KL}}(Q\|P) = \int q(x) \log \frac{q(x)}{p(x)}\,\mathrm{d}v(x).$$

In particular, we will use the fact: for $P = \mathcal{N}(\mu_1, \sigma^2)$ and $Q = \mathcal{N}(\mu_2, \sigma^2)$, we have

$$D_{\mathrm{KL}}(P\|Q) = \frac{|\mu_1 - \mu_2|^2}{2\sigma^2} \tag{34}$$

and the following lemma.

**Lemma B.17.** *Denote by $Q_n, P_n$ the $n$-fold product distribution of $Q$ and $P$, respectively. When $Q$ is absolutely continuous with respect to $P$, $D_{\mathrm{KL}}(Q_n\|P_n) = nD_{\mathrm{KL}}(Q\|P)$*

*Proof.* This follows trivially from the fact that $\frac{\mathrm{d}Q_n}{\mathrm{d}P_n} = \frac{\mathrm{d}(Q\times\cdots\times Q)}{\mathrm{d}(P\times\cdots\times P)} = \left(\frac{\mathrm{d}Q}{\mathrm{d}P}\right)^n$. $\qquad\square$

**Minimax Risk.** Let $\{P_\theta : \theta \in \Theta\}$ be a set of probability distributions indexed by $\theta \in \Theta$ and supported on $\Omega$. Let $Z_\theta$ be a sample drawn from $P_\theta$. A random estimator $\mathbb{A} : \Omega \mapsto \mathcal{M}(\Theta)$ returns a random variable taking values in $\Theta$, which is a random estimate of $\theta$ by using the sample $Z_\theta$. Note that in this definition, the estimator is a random variable and it can model the one produced by stochastic optimizers such as SGD and Adam. In addition, when it is deterministic, this definition recovers the one commonly used in the statistical literature.

Let $d : \Theta \times \Theta \mapsto \mathbb{R}_{\geq 0}$ be a metric over $\Theta$. The minimax risk of learning distributions in $\{P_\theta : \theta \in \Theta\}$ is defined by

$$\inf_{\mathbb{A}\in\mathcal{A}} \sup_{\theta\in\Theta} \mathbb{E}_{\theta,\mathbb{A}}\left[d\left(\theta, \mathbb{A}(Z_\theta)\right)^2\right], \tag{35}$$

where $\mathbb{E}_{\theta,\mathbb{A}}$ means the expectation with respect to $Z_\theta$ and $\mathbb{A}$, and the outer infimum is taken over $\mathcal{A}$, i.e., the set of all the mappings from $\Omega$ to $\mathcal{M}(\Theta)$.

We next present a modification of Fano's method for random estimators. The proof process resembles that of the original Fano's method, but with some necessary modifications. This extension, although not difficult, can be very useful as random estimators have become ubiquitous nowadays.

**Theorem B.18** (Fano's method for random estimators). *Suppose that for $M \in \mathbb{N}$ and $A > 0$, there exists a packing $\theta_1, \ldots, \theta_M \in \Theta$ such that $d\left(\theta_i, \theta_j\right)^2 \geq 4A$ for any $i \neq j$, then*

$$\inf_{\mathbb{A}\in\mathcal{A}} \sup_{\theta\in\Theta} \mathbb{E}_{\theta,\mathbb{A}}\left[d\left(\theta, \mathbb{A}(Z_\theta)\right)^2\right] \geq A \cdot \left(1 - \frac{1}{M^2 \log M}\sum_{i,j}^{M} D_{\mathrm{KL}}\left(P_{\theta_i}\|P_{\theta_j}\right) - \frac{\log 2}{\log M}\right) \tag{36}$$

**Proof.** By Markov's inequality, $\mathbb{E}_\theta\left[\mathbb{E}_\mathbb{A}\left[d\left(\theta,\mathbb{A}(Z_\theta)\right)^2\right]\right] \geqslant A \cdot \mathbb{P}_\theta\left(\mathbb{E}_\mathbb{A}\left[d\left(\theta,\mathbb{A}(Z_\theta)\right)^2\right] > A\right)$, it is sufficient to lower bound

$$\inf_{\mathbb{A}\in\mathcal{A}}\sup_{\theta\in\Theta}\mathbb{P}_\theta\left(\mathbb{E}_\mathbb{A}\left[d\left(\theta,\mathbb{A}(Z_\theta)\right)^2\right] > A\right) \tag{37}$$

for some $A > 0$. This will be useful for techniques based on information theory.

The principle is simple: pack the index set $\Theta$ with balls of some radius $4A$, that is find $\theta_1,\ldots,\theta_M \in \Theta$ such that

$$\forall i \neq j,\ d\left(\theta_i,\theta_j\right)^2 \geqslant 4A,$$

and transform the estimation problem into a hypothesis test, that is, an algorithm going from the data $Z_\theta$ to one out of $M$ potential outcomes.

To this end, we take the supremum over a smaller set:

$$\sup_{\theta\in\Theta}\mathbb{P}_\theta\left(\mathbb{E}_\mathbb{A}\left[d\left(\theta,\mathbb{A}(Z_\theta)\right)^2\right] > A\right) \geqslant \max_{j\in[M]}\mathbb{P}_{\theta_j}\left(\mathbb{E}_\mathbb{A}\left[d\left(\theta_j,\mathbb{A}(Z_{\theta_j})\right)^2\right] > A\right) \tag{38}$$

Any algorithm $\mathbb{A}$ gives a test

$$g(\mathbb{A}(Z_\theta)) = \arg\min_{j\in[M]}\mathbb{E}_\mathbb{A}\left[d\left(\theta_j,\mathbb{A}(Z_\theta)\right)^2\right] \in [M], \tag{39}$$

where ties are broken arbitrarily (e.g., by selecting the minimal index).

If, for some $j \in [M], g(\mathbb{A}(Z_{\theta_j})) \neq j$, there exists $k \neq j$, such that $\mathbb{E}_\mathbb{A}\left[d\left(\theta_k,\mathbb{A}(Z_{\theta_j})\right)^2\right] \leqslant \mathbb{E}_\mathbb{A}\left[d\left(\theta_j,\mathbb{A}(Z_{\theta_j})\right)^2\right]$. Moreover, using the triangle inequality for $d(\cdot,\cdot)$, we get:

$$d\left(\theta_j,\theta_k\right)^2 \leqslant 2\,\mathbb{E}_\mathbb{A}\left[d\left(\theta_j,\mathbb{A}(Z_{\theta_j})\right)^2 + d\left(\mathbb{A}(Z_{\theta_j}),\theta_k\right)^2\right], \tag{40}$$

then,

$$\begin{aligned}
\mathbb{E}_\mathbb{A}\left[d\left(\theta_j,\mathbb{A}(Z_{\theta_j})\right)^2\right] &\geqslant \frac{1}{2}d\left(\theta_j,\theta_k\right)^2 - \mathbb{E}_\mathbb{A}\left[d\left(\mathbb{A}(Z_{\theta_j}),\theta_k\right)^2\right] \\
&\geqslant \frac{1}{2}d\left(\theta_j,\theta_k\right)^2 - \mathbb{E}_\mathbb{A}\left[d\left(\mathbb{A}(Z_{\theta_j}),\theta_j\right)^2\right] \quad \text{(using the optimal } k\text{),}
\end{aligned} \tag{41}$$

which implies $\mathbb{E}_\mathbb{A}\left[d\left(\theta_j,\mathbb{A}(Z_{\theta_j})\right)^2\right] \geqslant \frac{1}{4}d\left(\theta_j,\theta_k\right)^2 \geqslant A$. Thus, we have

$$\mathbb{P}_{\theta_j}\left(\mathbb{E}_\mathbb{A}\left[d\left(\theta_j,\mathbb{A}(Z_{\theta_j})\right)^2\right] > A\right) \geqslant \mathbb{P}_{\theta_j}(g(\mathbb{A}(Z_{\theta_j})) \neq j), \tag{42}$$

leading to

$$\begin{aligned}
\inf_{\mathbb{A}\in\mathcal{A}}\sup_{\theta\in\Theta}\mathbb{E}_{\theta,\mathbb{A}}\left[d\left(\theta,\mathbb{A}(Z_\theta)\right)^2\right] &\geqslant A \cdot \inf_g \max_{j\in\{1,\ldots,M\}}\mathbb{P}_{\theta_j}(g(Z_{\theta_j}) \neq j) \\
&\geqslant A \cdot \inf_g \frac{1}{M}\sum_{j=1}^M \mathbb{P}_{\theta_j}(g(Z_{\theta_j}) \neq j),
\end{aligned} \tag{43}$$

which can be further lower bounded using the following lemma ([Bach, 2023](), Corollary 12.1).

**Lemma B.19** (Fano's inequality for multiple hypothesis testing)**.** *Given $M$ distributions $P_1, P_2, \ldots, P_M$ over $\Omega$, let $Z_{\theta_j} \sim P_{\theta_j}$ for $j = 1, 2, \ldots, M$. Then*

$$\inf_g \frac{1}{M}\sum_{j=1}^M \mathbb{P}_{\theta_j}(g(Z_{\theta_j}) \neq j) \geqslant 1 - \frac{1}{M^2\log M}\sum_{j,j'=1}^M D_{\mathrm{KL}}\left(P_{\theta_j}\|P_{\theta_{j'}}\right) - \frac{\log 2}{\log M}, \tag{44}$$

*where the infimum is taken over all testing methods, i.e., all maps from $\Omega$ to $[M]$.*

Combining Lemma B.19 and equation (43), we complete the proof. $\quad\square$

In this paper, we will mostly use a variant of the above Fano's inequality for regression problems. Recall that in regression, the data are generated by $y = h(x) + \xi$ with $x \sim P$ and $\xi \sim \mathcal{N}(0,\sigma^2)$. Let $Q_h$ be the joint distribution of $(\mathbf{x}, y)$ and $S_n = \{(\mathbf{x}_i, y_i)\}_{i=1}^n$ be $n$ i.i.d. samples drawn from $Q_h$. Denote by $Q_{n,h}$ the $n$-fold product distribution of $Q_h$. Let $\mathcal{F}$ be the class of target functions. In such a case, the set of probability distributions is indexed by $h \in \mathcal{F}$:

$$\{Q_{n,h} : h \in \mathcal{F}\}.$$

**Proposition B.20** (Fano's method for regression). *Let $\mathcal{F}$ be a target function class. If there exists a packing $h_1, \ldots, h_M \in \mathcal{F}$ such that for any $i \neq j$, $\|h_i - h_j\|_{L^2(P)}^2 \geqslant 4A$, we have*

$$\inf_{\mathbb{A} \in \mathcal{A}} \sup_{h^* \in \mathcal{F}} \mathbb{E}[\|h_{\mathbb{A}(S_n)} - h^*\|_{L^2(P)}^2] \geqslant A \cdot \left( 1 - \frac{n}{2\sigma^2 M^2 \log M} \sum_{j,j'=1}^{M} \|h_j - h_{j'}\|_{L^2(P)}^2 - \frac{\log 2}{\log M} \right),$$
(45)

*where the expectation is taken over both the sampling of $S_n$ and randomness of $\mathbb{A}$.*

*Proof.* Define $d(h, h')^2 = \|h - h'\|_{L^2(P)}^2$. WLOG, assume that $P$ has a density function $p$ and denote by $\phi_{h,x}$ the density function of $\mathcal{N}(h(x), \sigma^2)$. Then, the density function of $Q_h$ is $p(x)\phi_{h,x}(y)$. Then, we have

$$D_{\mathrm{KL}}(Q_{n,h} \| Q_{n,h'}) = n D_{\mathrm{KL}}(Q_h \| Q_{h'})$$

$$= n \int p(x)\phi_{x,h}(y) \log\left( \frac{p(x)\phi_{x,h}(y)}{p(x)\phi_{x,h'}(y)} \right) \mathrm{d}x \, \mathrm{d}y$$

$$= n \int p(x) \left( \int \phi_{x,h}(y) \log\left( \frac{\phi_{x,h}(y)}{\phi_{x,h'}(y)} \right) \mathrm{d}y \right) \mathrm{d}x$$

$$= \frac{n}{2\sigma^2} \int |h(x) - h'(x)|^2 p(x) \, \mathrm{d}x = \frac{n}{2\sigma^2} \|h - h'\|_{L^2(P)}^2,$$

where the first equality follows from Lemma B.17 and the third equality is due to equation (34). We remark that this equality is also right when $P$ does not admit a density. In this case, we can construct and verify the Radon-Nikodym derivative, and then compute the KL divergence by the definition. Lastly, by plugging the above estimate into (36) in Theorem B.18, we complete the proof. $\qquad\square$

**Remark B.21.** *When we use Fano's method (Proposition B.20) to lower-bound the minimax risk, the most important step is to construct a "proper" packing. Take a look at the lower bound in (45). For fixed $M \in \mathbb{N}$ and $A > 0$, a proper packing should ensure $\sum_{i,j=1}^{M} \|h_i - h_j\|_{L^2(P)}^2$ to be as small as possible; otherwise the packing may yield a suboptimal lower bound. In fact, in most applications, we will simply attempt to bound the worst-case resolution: $\sup_{i,j} \|h_i - h_j\|_{L^2(P)}^2$.*

## C   Capacity Control for CNNs and LCNs

We first define a norm on the parameter space for controlling the model capacity effectively. Let $\theta = \{\theta_1, \theta_2, \ldots, \theta_L, W_o\}$ denote all the parameters of our network, where $\theta_l \in \Theta_l$ and $W_o \in \Theta_o$ denote the parameters used to parameterize $\mathcal{T}_l$ and $\mathcal{M}_o$, respectively.

Denote by $V_l$ the space of $l$-th layer hidden state. Let $\widetilde{V}_l \in \mathbb{R}^{p_l}$ be the vectorized version of $V_l$ armed with standard $\ell_2$ norm and $\widetilde{\mathcal{T}}_l : \widetilde{V}_{l-1} \mapsto \widetilde{V}_l$ be the corresponding representation of $\mathcal{T}_l$. Since $\widetilde{\mathcal{T}}_l$ is linear, there must exist unique $A : \Theta_l \mapsto \mathbb{R}^{p_l \times p_{l-1}}$ and $B : \Theta_l \mapsto \mathbb{R}^{p_l}$ such that $\widetilde{\mathcal{T}}_l \operatorname{vec}(\mathbf{z}^{(l)}) = A(\theta_l) \operatorname{vec}(\mathbf{z}^{(l)}) + B(\theta_l)$. Then we define

$$\|\theta_l\| = \|\widetilde{\mathcal{T}}_l\| := \|A(\theta_l)\|_2 + \|B(\theta_l)\|_2,$$
(46)

to control the "magnitude" of $\widetilde{\mathcal{T}}_l$, which ensures that for any $\widetilde{\mathbf{z}} \in \widetilde{V}_{l-1}$ we have $\|\widetilde{\mathcal{T}}_l \widetilde{\mathbf{z}}\|_2 \leqslant \|\widetilde{\mathcal{T}}_l\|(\|\widetilde{\mathbf{z}}\|_2 + 1)$. Furthermore, we consider a new norm over the whole parameter space $\Theta$ by

$$\|\theta\| := \|W_o\|_2 + \sum_{l=1}^{L} \|\theta_l\|.$$
(47)

It is equivalent to say $\|\theta\| = \|\mathcal{M}_o\| + \sum_{l=1}^{L} \|\widetilde{\mathcal{T}}_l\|$, which is the sum of operator norms of all linear transforms.

We remark that we will show the norm (47) provides tight control of covering number of CNNs and LCNs, but is less informative and interpretable. Thus, we introduce the $\|\theta\|_{\mathcal{P}}$ norm in (3) for CNNs and LCNs, which is an upper bound of $\|\theta\|$ as demonstrated below. As opposed to $\|\theta\|$, $\|\theta\|_{\mathcal{P}}$ is much more explicit, interpretable, and easier to compute. In the following, for both CNNs and LCNs, we will first derive the matrix form of $\widetilde{\mathcal{T}}_l$, thereby obtaining $A(\cdot)$ and $B(\cdot)$. Abusing notation, we will use $\mathcal{T}_l$ to denote $\widetilde{\mathcal{T}}_l$ for simplicity.

## C.1 CNNs

Recall that for $l \in [L]$, $\mathcal{T}_l : \mathbb{R}^{D_{l-1} \times C_{l-1}} \mapsto \mathbb{R}^{D_l \times C_l}$ is parameterized by a kernel $W^{(l)} \in \mathbb{R}^{C_l \times C_{l-1} \times s}$ and bias $\mathbf{b}^{(l)} \in \mathbb{R}^{C_l}$ as follows

$$(\mathcal{T}_l(\mathbf{z}))_{:,j} = \sum_{i=1}^{C_{l-1}} \mathbf{z}_{:,i} *_s W^{(l)}_{j,i,:} + b_j^{(l)} \mathbf{1}, \quad \text{for } j = 1, \dots, C_l. \tag{48}$$

We further point out that the number of parameters in our CNN is

$$N_{\text{cnn}} = C_L D_L + \sum_{l=0}^{L-1} (2C_l + 1) C_{l+1} \tag{49}$$

**A patch-based reformulation of CNNs.** For technical simplicity, it is easier to take a patch-based viewpoint of $\mathcal{T}_l$: $\mathcal{T}_l$ operates on each patch with an identical local linear transform. Note that $D_l = D_{l-1}/s$ and we can divide $\mathbf{z}^{(l-1)}$ into $D_l$ patchs of the shape $s \times C_{l-1}$:

$$\mathbf{z}^{(l-1)} = \begin{pmatrix} P_1^{(l-1)} \\ P_2^{(l-1)} \\ \vdots \\ P_{D_l}^{(l-1)} \end{pmatrix} \in \mathbb{R}^{D_{l-1} \times C_{l-1}}, \quad P_j^{(l-1)} = \mathbf{z}_{I_j,:}^{(l-1)} \in \mathbb{R}^{s \times C_{l-1}} \text{ for } j \in [D_l],$$

where $I_j := [(j-1)s + 1, js]$ denotes the $j$-th patch. Then $\mathcal{T}_l$ essentially makes the following transform

$$\mathcal{T}_l \mathbf{z}^{(l-1)} = \begin{pmatrix} \mathcal{A}^{(l)} P_1^{(l-1)} \\ \mathcal{A}^{(l)} P_2^{(l-1)} \\ \vdots \\ \mathcal{A}^{(l)} P_{D_l}^{(l-1)} \end{pmatrix},$$

where $\mathcal{A}^{(l)} : \mathbb{R}^{s \times C_{l-1}} \to \mathbb{R}^{1 \times C_l}$ is the local patch transform given by

$$(\mathcal{A}^{(l)} P)_i = \sum_{j=1}^{C_{l-1}} \sum_{k=1}^{s} W_{i,j,k} P_{j,k} + b_i^{(l)} \quad \text{for } i = 1, 2, \dots, C_l.$$

In a matrix form, $\mathcal{A}^{(l)}$ is given by

$$\mathcal{A}^{(l)} P = \widetilde{W}^{(l)} \operatorname{vec}(P) + \mathbf{b}^{(l)},$$

where

$$\widetilde{W}^{(l)} = \begin{pmatrix} \operatorname{vec}(W_{1,:,:})^\top \\ \operatorname{vec}(W_{2,:,:})^\top \\ \vdots \\ \operatorname{vec}(W_{C_l,:,:})^\top \end{pmatrix} \in \mathbb{R}^{C_l \times (sC_{l-1})}. \tag{50}$$

*The matrix form of $\mathcal{T}_l$.* Let

$$\widetilde{\mathbf{z}}^{(l)} = (\operatorname{vec}(P_1^{(l)})^\top, \operatorname{vec}(P_2^{(l)})^\top, \dots, \operatorname{vec}(P_{D_l}^{(l)})^\top)^\top \in \mathbb{R}^{D_l C_l}.$$

Abusing the notation, we still use $\mathcal{T}_l : \mathbb{R}^{D_{l-1} C_{l-1}} \to \mathbb{R}^{D_l C_l}$ to denote the matrix form of the one defined in (48). Then, we have

$$\mathcal{T}_l \widetilde{\mathbf{z}}^{(l-1)} = K^{(l)} \widetilde{\mathbf{z}}^{(l-1)} + \mathbf{s}^{(l)} \tag{51}$$

where

$$K^{(l)} = \begin{pmatrix} \widetilde{W}^{(l)} & 0 & \cdots & 0 \\ 0 & \widetilde{W}^{(l)} & \cdots & 0 \\ \vdots & \vdots & \ddots & \vdots \\ 0 & 0 & \cdots & \widetilde{W}^{(l)} \end{pmatrix} \in \mathbb{R}^{D_l C_l \times D_{l-1} C_{l-1}}, \quad \mathbf{s}^{(l)} = \begin{pmatrix} \mathbf{b}^{(l)} \\ \mathbf{b}^{(l)} \\ \vdots \\ \mathbf{b}^{(l)} \end{pmatrix} \in \mathbb{R}^{D_l C_l},$$

where the block matrix has $D_l$ blocks and $\widetilde{W}^{(l)}$ is given by (50).

**The parameter norm.** By (51), it is easy to see that

$$\|\mathcal{T}_l\| \leqslant \left\|K^{(l)}\right\|_2 + \left\|\mathbf{s}^{(l)}\right\|_2 = \left\|\widetilde{W}^{(l)}\right\|_2 + \sqrt{\frac{d}{2^{l-2}}} \left\|\mathbf{b}^{(l)}\right\|_2 \leqslant \left\|W^{(l)}\right\|_F + \sqrt{D_l} \left\|\mathbf{b}^{(l)}\right\|_2 \quad (52)$$

Then, we have

$$\|\theta\| = \|W_o\|_2 + \sum_{l=1}^{L} \|\mathcal{T}_l\| \leqslant \|W_o\|_2 + \sum_{l=1}^{L} (\|W^{(l)}\|_F + \sqrt{D_l}\|\mathbf{b}^{(l)}\|_2) = \|\theta\|_{\mathcal{P}}. \quad (53)$$

**Lemma C.1.** *For any $\theta, \theta'$ satisfy that $\|\theta\| \leqslant J$ and $\|\theta'\| \leqslant J$ and any $\mathbf{x} \in \mathcal{X}$, assume $\sigma_l$ satisfies $\sigma_l(0) = 0$ and locally Lipschitz condition $\|\sigma_l(\mathbf{y}) - \sigma_l(\mathbf{z})\|_2 \leqslant Q_l(\mathbf{x}) \|\mathbf{y} - \mathbf{z}\|_2$ for any $\mathbf{y}, \mathbf{z} \in \{\mathcal{T}_l \circ \cdots \circ \sigma_1 \circ \mathcal{T}_1(\mathbf{x}) : \|\theta\| \leqslant J\}$. We further denote $\bar{Q}_\sigma(\mathbf{x}) = \prod_{l=1}^{L}(Q_l(\mathbf{x}) + 1)$. Then, we have*

$$|h_\theta(\mathbf{x}) - h_{\theta'}(\mathbf{x})| \leqslant \bar{Q}_\sigma(\mathbf{x})(\|\mathbf{x}\|_2 + 1)(1 + J)^L \|\theta - \theta'\| \quad (54)$$

*Proof.* First assume that $\theta, \theta'$ only differ in the $l$-th layer, $l = 1, \ldots, L$.

$$|h_\theta(\mathbf{x}) - h_{\theta'}(\mathbf{x})| = \left| \mathcal{M}_o \circ \cdots \circ \sigma_l \circ \mathcal{T}_l \circ \cdots \circ \sigma_1 \circ \mathcal{T}_1(\mathbf{x}) - \mathcal{M}_o \circ \cdots \circ \sigma_l \circ \widetilde{\mathcal{T}}_l \circ \cdots \circ \sigma_1 \circ \mathcal{T}_1(\mathbf{x}) \right|$$

$$\leqslant \mathrm{Lip}(\mathcal{M}_o \circ \cdots \circ \sigma_l) \left\|\mathcal{T}_l - \widetilde{\mathcal{T}}_l\right\| (\|\sigma_{l-1} \circ \cdots \circ \sigma_1 \circ \mathcal{T}_1(\mathbf{x})\|_2 + 1)$$

$$\leqslant \mathrm{Lip}(\mathcal{M}_o \circ \cdots \circ \sigma_l) \left\|\mathcal{T}_l - \widetilde{\mathcal{T}}_l\right\| (\|\mathbf{x}\|_2 + 1)(1 + J)^{l-1} \left( \prod_{j=1}^{l-1}(Q_j(\mathbf{x}) + 1) \right)$$

$$\leqslant \left( \prod_{j=l}^{L}(Q_j(\mathbf{x}) + 1) \right) (1 + J)^{L-l+1} \left\|\mathcal{T}_l - \widetilde{\mathcal{T}}_l\right\| (\|\mathbf{x}\|_2 + 1)(1 + J)^{l-1} \left( \prod_{j=1}^{l-1}(Q_j(\mathbf{x}) + 1) \right)$$

$$= \bar{Q}_\sigma(\mathbf{x})(\|\mathbf{x}\|_2 + 1)(1 + J)^L \left\|\mathcal{T}_l - \widetilde{\mathcal{T}}_l\right\|$$

$$(55)$$

Second, assume $\theta, \theta'$ only differ in the final layer. We have

$$|h_\theta(\mathbf{x}) - h_{\theta'}(\mathbf{x})| = \left| \mathcal{M}_o \circ \cdots \circ \sigma_l \circ \mathcal{T}_l \circ \cdots \circ \sigma_1 \circ \mathcal{T}_1(\mathbf{x}) - \widetilde{\mathcal{M}}_o \circ \cdots \circ \sigma_l \circ \mathcal{T}_l \circ \cdots \circ \sigma_1 \circ \mathcal{T}_1(\mathbf{x}) \right|$$

$$\leqslant \left\|W_o - \widetilde{W}_o\right\|_2 \|(\sigma_L \circ \cdots \circ \sigma_l \circ \mathcal{T}_l \circ \cdots \circ \sigma_1 \circ \mathcal{T}_1(\mathbf{x}))\|_2$$

$$\leqslant \left\|W_o - \widetilde{W}_o\right\|_2 \bar{Q}_\sigma(\mathbf{x})(\|\mathbf{x}\|_2 + 1)(1 + J)^L$$

$$(56)$$

Lastly, we consider general $\theta, \theta'$ with $\|\theta\| \leqslant J$ and $\|\theta'\| \leqslant J$. We claim that one can find $\theta_1, \ldots, \theta_{L+1}$ such that

- $\|\theta_l\| \leqslant J$ for all $l \in [L + 1]$;
- $\theta' = \theta_{L+1}$, $\theta$ and $\theta_1$ only differ in one layer;
- for any $l$, $\theta_l$ and $\theta_{l+1}$ only differ in one layer.

It is easy to check that this can be done by first replacing the maximum norm layer in $\theta$ by the minimum norm layer in $\theta'$, and so on.

Then, by applying the telescoping sum, we complete the proof. $\square$

**Theorem C.2** (Covering number of CNN). *Let $h_\theta$ denote the CNN described in Section 2.1 and define $\mathcal{H}_J^{\mathrm{CNN}} = \{h_\theta : \|\theta\| \leqslant J\}$. Given $\mathbf{x}_1, \mathbf{x}_2, \ldots, \mathbf{x}_n \in \mathcal{X}$, denote $\hat{M}_n = \sqrt{\frac{1}{n} \sum_{i=1}^{n} \left(\bar{Q}_\sigma(\mathbf{x}_i)\right)^2 (\|\mathbf{x}_i\|_2 + 1)^2}$. Then, we have*

$$\mathcal{N}(\mathcal{H}_J^{\mathrm{CNN}}, \hat{\rho}_n, t) \leqslant \left( \frac{3\hat{M}_n J(1 + J)^L}{t} \right)^{N_{\mathrm{cnn}}}, \quad (57)$$

*given* $0 < t \leqslant \hat{M}_n J (1+J)^L$.

*Proof.* The conclusion follows trivially from Lemma C.1 and Lemma B.8. $\qquad \square$

## C.2 LCNs

Recall that for LCNs, the linear transform $\mathcal{T}_l : \mathbb{R}^{D_{l-1} \times C_{l-1}} \mapsto \mathbb{R}^{D_l \times C_l}$ is parameterized by $W^{(l)} \in \mathbb{R}^{C_l \times C_{l-1} \times D_{l-1}}$ and $\mathbf{b}^{(l)} \in \mathbb{R}^{D_l \times C_l}$ as follows

$$(\mathcal{T}_l(\mathbf{z}))_{:,j} = \sum_{i=1}^{C_{l-1}} \mathbf{z}_{:,i} \star_s W^{(l)}_{j,i,:} + \mathbf{b}^{(l)}_{:,j}, \quad \text{for } j = 1, \dots, C_l.$$

We further point out that the number of parameters in our LCN model is

$$N_{\text{lcn}} = C_L D_L + \sum_{l=0}^{L-1} (2C_l + 1) C_{l+1} D_{l+1} \tag{58}$$

**A patch-based reformulation of LCNs.** We reformulate deep LCNs in the same way as CNNs. Let $\widetilde{\mathbf{z}}^{(l)} \in \mathbb{R}^{D_l C_l}$ be the vectorized features; then the matrix form of $\mathcal{T}_l : \mathbb{R}^{D_{l-1} C_{l-1}} \rightarrow \mathbb{R}^{D_l C_l}$ is given by

$$\mathcal{T}_l \mathbf{z}^{(l-1)} = K^{(l)} \widetilde{\mathbf{z}}^{(l-1)} + \mathbf{s}^{(l)} \tag{59}$$

where $K^{(l)} = \text{diag}\{\widetilde{W}_1^{(l)}, \cdots, \widetilde{W}_{D_l}^{(l)}\}$ and $\widetilde{W}_i^{(l)} \in \mathbb{R}^{C_l \times (sC_{l-1})}$, where

$$\widetilde{W}^{(l)} = \begin{pmatrix} \text{vec}(W_{1,:,I_1})^\top \\ \text{vec}(W_{2,:,I_2})^\top \\ \vdots \\ \text{vec}(W_{C_l,:,I_{C_l}})^\top \end{pmatrix} \in \mathbb{R}^{C_l \times (sC_{l-1})}, \quad \mathbf{s}^{(l)} = \begin{pmatrix} \mathbf{b}_{1,:}^\top \\ \mathbf{b}_{2,:}^\top \\ \vdots \\ \mathbf{b}_{D_l,:}^\top \end{pmatrix} \in \mathbb{R}^{D_l C_l}, \tag{60}$$

where $I_j = [(j-1)s + 1, js]$.

**Parameter norm.** By (59), it is obvious that

$$\|\mathcal{T}_l\| \leqslant \|K^{(l)}\|_2 + \|\mathbf{s}^{(l)}\|_2 \leqslant \max_{j \in [D_l]} \|\widetilde{W}_j^{(l)}\| + \|\mathbf{b}^{(l)}\|_F$$

$$\leqslant \|W^{(l)}\|_F + \|\mathbf{b}^{(l)}\|_F.$$

Thus

$$\|\theta\| = \|W_o\| + \sum_{l=1}^{L} \|\mathcal{T}_l\| \leqslant \|W_o\| + \sum_{l=1}^{L} (\|W^{(l)}\|_F + \|\mathbf{b}^{(l)}\|_F) = \|\theta\|_{\mathcal{P}}. \tag{61}$$

In the following, we provide an upper bound of the covering number of deep LCNs, whose proof is nearly the same as that of deep CNNs.

**Lemma C.3.** *For any $\theta, \theta'$ satisfy that $\|\theta\| \leqslant J$ and $\|\theta'\| \leqslant J$ and any $\mathbf{x} \in \mathcal{X}$, assume $\sigma_l$ satisfies $\sigma_l(0) = 0$ and locally Lipschitz condition $\|\sigma_l(\mathbf{y}) - \sigma_l(\mathbf{z})\|_2 \leqslant Q_l(\mathbf{x}) \|\mathbf{y} - \mathbf{z}\|_2$ for any $\mathbf{y}, \mathbf{z} \in \{\mathcal{T}_l \circ \cdots \circ \sigma_1 \circ \mathcal{T}_1(\mathbf{x}) : \|\theta\| \leqslant J\}$. We further denote $\bar{Q}_\sigma(\mathbf{x}) = \prod_{l=1}^{L} (Q_l(\mathbf{x}) + 1)$. Then, we have*

$$|h_\theta(\mathbf{x}) - h_{\theta'}(\mathbf{x})| \leqslant \bar{Q}_\sigma(\mathbf{x})(\|\mathbf{x}\|_2 + 1)(1 + J)^L \|\theta - \theta'\| \tag{62}$$

The proof is exactly the same as that of Lemma C.1, i.e., the case of CNNs.

*Proof.* First assume that $\theta, \theta'$ only differ in the $l$-th layer, $l = 1, \ldots, L$.

$$
\begin{aligned}
|h_\theta(\mathbf{x}) - h_{\theta'}(\mathbf{x})| &= \left| \mathcal{M}_o \circ \cdots \circ \sigma_l \circ \mathcal{T}_l \circ \cdots \circ \sigma_1 \circ \mathcal{T}_1(\mathbf{x}) - \mathcal{M}_o \circ \cdots \circ \sigma_l \circ \widetilde{\mathcal{T}_l} \circ \cdots \circ \sigma_1 \circ \mathcal{T}_1(\mathbf{x}) \right| \\
&\leqslant \mathrm{Lip}(\mathcal{M}_o \circ \cdots \circ \sigma_l) \left\| \mathcal{T}_l - \widetilde{\mathcal{T}_l} \right\| \left( \left\| \sigma_{l-1} \circ \cdots \circ \sigma_1 \circ \mathcal{T}_1(\mathbf{x}) \right\|_2 + 1 \right) \\
&\leqslant \mathrm{Lip}(\mathcal{M}_o \circ \cdots \circ \sigma_l) \left\| \mathcal{T}_l - \widetilde{\mathcal{T}_l} \right\| (\|\mathbf{x}\|_2 + 1)(1 + J)^{l-1} \left( \prod_{j=1}^{l-1} (Q_j(\mathbf{x}) + 1) \right) \\
&\leqslant \left( \prod_{j=l}^{L} (Q_j(\mathbf{x}) + 1) \right) (1 + J)^{L-l+1} \left\| \mathcal{T}_l - \widetilde{\mathcal{T}_l} \right\| (\|\mathbf{x}\|_2 + 1)(1 + J)^{l-1} \left( \prod_{j=1}^{l-1} (Q_j(\mathbf{x}) + 1) \right) \\
&= \bar{Q}_\sigma(\mathbf{x})(\|\mathbf{x}\|_2 + 1)(1 + J)^L \left\| \mathcal{T}_l - \widetilde{\mathcal{T}_l} \right\|
\end{aligned}
\tag{63}
$$

Second, assume $\theta, \theta'$ only differ in the final layer. We have

$$
\begin{aligned}
|h_\theta(\mathbf{x}) - h_{\theta'}(\mathbf{x})| &= \left| \mathcal{M}_o \circ \cdots \circ \sigma_l \circ \mathcal{T}_l \circ \cdots \circ \sigma_1 \circ \mathcal{T}_1(\mathbf{x}) - \widetilde{\mathcal{M}_o} \circ \cdots \circ \sigma_l \circ \mathcal{T}_l \circ \cdots \circ \sigma_1 \circ \mathcal{T}_1(\mathbf{x}) \right| \\
&\leqslant \left\| W_o - \widetilde{W_o} \right\|_2 \left\| (\sigma_L \circ \cdots \circ \sigma_l \circ \mathcal{T}_l \circ \cdots \circ \sigma_1 \circ \mathcal{T}_1(\mathbf{x})) \right\|_2 \\
&\leqslant \left\| W_o - \widetilde{W_o} \right\|_2 \bar{Q}_\sigma(\mathbf{x})(\|\mathbf{x}\|_2 + 1)(1 + J)^L
\end{aligned}
\tag{64}
$$

Lastly, we consider general $\theta, \theta'$ with $\|\theta\| \leqslant J$ and $\|\theta'\| \leqslant J$. We claim that one can find $\theta_1, \ldots, \theta_{L+1}$ such that

- $\|\theta_l\| \leqslant J$ for all $l \in [L + 1]$;

- $\theta' = \theta_{L+1}$, $\theta$ and $\theta_1$ only differ in one layer;

- for any $l$, $\theta_l$ and $\theta_{l+1}$ only differ in one layer.

It is easy to check that this can be done by first replacing the maximum norm layer in $\theta$ by the minimum norm layer in $\theta'$, and so on.

Then, by applying the telescoping sum, we complete the proof. $\qquad \square$

**Theorem C.4** (Covering number of LCNs). *Let $h_\theta$ denote the LCN model described in Section 2.1 and define $\mathcal{H}_J^{\mathrm{LCN}} = \{h_\theta : \|\theta\| \leqslant J\}$. Given $\mathbf{x}_1, \mathbf{x}_2, \ldots, \mathbf{x}_n \in \mathcal{X}$, denote $\hat{M}_n = \sqrt{\frac{1}{n} \sum_{i=1}^{n} \left( \bar{Q}_\sigma(\mathbf{x}_i) \right)^2 (\|\mathbf{x}_i\|_2 + 1)^2}$. Then, we have*

$$
\mathcal{N}(\mathcal{H}_J^{\mathrm{LCN}}, \hat{\rho}_n, t) \leqslant \left( \frac{3\hat{M}_n J(1 + J)^L}{t} \right)^{N_{\mathrm{lcn}}},
\tag{65}
$$

*given $0 < t \leqslant \hat{M}_n J(1 + J)^L$.*

*Proof.* The conclusion follows trivially from Lemma C.3 and Lemma B.8. $\qquad \square$

**Remark C.5.** *Note that the number of parameters of a LCN is much larger that of its CNN counterpart due to the lack of weight sharing. As a result, the covering number of LCNs is accordingly much larger than that of CNNs.*

## C.3 Comparison with Long and Sedghi (2019).

We acknowledge that our proofs follow a similar approach developed in Long and Sedghi (2019) but we improve Long and Sedghi (2019) in various aspects.

- We provide an upper bound of the covering number of LCNs while Long and Sedghi (2019) only considered CNNs. Moreover, our bounds apply to activation functions that are not Lipschitz continuous such as the squared ReLU: $\sigma(z) = \max(0, z)^2$. Theis activation function has become popular recently in solving scientific computing problems (Li and Yu, 2019; E and Yu, 2018) and training large NLP pre-trained models (So et al., 2021).

- The bound in Long and Sedghi (2019) depends on the parameter norm $J$ exponentially. In contrast, our bound depends on $J$ only polynomially. This improvement is critical for obtaining a sharp generalization bound for the case where the parameter norm is relatively large. For instance, by applying Long and Sedghi (2019) to the setting of learning the separation task in Theorem 5.5, the corresponding sample complexity bound of LCNs will become $\widetilde{\mathcal{O}}(d^2)$. In contrast, our bound is $\widetilde{\mathcal{O}}(d)$. This improvement is very critical since the lower bound of learning with FCNs is $\Omega(d^2)$ and therefore, without this improvement, we are unable to establish the provable separation between LCNs and FCNs.

# D    Universal Approximation: Proofs in Section 3

## D.1    Proof of Theorem 3.1

We first recall a well-known universality result of two-layer ReLU networks.

**Lemma D.1.** *(Leshno et al., 1993) Let $\Omega$ be any compact set in $\mathbb{R}^d$ and $\sigma$ be the ReLU function. For any $h \in C(\Omega)$ and any $\epsilon > 0$, there exists a a two-layer neural network $f_m(\mathbf{x}; \theta') = \sum_{j=1}^m a_j \sigma(\mathbf{u}_j^\top \mathbf{x} + c_j)$ such that*

$$\sup_{\mathbf{x} \in \Omega} |f_m(\mathbf{x}; \theta') - h(\mathbf{x})| \leqslant \epsilon. \tag{66}$$

**Lemma D.2** (Feature extraction). *With the choice of architecture in Theorem 3.1, the exist parameters such that $\mathbf{z}^{(L-1)}(\mathbf{x}) \in \mathbb{R}^{2 \times (4d)}$ satisfies*

$$\mathbf{z}^{(L-1)}(\mathbf{x}) = \begin{pmatrix} \sigma(x_1) & \sigma(-x_1) & \sigma(x_2) & \sigma(-x_2) & \cdots & \sigma(x_{2d}) & \sigma(-x_{2d}) \\ \sigma(x_{2d+1}) & \sigma(-x_{2d+1}) & \sigma(x_{2d+2}) & \sigma(-x_{2d+2}) & \cdots & \sigma(x_{4d}) & \sigma(-x_{4d}) \end{pmatrix} \in \mathbb{R}^{2 \times 4d}.$$

**Remark D.3.** *This lemma shows that $\mathbf{z}^{(L-1)}(\mathbf{x})$ strictly stores spatial information into different channels. Note that $4d$ is the number of channels and $2$ is the spatial dimension. The stored information information is in the form of $\{\sigma(x_i), \sigma(-x_i)\}_{i=1}^{4d}$. Noticing $t = \sigma(t) - \sigma(-t)$, we can conclude that there is no information loss since for any $h : \mathbb{R}^{4d} \mapsto \mathbb{R}$, $h(x_1, x_2, \ldots, x_{4d})$ can be represented using the stored features as follows*

$$h(\sigma(x_1) - \sigma(-x_1), \sigma(x_2) - \sigma(-x_2), \ldots, \sigma(x_{4d}) - \sigma(-x_{4d})).$$

*It is worth noting that obviously, the weights and bias in $\mathcal{T}_1, \ldots, \mathcal{T}_{L-1}$ do not depend on the target function $h^*$.*

*Proof.* We provide a constructive proof to this lemma. First, we set all bias to be zero, i.e., $\mathbf{b}^{(l)} = \mathbf{0}$ for all $l \in [L-1]$. Next we state how to set the weights for different layers.

- When $l = 0$, $\mathbf{z}^{(0)} = \mathbf{x} = (x_1, x_2, \ldots, x_{4d})$.

- When $l = 1$, set $W^{(1)} \in \mathbb{R}^{1 \times 4 \times 2}$ as follows $W_{1,1,:}^{(1)} = (1, 0)^\top$, $W_{1,2,:}^{(1)} = (-1, 0)^\top$, $W_{1,3,:}^{(1)} = (0, 1)^\top$, $W_{1,4,:}^{(1)} = (0, -1)^\top$. Under this construction, it is easy to verify that

$$(\mathbf{z}^{(1)})^\top = \begin{pmatrix} \sigma(x_1) & \sigma(x_3) & \ldots & \sigma(x_{4d-1}) \\ \sigma(-x_1) & \sigma(-x_3) & \ldots & \sigma(-x_{4d-1}) \\ \sigma(x_2) & \sigma(x_4) & \ldots & \sigma(x_{4d}) \\ \sigma(-x_2) & \sigma(-x_4) & \ldots & \sigma(-x_{4d}) \end{pmatrix} \in \mathbb{R}^{4 \times 2d} \tag{67}$$

- Similarly, for $l = 2$, we can set $W^{(2)} \in \mathbb{R}^{4 \times 8 \times 2}$ such that

$$(\mathbf{z}^{(2)})^\top = \begin{pmatrix} \sigma(x_1) & \sigma(x_5) & \dots & \sigma(x_{4d-3}) \\ \sigma(-x_1) & \sigma(-x_5) & \dots & \sigma(-x_{4d-3}) \\ \sigma(x_2) & \sigma(x_6) & \dots & \sigma(x_{4d-2}) \\ \sigma(-x_2) & \sigma(-x_6) & \dots & \sigma(-x_{4d-2})) \\ \sigma(x_3) & \sigma(x_7) & \dots & \sigma(x_{4d-1})) \\ \sigma(-x_3) & \sigma(-x_7) & \dots & \sigma(-x_{4d-1})) \\ \sigma(x_4) & \sigma(x_8) & \dots & \sigma(x_{4d})) \\ \sigma(-x_4) & \sigma(-x_8) & \dots & \sigma(-x_{4d})) \end{pmatrix} \in \mathbb{R}^{8 \times d}$$

- By induction, when $l = L - 1$, we have

$$(\mathbf{z}^{(L-1)})^\top = \begin{pmatrix} \sigma(x_1) & \sigma(x_{2d+1}) \\ \sigma(-x_1) & \sigma(-x_{2d+1}) \\ \sigma(x_2) & \sigma(x_{2d+2}) \\ \sigma(-x_2) & \sigma(-x_{2d+2}) \\ \vdots & \vdots \\ \sigma(x_{2d-1}) & \sigma(x_{4d-1}) \\ \sigma(-x_{2d-1}) & \sigma(-x_{4d-1}) \\ \sigma(x_{2d}) & \sigma(x_{4d}) \\ \sigma(-x_{2d}) & \sigma(-x_{4d}) \end{pmatrix} \in \mathbb{R}^{4d \times 2}$$

Specifically, the above result is achieved by setting the weights of $l = 2, \dots, L - 1$ as follows [2]. For $1 \leqslant j \leqslant C_{l-1}$ and $j$ is odd, we set

$$W^{(l)}_{j,j,:} = -W^{(l)}_{j+1,j,:} = W^{(l)}_{j+1,j+1,:} = -W^{(l)}_{j,j+1,:} = (1,0)^\top$$
$$W^{(l)}_{j,j+C_{l-1},:} = -W^{(l)}_{j+1,j+C_{l-1},:} = W^{(l)}_{j+1,j+1+C_{l-1},:} = -W^{(l)}_{j,j+1+C_{l-1},:} = (0,1)^\top$$

Recall that $W^{(l)}_{i,j,:}$ represent the filter used to extract the information from the $i$-th input channel to the $j$-th output channel. Let $j' = (j+1)/2$ if $j$ is odd. To prove the above result, we only show that for any $l \in [L - 1]$,

$$\mathbf{z}^{(l)}_{:,j} = \begin{cases} (\sigma(x_{j'}), \sigma(x_{j'+2^l}), \dots, \sigma(x_{j'+(D_l-1)2^l})) & \text{if } j \text{ is odd} \\ (\sigma(-x_{j'}), \sigma(-x_{j'+2^l}), \dots, \sigma(-x_{j'+(D_l-1)2^l})) & \text{if } j \text{ is even.} \end{cases} \tag{68}$$

We verify this below by induction.

First, the case of $l = 1$ holds due to (68). Assume (68) holds for $1, \dots, l-1$, let us compute $(\mathcal{T}_l(\mathbf{z}^{(l-1)}))_{:,j}$. Without loss of generality, we only consider the case where $j$ is odd.

- When $j \leqslant C_{l-1}$,

$$(\mathcal{T}_l \mathbf{z}^{(l-1)})_{:,j} = \sum_{i=1}^{C_{l-1}} \mathbf{z}^{(l-1)}_{:,i} *_s W^{(l)}_{j,i,:} + b^{(l)}_j$$
$$= \mathbf{z}^{(l-1)}_{:,j} *_s W^{(l)}_{j,j,:} + \mathbf{z}^{(l-1)}_{:,j+1} *_s W^{(l)}_{j+1,j,:}$$

Hence,

$$(\mathcal{T}_l \mathbf{z}^{(l-1)})_{i,j} = \mathbf{z}^{(l-1)}_{2i-1,j} - \mathbf{z}^{(l-1)}_{2i-1,j+1}$$
$$= \sigma(x_{j'+(2i-2)2^{l-1}}) - \sigma(-x_{j'+(2i-2)2^{l-1}}) = x_{j'+(i-1)2^l}$$

- When $j \geqslant C_{l-1}$ we similarly have

$$(\mathcal{T}_l \mathbf{z}^{(l-1)})_{:,j} = \sum_{i=1}^{C_{l-1}} \mathbf{z}^{(l-1)}_{:,i} *_s W^{(l)}_{j,i,:} + b^{(l)}_j$$
$$= \mathbf{z}^{(l-1)}_{:,j-C_{l-1}} *_s W^{(l)}_{j-C_{l-1},j,:} + \mathbf{z}^{(l-1)}_{:,j+1-C_{l-1}} *_s W^{(l)}_{j+1-C_{l-1},j,:}$$

---

[2]The following verification is rigorous but not intuitive.

So

$$(\mathcal{T}_l \mathbf{z}^{(l-1)})_{i,j} = \mathbf{z}^{(l-1)}_{2i,j-C_{l-1}} - \mathbf{z}^{(l-1)}_{2i,j+1-C_{l-1}}$$
$$= \sigma(x_{j'-C_{l-1}/2+(2i-1)2^{l-1}}) - \sigma(-x_{j'-C_{l-1}/2+(2i-1)2^{l-1}}) = x_{j'+(i-1)2^l}$$

Thus, the case of $l+1$ also holds.

$\square$

**Proof of Theorem 3.1.** By setting the weights and biases of the first $L-1$ layers according to Lemma D.2, we have $h_\theta(\mathbf{x}) = \mathcal{M}_o \circ \sigma \circ \mathcal{T}_L(\mathbf{z}^{(L-1)}(\mathbf{x}))$, where

$$\mathbf{z}^{(L-1)}(\mathbf{x}) = \begin{pmatrix} \sigma(x_1) & \sigma(-x_1) & \sigma(x_2) & \sigma(-x_2) & \cdots & \sigma(x_{2d}) & \sigma(-x_{2d}) \\ \sigma(x_{2d+1}) & \sigma(-x_{2d+1}) & \sigma(x_{2d+2}) & \sigma(-x_{2d+2}) & \cdots & \sigma(x_{4d}) & \sigma(-x_{4d}) \end{pmatrix}$$

By Lemma D.1, for any $h^* \in C(\Omega)$, there exists a a two-layer neural network $f_m(\mathbf{x}; \theta') = \sum_{j=1}^m a_j \sigma(\mathbf{u}_j^\top \mathbf{x} + c_j)$ such that $\sup_{\mathbf{x} \in \Omega} |f_m(\mathbf{x}; \theta') - h^*(\mathbf{x})| \leqslant \epsilon$. Then our proof is completed by showing that we can construct appropriate $\mathcal{T}_L$ and $\mathcal{M}_o$ such that the deep CNN $h_\theta$ can simulate the two-layer neural network $f_m(\cdot; \theta')$.

1. set $C_L = m$, $b_j^{(L)} = c_j$. For odd $i$, let $i' = (i+1)/2$ and set $W_{i,j,:}^{(L)} = ((\mathbf{u}_j)_{i'}, (\mathbf{u}_j)_{i'+2d})$ and for even $i$ we set $W_{j,i,:}^{(L)} = -W_{j,i-1,:}^{(L)}$. Under this construction,

$$(\mathbf{z}^{(L)}(\mathbf{x}))_{1,j} = \sigma\left(\sum_{i=1}^{C_{L-1}} (\mathbf{z}^{(L-1)}(\mathbf{x}))_{:,i} *_s W_{i,j,:}^{(L)} + b_j^{(L)}\right)$$
$$= \sigma\left(\sum_{i=1}^{4d} (\mathbf{u}_j)_i(\sigma(x_i) - \sigma(-x_i)) + c_j\right)$$
$$= \sigma(\mathbf{u}_j^\top \mathbf{x} + c_j)$$

2. Set $W_o = (a_1, \ldots, a_m)$. Then, the CNN $h_\theta$ represents exactly the same function as $f_m(\cdot; \theta')$:

$$h_\theta(\mathbf{x}) = \mathcal{M}_o \circ \mathbf{z}^{(L)}(\mathbf{x}) = \sum_{j=1}^m a_j(\mathbf{z}^{(L)}(\mathbf{x}))_{1,j} = \sum_{j=1}^m a_j \sigma(\mathbf{u}_j^\top \mathbf{x} + c_j) = f_m(\mathbf{x}; \theta').$$

Thus, we complete the proof. $\square$

### D.2   Proof of Proposition 3.2

Given two functions $f, g$ over $\mathcal{X}$, let $\rho_\infty(f,g) := \sup_{\mathbf{x} \in \mathcal{X}} |f(\mathbf{x}) - g(\mathbf{x})|$.

We prove this theorem by contradiction. First, any CNN $h_\theta$ with the depth $L \leqslant \log_2(d) + 1$ can be represent as

$$h_\theta(\mathbf{x}) = g_1(x_1, \ldots, x_{2d}) + g_2(x_{2d+1}, \ldots, x_{4d}) \tag{69}$$

for some $g_1$ and $g_2$. This is because that the size of receptive field is no greater than $2d$. Obviously, $h_\theta$ cannot represent long-range functions like $h^*(\mathbf{x}) := x_1 x_{2d+1}$. Specifically, we have

$$\inf_\theta \rho_\infty(h_\theta, h^*) \geqslant \inf_{g_1, g_2} \sup_{\mathbf{x} \in \mathcal{X}} |g_1(x_1, \ldots, x_{2d}) + g_2(x_{2d+1}, \ldots, x_{4d}) - x_1 x_{2d+1}| \tag{70}$$

If there exist $g_1$ and $g_2$ such that

$$\sup_{\mathbf{x} \in \mathcal{X}} |g_1(x_1, \ldots, x_{2d}) + g_2(x_{2d+1}, \ldots, x_{4d}) - x_1 x_{2d+1}| \leqslant \frac{1}{8}, \tag{71}$$

then taking special $\mathbf{x}$'s gives

$$|g_1(0,0,\ldots,0) + g_2(0,0,\ldots,0)| \leqslant \frac{1}{8}$$

$$|g_1(0,0,\ldots,0) + g_2(1,0,\ldots,0)| \leqslant \frac{1}{8}$$

$$|g_1(1,0,\ldots,0) + g_2(0,0,\ldots,0)| \leqslant \frac{1}{8} \tag{72}$$

$$|g_1(1,0,\ldots,0) + g_2(1,0,\ldots,0) - 1| \leqslant \frac{1}{8}$$

From the first three inequalities, we have

$|g_1(1,0,\ldots,0) + g_2(1,0,\ldots,0)|$
$= |g_1(1,0,\ldots,0) + g_2(0,0,\ldots,0) - g_2(0,0,\ldots,0) - g_1(0,0,\ldots,0) + g_1(0,0,\ldots,0) + g_2(1,0,\ldots,0)|$
$\leqslant |g_1(1,0,\ldots,0) + g_2(0,0,\ldots,0)| + |g_2(0,0,\ldots,0) + g_1(0,0,\ldots,0)| + |g_1(0,0,\ldots,0) + g_2(1,0,\ldots,0)|$
$\leqslant \frac{3}{8},$

which is contradictory to the fourth inequality in (72). Thus (71) cannot hold and we complete the proof. $\square$

### D.3  Proof of Proposition 3.3

We first recall that $\rho_\infty(f,g) := \sup_{\mathbf{x}\in\mathcal{X}} |f(\mathbf{x}) - g(\mathbf{x})|$.

We prove this theorem by contradiction. First, without downsampling, any CNN with depth $L \leqslant 4d - 2$ can be represented as

$$h_\theta(\mathbf{x}) = g_1(x_1, x_2, \ldots, x_{4d-1}) + g_2(x_2, x_3, \ldots, x_{4d}), \tag{73}$$

for some $g_1, g_2$. In this form, $x_{4d}$ and $x_1$ do not have direct correlation and intuitively, it should be impossible to represent functions like $h^*(\mathbf{x}) := x_1 x_{4d}$ by using this CNN. This intuition can be made rigorously as follows. Note that

$$\inf_\theta \rho_\infty(h_\theta, h^*) \geqslant \inf_{g_1, g_2} \sup_{\mathbf{x}\in\mathcal{X}} |g_1(x_1, \ldots, x_{4d-1}) + g_2(x_2, \ldots, x_{4d}) - x_1 x_{4d}| \tag{74}$$

If there is $g_1, g_2$ such that

$$\sup_{\mathbf{x}\in\mathcal{X}} |g_1(x_1, x_2, \ldots, x_{4d-1}) + g_2(x_2, x_4, \ldots, x_{4d}) - x_1 x_{4d}| \leqslant \frac{1}{8}, \tag{75}$$

then

$$|g_1(0,0,\ldots,0) + g_2(0,0,\ldots,0)| \leqslant \frac{1}{8}$$

$$|g_1(0,0,\ldots,0) + g_2(0,0,\ldots,1)| \leqslant \frac{1}{8}$$

$$|g_1(1,0,\ldots,0) + g_2(0,0,\ldots,0)| \leqslant \frac{1}{8} \tag{76}$$

$$|g_1(1,0,\ldots,0) + g_2(0,0,\ldots,1) - 1| \leqslant \frac{1}{8}$$

From the first three inequalities, we have

$|g_1(1,0,\ldots,0) + g_2(0,0,\ldots,1)|$
$= |g_1(1,0,\ldots,0) + g_2(0,0,\ldots,0) - g_2(0,0,\ldots,0) - g_1(0,0,\ldots,0) + g_1(0,0,\ldots,0) + g_2(0,0,\ldots,1)|$
$\leqslant |g_1(1,0,\ldots,0) + g_2(0,0,\ldots,0)| + |g_2(0,0,\ldots,0) + g_1(0,0,\ldots,0)| + |g_1(0,0,\ldots,0) + g_2(0,0,\ldots,1)|$
$\leqslant \frac{3}{8},$

which is contradictory to the fourth inequalities in (76). Therefore, (75) cannot hold and we complete the proof. $\square$

# E  Learning Sparse Functions: Proofs of Section 4

## E.1  Proof of Lemma 4.4

Here we prove a complete version of Lemma 4.4.

**Lemma E.1** (Adaptive coordinate selection). *Given $k, m \in \mathbb{N}$, consider a* ReLU *CNN model with depth $L = \log_2(4d)$ and the channel numbers satisfying $C_l = 2k$ for all $l = 1, \ldots, L-1$ and $C_L = m$. Then for any $\mathcal{I} = (i_1, \ldots, i_k) \subset [d]$ with $1 \leqslant i_1 < \cdots < i_k \leqslant d$, $\mathbf{u}_1, \ldots, \mathbf{u}_m \in \mathbb{R}^k$, and $c_1, \ldots, c_m \in \mathbb{R}$, there exists $\theta \in \Theta$ such that the CNN $h_\theta$ outputs:*

$$\mathbf{z}^{(L)}(\mathbf{x}) = \left(\sigma(\mathbf{u}_1^\top \mathbf{x}_\mathcal{I} + c_1), \ldots, \sigma(\mathbf{u}_m^\top \mathbf{x}_\mathcal{I} + c_m)\right) \in \mathbb{R}^{1 \times m}.$$

*Furthermore, for this CNN, the number of parameters is $\mathcal{O}(k^2 \log d + km)$ and the parameter norm satisfies*

$$\|\theta\|_{\mathcal{P}} \lesssim \sqrt{k} \log d + \sqrt{\sum_{i=1}^m \|\mathbf{u}_i\|_2^2} + \sqrt{\sum_{i=1}^m c_i^2} + \|W_o\|_2$$

In this case, the proof is the same as that of linear CNNs (Lemma 4.2) since we only need to double the channels: every two channels form a group, storing $\{\sigma(x_i), \sigma(-x_i)\}_{i=1}^k$; different groups of channels proceed still independently like the case of linear CNNs. Specifically, for $l = 1, 2, \ldots, L-1$, we follow a similar idea to set weights and biases such that

$$\mathbf{z}^{(L-1)}(\mathbf{x}) = \begin{pmatrix} \sigma(x_{t_1}), \sigma(-x_{t_1}), \sigma(x_{t_2}), \sigma(-x_{t_2}), \ldots, \sigma(x_{t_k}), \sigma(-x_{t_k}) \\ \sigma(x_{s_1}), \sigma(-x_{s_1}), \sigma(x_{s_2}), \sigma(-x_{s_2}), \ldots, \sigma(x_{s_k}), \sigma(-x_{s_k}) \end{pmatrix} \in \mathbb{R}^{2 \times 2k},$$

where for any $j \in [k]$, either $t_j = i_j$ or $s_j = i_j$. Then, we set the weights and bias of $L$-th layer according to $\{\mathbf{u}_i\}_{i=1}^m$ and $\{c_i\}_{i=1}^m$ such that the outputs are $\{\sigma(\mathbf{u}_i^\top \mathbf{x}_\mathcal{I} + c_i)\}_{i=1}^m$. We refer to the following proof for details.

*Proof.* For any $j \in [k]$, we write the $i_j$'s binary representation as follows

$$i_j - 1 = \sum_{l=0}^{L-1} a_{j,l} 2^l \tag{77}$$

where $a_{j,l} \in \{0, 1\}$. Here, "$-1$" is used to ensure the $i_j \in [0, 2^L - 1]$. Next, we set the weights and bias according to the binary representation (77).

*Set weights and bias adaptively*:

- For $l = 1$, $W^{(1)} \in \mathbb{R}^{1 \times 2k \times 2}$. For $j \in [k]$, set

$$W_{2j-1,1,:}^{(1)} = -W_{2j,1,:}^{(1)} = (1 - a_{j,0}, a_{j,0}).$$

- For $l = 2, \ldots, L-1$, $W^{(l)} \in \mathbb{R}^{2k \times 2k \times 2}$. For $j \in [k]$, set

$$W_{2j-1,2j-1,:}^{(l)} = W_{2j,2j,:}^{(l)} = -W_{2j-1,2j,:}^{(l)} = -W_{2j,2j-1,:}^{(l)} = (1 - a_{j,l-1}, a_{j,l-1}).$$

- For any $l \in [L-1]$, set all entries in $W^{(l)}$ unmentioned above and $\mathbf{b}^{(l)}$ to zeros.

Note that $a_{j,l} \in \{0, 1\}$ for all $j \in [k]$ and $l \in [L]$. Thus, under the above construction, all nonzero filters are taken from $\{(1, 0), (-1, 0), (0, 1), (0, -1)\}$.

*The forward selection process.* Let us compute $\mathcal{T}_l \mathbf{z}^{(l-1)}$ for $l = 1, \ldots, L-1$. Let $s_{j,l} = 1 + \sum_{p=0}^{l-1} a_{j,p} 2^p$. We claim that for $l \in [L-1]$,

$$(\mathcal{T}_l \mathbf{z}^{(l-1)})_{:,j} = \begin{cases} (x_{s_{j',l}}, x_{2^l + s_{j',l}}, \cdots, x_{4d - 2^l + s_{j',l}}) & \text{if } j \text{ is odd}, \\ -(\mathcal{T}_l \mathbf{z}^{(l-1)})_{:,j-1} & \text{if } j \text{ is even} \end{cases}$$

We will verify this by induction.

For $l = 1$

$$(\mathcal{T}_1 \mathbf{z}^{(0)})_{:,j} = (\mathcal{T}_1(\mathbf{x}))_{:,j} = \mathbf{x} *_s W^{(1)}_{j,1,:}$$

So when $j$ is even $(\mathcal{T}_1 \mathbf{z}^{(0)})_{:,j} = -(\mathcal{T}_1 \mathbf{z}^{(0)})_{:,j-1}$ and when $j$ is odd we have

$$(\mathcal{T}_1 \mathbf{z}^{(0)})_{:,j} = (x_{1+a_{j',0}}, x_{3+a_{j',0}}, \cdots, x_{4d-1+a_{j',0}})$$

where $j' = (j+1)/2$ as above. $l = 1$ has been verified.

Then, assume that the case of $1, \cdots, l-1$ hold, we compute $(\mathcal{T}_l \mathbf{z}^{(l-1)})_{:,j}$. Without loss of generality, we only consider the case where $j$ is odd. Observing that

$$
\begin{aligned}
(\mathcal{T}_l \mathbf{z}^{(l-1)})_{:,j} &= \sum_{i=1}^{C_{l-1}} \mathbf{z}^{(l-1)}_{:,i} *_s W^{(l)}_{j,i,:} + b^{(l)}_j \\
&= \mathbf{z}^{(l-1)}_{:,j} *_s W^{(l)}_{j,j,:} + \mathbf{z}^{(l-1)}_{:,j+1} *_s W^{(l)}_{j,j+1,:}
\end{aligned}
,
$$

which implies

$$
\begin{aligned}
(\mathcal{T}_l \mathbf{z}^{(l-1)})_{i,j} &= (1 - a_{j',l-1})\sigma(x_{2^{l-1}(2i-2)+s_{j',l-1}}) + a_{j',l-1}\sigma(x_{2^{l-1}(2i-1)+s_{j',l-1}}) \\
&\quad - \left[ (1 - a_{j',l-1})\sigma(-x_{2^{l-1}(2i-2)+s_{j',l-1}}) + a_{j',l-1}\sigma(-x_{2^{l-1}(2i-1)+s_{j',l-1}}) \right]. \\
&= x_{2^l(i-1)+s_{j',l-1}+a_{j',l-1}2^{l-1}} \\
&= x_{(i-1)2^l+s_{j',l}}
\end{aligned}
$$

Thus, the claim is verified and after $L-1$ layer,

$$
\begin{aligned}
\mathbf{z}^{(L-1)}_{1,:} &= (\sigma(x_{s_{1,L-1}}), \sigma(-x_{s_{1,L-1}}), \cdots, \sigma(x_{s_{k,L-1}}), \sigma(-x_{s_{k,L-1}})) \\
\mathbf{z}^{(L-1)}_{2,:} &= (\sigma(x_{2^{L-1}+s_{1,L-1}}), \sigma(-x_{2^{L-1}+s_{1,L-1}}), \cdots, \sigma(x_{2^{L-1}+s_{k,L-1}}), \sigma(-x_{2^{L-1}+s_{k,L-1}}))
\end{aligned}
$$

For the $L$-th layer, for $j = 1, 2, \ldots, k$ set

$$
\begin{aligned}
W^{(L)}_{2i-1,j,:} &= -W^{(L)}_{2i,j,:} = (\mathbf{u}_j)_i \cdot (1 - a_{i,L-1}, a_{i,L-1}) \\
b^{(L)}_j &= c_j
\end{aligned}
$$

We compute

$$
\begin{aligned}
(\mathcal{T}_L \mathbf{z}^{(L-1)})_{:,j} &= \sum_{i=1}^{2k} \mathbf{z}^{(L-1)}_{:,i} *_s W^{(L)}_{j,i,:} + b^{(L)}_j \\
&= \sum_{p=1}^{k} (\mathbf{u}_j)_p (\sigma(x_{i_p}) - \sigma(-x_{i_p})) + c_j \\
&= \mathbf{u}_j^\top \mathbf{x}_\mathcal{I} + c_j.
\end{aligned}
$$

Thus we complete the proof of the first part.

*Bounding the parameter norm*. Under the above construction, we have:

- When $l = 1, 2, \ldots, L-1$, there exists an absolute constant $C > 0$ such that $\|W^{(l)}\|_F \leq C\sqrt{k}$ and $\mathbf{b}^{(l)} = 0$. Thus, by equation (52) we have

$$\left\|W^{(l)}\right\|_F + \sqrt{\frac{d}{2^{l-2}}} \left\|\mathbf{b}^{(l)}\right\|_2 \leq C\sqrt{k};$$

- Furthermore, the parameter norm in $L$-th layer satisfies

$$\left\|W^{(L)}\right\|_F + \left\|\mathbf{b}^{(L)}\right\|_2 \lesssim \sqrt{\sum_{i=1}^{m} \|\mathbf{u}_i\|_2^2} + \sqrt{\sum_{i=1}^{m} c_i^2}.$$

Combining them, we complete the proof. $\qquad\square$

## E.2 Proof of Theorem 4.6

We will need following approximation result of two-layer ReLU networks .

**Lemma E.2.** *(A restatement of (E et al., 2022, Theorem 4)) For any $f \in \mathcal{B}$, any probability distribution $P$ on $\mathcal{X} = [0,1]^d$ and integer $m \geqslant 1$, there exists a two-layer neural network $f_m(\mathbf{x}; \theta) = \frac{1}{m} \sum_{k=1}^m a_k \sigma \left( \mathbf{u}_k^\top \mathbf{x} + c_k \right)$ where $\theta$ denotes the parameters $\{(a_k, \mathbf{u}_k, c_k), k \in [m]\}$ in the neural network, such that*

$$\|f - f_m(\cdot; \theta)\|_{L^2(P)}^2 \leqslant \frac{3\|f\|_{\mathcal{B}}^2}{m}, \tag{78}$$

*Furthermore, we have*

$$\frac{1}{m} \sum_{j=1}^m |a_j| \left( \|\mathbf{u}_j\|_1 + |c_j| \right) \leqslant 2\|f\|_{\mathcal{B}} \tag{79}$$

**Proposition E.3.** *Let $\mathcal{X} = [0,1]^{4d}$. Suppose $h^*(\mathbf{x}) = g^*(\mathbf{x}_{\mathcal{I}})$ with $|\mathcal{I}| = k$. Consider deep ReLU CNNs with $L = \log_2(4d)$ and $C_l = 2k$ for $l \in [L-1]$ and $C_L = m$. Then, for any $g^* \in \mathcal{B}$, there exist $\theta^*$ such that*

$$\|h_{\theta^*} - h^*\|_{L^2(P)}^2 \leqslant \frac{3\|g^*\|_{\mathcal{B}}^2}{m} \tag{80}$$

*with the number of parameters being $\mathcal{O}(k^2 \log d + km)$ and*

$$\|\theta^*\|_{\mathcal{P}} \lesssim \sqrt{k} \log d + m + \|g^*\|_{\mathcal{B}} \tag{81}$$

**Proof.** Using lemma E.2, there exists a two layer ReLU network $f_m(\mathbf{x}_{\mathcal{I}}) = \frac{1}{m} \sum_{k=1}^m a_k \sigma \left( \mathbf{u}_k^\top \mathbf{x}_{\mathcal{I}} + c_k \right)$ such that

$$\|h^*(\mathbf{x}) - f_m(\mathbf{x}_{\mathcal{I}})\|_{L^2(\mu)}^2 = \|g^*(\mathbf{x}_{\mathcal{I}}) - f_m(\mathbf{x}_{\mathcal{I}})\|_{L^2(\mu)}^2 \leqslant \frac{3\|g^*\|_{\mathcal{B}}^2}{m}$$
$$\|\mathbf{u}_k\|_1 + |c_k| = 1, \forall k \tag{82}$$
$$\frac{1}{m} \sum_{k=1}^m |a_k| \leqslant 2\|g^*\|_{\mathcal{B}}$$

By Lemma 4.4, we can choose the weights and bias such that the feature map of $L$-th layer is

$$\mathbf{z}^{(L)}(\mathbf{x}) = \left( \sigma(\mathbf{u}_1^\top \mathbf{x}_{\mathcal{I}} + c_1), \ldots, \sigma(\mathbf{u}_m^\top \mathbf{x}_{\mathcal{I}} + c_m) \right)^\top \tag{83}$$

By setting the parameters of output layer as $W_o = \frac{1}{m}(a_1, \ldots, a_m)$, it is easy to check that $h_{\theta^*}(\mathbf{x}) = \mathcal{M}_o \circ \mathbf{z}^{(L)}(\mathbf{x}) = f_m(\mathbf{x}_{\mathcal{I}})$ and the parameter number $N_{\mathrm{cnn}} = \mathcal{O}(k^2 \log d + km)$.

Next, we turn to bound the parameter norm. Under the above parameter setting, we have

$$\|W_o\|_2 \leqslant \|W_o\|_1 \leqslant \frac{1}{m} \sum_{k=1}^m |a_k| \leqslant 2\|g^*\|_{\mathcal{B}}$$
$$\sqrt{\sum_{i=1}^m \|\mathbf{u}_i\|_2^2} + \sqrt{\sum_{i=1}^m c_i^2} \leqslant \sum_{i=1}^m \left( \|\mathbf{u}_i\|_1 + |c_i| \right) = m.$$

Applying Lemma 4.4 gives

$$\|\theta^*\|_{\mathcal{P}} \lesssim \|g^*\|_{\mathcal{B}} + \sqrt{k} \log d + \sqrt{\sum_{i=1}^m \|\mathbf{u}_i\|_2^2} + \sqrt{\sum_{i=1}^m c_i^2} \tag{84}$$
$$\lesssim \|g^*\|_{\mathcal{B}} + \sqrt{k} \log d + m$$

Thus, we complete the proof. $\quad\square$

**Proof of Theorem 4.6.** Let $\mathcal{H}_J^{\mathrm{CNN}} = \{h_\theta : \|\theta\| \leqslant J\}$. By proposition B.14 and the covering number satisfies

$$\mathcal{N}(\mathcal{H}_J^{\mathrm{CNN}}, \hat{\rho}_n, t) \leqslant \left(\frac{3\hat{\gamma}_n(J)}{t}\right)^{N_{\mathrm{cnn}}} \tag{85}$$

where $\hat{\gamma}_n(J) = \hat{M}_n J(1+J)^L$ and $N_{\mathrm{cnn}} = \mathcal{O}(k^2 \log d + km)$. Here we use the fact that $\bar{Q}_\sigma(\mathbf{x}) \leqslant 2^L$ for all $\mathbf{x}$ and $L = \log_2 d + 2$ for deep ReLU CNNs. In addition, note that

$$\gamma(J) = \mathbb{E}[\hat{\gamma}_n(J)] \leqslant 2^L J(1+J)^L \, \mathbb{E}\left[\sqrt{\frac{1}{n}\sum_{i=1}^n (\|\mathbf{x}_i\| + 1)^2}\right]$$

$$\leqslant 2^L J(1+J)^L \sqrt{\frac{1}{n}\sum_{i=1}^n \mathbb{E}[(\|\mathbf{x}_i\| + 1)^2]} \lesssim \sqrt{d} 2^L J(1+J)^L \tag{86}$$

where the second inequality follows from the Jensen's inequality since $\sqrt{\cdot}$ is concave.

Recalling we have $\alpha_{\mathcal{H}} = 1$ from equation (53) and applying Proposition B.14, we have

$$\left\|\pi_A \circ h_{\hat{\theta}_n} - h^*\right\|_{L^2(P)}^2 - \epsilon_* \lesssim \frac{\sigma^3 B}{(B-2A)^2} e^{-\frac{(B-2A)^2}{2\sigma^2}} + \lambda M_* + B^2\sqrt{\frac{\log(4/\delta)}{n}} + B^2\sqrt{\frac{N_{\mathrm{cnn}}\log(B\gamma(U_\lambda))}{n}},$$

where

$$\epsilon_* = \frac{3\|g^*\|_{\mathcal{B}}^2}{m}, \qquad M_* \lesssim \sqrt{k}\log d + m + \|g^*\|_{\mathcal{B}},$$

Taking $A = 1, B = 2 + \sigma\sqrt{\log n}, \lambda = \epsilon/n^2, m = \frac{6\|g^*\|_{\mathcal{B}}^2}{\epsilon}$ and applying Lemma B.13, we have

$$\left\|\pi_A \circ h_{\hat{\theta}_n} - h^*\right\|_{L^2(P)}^2 - \epsilon/2 \lesssim \frac{\sigma^2}{\sqrt{n}} + \epsilon\frac{\sqrt{k}\log d + \frac{\|g^*\|_{\mathcal{B}}^2}{\epsilon} + \|g^*\|_{\mathcal{B}}}{n^2} + B^2\sqrt{\frac{\log(1/\delta)}{n}}$$

$$+ B^2\sqrt{\frac{(k^2\log d + k\frac{\|g^*\|_{\mathcal{B}}^2}{\epsilon})(\log(B) + \log(\gamma(U_\lambda)))}{n}}, \tag{87}$$

where

$$\log(\gamma(U_\lambda)) \lesssim \log(d)\log\left(\frac{1}{2\lambda}\left(1 + \sigma^2 + B^2\sqrt{\frac{2\log(2/\delta)}{n}}\right) + \sqrt{k}\log d + \frac{\|g^*\|_{\mathcal{B}}^2}{\epsilon} + \|g^*\|_{\mathcal{B}}\right).$$

Simplify the right side of equation (87) becomes

$$\left\|\pi_A \circ h_{\hat{\theta}_n} - h^*\right\|_{L^2(P)}^2 - \epsilon/2 \lesssim$$

$$\mathrm{poly}(\|g^*\|_{\mathcal{B}}, k, \sigma, \log(1/\delta), \log\log n, \log(1/\epsilon))\frac{\log n}{\sqrt{n}}\sqrt{\log(d)\left(\log d + \frac{1}{\epsilon}\right)(\log n + \log\log d)}$$

This implies that, for a target accuracy $\epsilon$ and failure probability $\delta$,

$$n \geqslant \mathrm{poly}(\|g^*\|_{\mathcal{B}}, k, \sigma, \log\frac{1}{\delta}, \log\frac{1}{\epsilon})\left(\frac{\log(d)(\log\log d)^3}{\epsilon^3} + \frac{\log^2(d)(\log\log d)^3}{\epsilon^2}\right).$$

is enough.

$\square$

# F  Symmetries of Learning Algorithm and Proofs of Lemma 5.3 and 5.6

## F.1  Learning Algorithm and Group Equivariance

(Li et al., 2020, Appendix C) established a framework to verify the group equivariance of an iterative algorithm, which however considers neither stochastic nor adaptive algorithms such as Adam. In this

section, we provide an extension of (Li et al., 2020, Appendix C) to include these situations. It should be stressed that the extension is mostly straightforward compared with (Li et al., 2020, Appendix C) and we provide the extension here only for clarity and completeness.

In the following, denote $A \stackrel{\mathrm{d}}{=} B$ as $\mathrm{Law}(A) = \mathrm{Law}(B)$. Let $G_{\mathcal{X}}$ be a group acting on $\mathcal{X}$. Then, for any $\tau \in G_{\mathcal{X}}$ and $S_n = \{\mathbf{x}_i, y_i\}_{i=1}^n$, let $\tau(S_n) = \{(\tau(\mathbf{x}_i), y_i)\}_{i=1}^n$. We also occasionally write the group action $\tau\theta = \tau(\theta)$ for simplicity when it is clear from the context.

Recall that $h_\theta$ is our parametric model and $\theta \in \Theta = \mathbb{R}^p$. We make the following assumption.

**Assumption 3.** *Given a group $G_{\mathcal{X}}$ acting on $\mathcal{X}$. We assume that there exist a group $G_\Theta$ acting on $\Theta$ and a group isomorphism $Q : G_{\mathcal{X}} \to G_\Theta$ such that for all $\tau \in G_{\mathcal{X}}$,*

$$h_{Q(\tau)(\theta)} \circ \tau = h_\theta.$$

The above assumption is satisfied by both FCNs and LCNs and we will verify it later. Here, we take linear regression as an example to gain some intuition. In such a case, $h_\theta(\mathbf{x}) = \langle \mathbf{w}, \mathbf{x} \rangle$. Given the group $G_{\mathcal{X}} = O(d)$ acting on the input domain, we can set $Q = \mathrm{id}$ and $G_\Theta = O(d)$. Then, for any $U \in O(d)$, we have $h_{Q(U)(\theta)} \circ U(\mathbf{x}) = \langle U\mathbf{w}, U\mathbf{x} \rangle = \langle \mathbf{w}, \mathbf{x} \rangle = h_\theta(\mathbf{x})$.

Suppose Assumption 3 holds. Given a learning algorithm $\mathbb{A} : (\mathcal{X} \times \mathcal{Y})^n \mapsto \mathcal{M}(\Theta)$, we say $\mathbb{A}$ is $G_{\mathcal{X}}$-equivariant if $\forall S_n \in (\mathcal{X} \times \mathcal{Y})^n$ and $\tau \in G_{\mathcal{X}}$:

$$\mathbb{A}(\tau(S_n)) \stackrel{\mathrm{d}}{=} Q(\tau) \circ \mathbb{A}(S_n). \tag{88}$$

We can also informally define the equivariance in the model space as follows

$$h_{\mathbb{A}(\tau(S_n))} \circ \tau \stackrel{\mathrm{d}}{=} h_{\mathbb{A}(S_n)}, \tag{89}$$

where $h_{\mathbb{A}(S_n)}$ and $h_{\mathbb{A}(\tau(S_n))}$ should be understood as random variables taking values in the space of all the algorithms $\mathcal{A}$. The definition (89) is intuitive and has been adopted in Abbe and Boix-Adserà (2022); Li et al. (2020) but it should be stressed that (89) is not rigorous as it is generally unclear how to deal with the measurability issue in $\mathcal{A}$, the space of algorithm. We thus will adopt (88) for its rigorous nature.

**Iterative algorithm.** In the following, we will focus on the learning algorithm given by

$$\begin{aligned} &\theta_0 \sim P_{\mathrm{init}} \\ &\theta_{t+1} = F_{t+1}(\theta_t, \ldots, \theta_0, S_n, \xi_{t+1}) \ \text{ for } t = 0, 1, \ldots, T-1, \end{aligned} \tag{90}$$

where $F_t : (\Theta)^t \times (\mathcal{X} \times \mathcal{Y})^n \times \Omega \mapsto \Theta$ is a deterministic update map and $\{\xi_t\}_{t=1}^T$ are i.i.d. freshly generated random variables taking values in $\Omega$, which encode the algorithm randomness and $\xi_t$ is independent of $S_n$ and $\{\theta_0, \theta_1, \ldots, \theta_t\}$.

Take SGD as a concrete example. Let $\Omega = \{0, 1\}^n$ denote the set of minibatch selection masks and the corresponding algorithm map is given by

$$F_{t+1}(\theta_t, \ldots, \theta_0, S_n, \xi_{t+1}) = \theta_t - \eta_t \nabla_\theta \left( \frac{1}{n} \sum_{i=1}^n (\xi_{t+1})_i \, \ell(h_{\theta_t}(\mathbf{x}_i), y_i) + \lambda r(\|\theta_t\|_p) \right).$$

Note that one should not confuse $\xi_t$ with the SGD noise, where latter is state-dependent.

**Assumption 4.** $\quad \bullet$ $P_{\mathrm{init}}$ *is $G_\Theta$-invariant.*

$\bullet$ $\forall \tau \in G_{\mathcal{X}}, \theta \in \Theta, t \in \mathbb{N}, S_n \in (\mathcal{X} \times \mathcal{Y})^n$, *and* $\xi \in \Omega$:

$$Q(\tau) \circ F_{t+1}(\theta_t, \ldots, \theta_0, S_n, \xi) = F_{t+1}(Q(\tau)\theta_t, \ldots, Q(\tau)\theta_0, \tau(S_n), \xi). \tag{91}$$

The second assumption can be thought as that the update map $F_t(\cdot)$ is equivariant under the joint group action $(\tau, Q(\tau))$. Still taking the linear regression as an example, let $X = (\mathbf{x}_1, \ldots, \mathbf{x}_n) \in \mathbb{R}^{d \times n}$ be $n$ inputs and $\mathbf{y} \in \mathbb{R}^n$ be the labels. Then, the GD update for minimizing $\hat{L}(\mathbf{w}) := \frac{1}{2} \|X^\top \mathbf{w} - \mathbf{y}\|_2^2$ would be

$$\mathbf{w}_{t+1} = \mathbf{w}_t - \eta X \left( X^\top \mathbf{w}_t - \mathbf{y} \right) =: F_{t+1}(\mathbf{w}_t, S_n).$$

Let $G_{\mathcal{X}} = G_{\Theta} = G_{\text{ort}}(d)$, $Q = \text{id}$. We thus have for any $U \in O(d)$, $Q(U) = U$ and

$$Q(U)F_{t+1}(\mathbf{w}_t, S_n) = U\left(\mathbf{w}_t - \eta X\left(X^\top \mathbf{w}_t - \mathbf{y}\right)\right)$$
$$= U\mathbf{w}_t - \eta(UX)\left((UX)^\top(U\mathbf{w}_t) - \mathbf{y}\right) = F_{t+1}(Q(U)\mathbf{w}_t, U(S_n)),$$

which verifies the equivariance (91).

**Proposition F.1.** *Let $\mathbb{A}_T$ denote the iterative algorithm* (90)*. Under Assumption 3 and 4, we have for any $T \in \mathbb{N}$, $\mathbb{A}_T$ is $G_{\mathcal{X}}$-equivariant.*

*Proof.* Fix $S_n \in (\mathcal{X} \times \mathcal{Y})^n$ and $\tau \in G_{\mathcal{X}}$. Let $\{\theta_t\}_{t=0}^\top$ and $\{\widetilde{\theta}_t\}_{t=0}^\top$ be the trajectories independently generated by the iterative algorithm by using the data $S_n$ and $\tau(S_n)$, respectively. By the definition (88), it suffices to prove that for any $t \in \mathbb{N}$,

$$(\widetilde{\theta}_t, \ldots, \widetilde{\theta}_0) \overset{\mathrm{d}}{=} (Q(\tau)\theta_t, \ldots, Q(\tau)\theta_0). \tag{92}$$

Next we prove it by induction.

When $T = 0$, $\theta_0, \widetilde{\theta}_0 \sim P_{\text{init}}$. For any $\tau \in G_{\mathcal{X}}$, we have $Q(\tau) \in G_{\Theta}$. The assumption that $P_{\text{init}}$ is $G_{\Theta}$-invariant implies $\widetilde{\theta}_0 \overset{\mathrm{d}}{=} Q(\tau)\theta_0$.

Assume (92) holds for $T = 0, 1, \ldots, t$. Let $\{\xi_t\}$ and $\{\widetilde{\xi}_t\}$ be the "noise" used to generate $\{\theta_t\}$ and $\{\widetilde{\theta}_t\}$, respectively. Then, we have

$$(Q(\tau)\theta_{t+1}, \ldots, Q(\tau)\theta_0) \overset{\mathrm{d}}{=} (Q(\tau)F_{t+1}(\theta_t, \ldots, \theta_0, S_n, \xi_{t+1}), Q(\tau)\theta_t, \ldots, Q(\tau)\theta_0)$$
$$\overset{\mathrm{d}}{=} (F_{t+1}(Q(\tau)\theta_t, \ldots, Q(\tau)\theta_0, \tau(S_n), \xi_{t+1}), Q(\tau)\theta_t, \ldots, Q(\tau)\theta_0)$$
$$\overset{\mathrm{d}}{=} (F_{t+1}(\widetilde{\theta}_t, \ldots, \widetilde{\theta}_0, \tau(S_n), \widetilde{\xi}_{t+1}), \widetilde{\theta}_t, \ldots, \widetilde{\theta}_0)$$
$$\overset{\mathrm{d}}{=} (\widetilde{\theta}_{t+1}, \ldots, \widetilde{\theta}_0),$$

where the second step follows from Assumption (4); the third step follows from the assumption that (92) holds for $T = t$. Thus, we prove that (92) holds for $T = t + 1$.

By induction, we complete the proof. $\qquad\square$

### F.2 Verifying the Group Equivariance of SGD and Adam

Let $\ell(\cdot, \cdot)$ be a general loss function. Consider a regularized empirical risk

$$\hat{L}_n(\theta) := \frac{1}{n}\sum_{i=1}^n \ell(h_\theta(\mathbf{x}_i), y_i) + r(\|\theta\|_p)$$

where $p \geqslant 1$ and and $r : [0, +\infty) \to [0, +\infty)$.

At the $t$-th step, let $S_t = \{i_1, \ldots, i_k\}$ be $k$ i.i.d. indices uniformly drawn from $[n]$. Define the minibatch risk by

$$\hat{L}_{S_t}(\theta) := \frac{1}{|S_t|}\sum_{i \in S_t} \ell(h_\theta(\mathbf{x}_i), y_i) + r(\|\theta\|_p).$$

In each step, stochastic gradient descent (SGD) updates as follows

$$\theta_{t+1} = \theta_t - \eta_t \nabla_\theta \hat{L}_{S_t}(\theta_t).$$

Adam optimizer (Kingma and Ba, 2014) updates as follows

$$\mathbf{v}_{t+1} = \alpha\mathbf{v}_t + (1 - \alpha)\left(\nabla_\theta \hat{L}_{S_t}(\theta_t)\right)^2$$
$$\mathbf{m}_{t+1} = \beta\mathbf{m}_t + (1 - \beta)\nabla_\theta \hat{L}_{S_t}(\theta_t) \tag{93}$$
$$\theta_{t+1} = \theta_t - \eta_t \frac{\mathbf{m}_{t+1}/\left(1 - \beta^{t+1}\right)}{\sqrt{\mathbf{v}_{t+1}/\left(1 - \alpha^{t+1}\right)} + \epsilon\mathbf{1}},$$

where $\alpha, \beta \in (0, 1), \epsilon > 0$, and the square and division should be understood in an element-wise manner. We just consider initialization $\mathbf{v}_0 = \mathbf{m}_0 = \mathbf{0}$ for simplicity. For both SGD and Adam, $\eta_t$ is the learning rate at the $t$-th step.

**FCNs: Proof of Lemma 5.6**  For $h_\theta^{\mathrm{fcn}}$, write $\theta = (\mathbf{w}_1, \ldots, \mathbf{w}_m, \bar\theta) \in \mathbb{R}^p$, where $m$ is the width of the first layer, $\mathbf{w}_j \in \mathbb{R}^d$ for $j = 1, \ldots, m$ denotes the weights of the first layer, and $\bar\theta$ denotes other parameters. Then, we can write $h_\theta^{\mathrm{fcn}}(\mathbf{x}) = g_{\bar\theta}(\mathbf{w}_1^\top \mathbf{x}, \ldots, \mathbf{w}_m^\top \mathbf{x})$ for some $g : \mathbb{R}^m \times \mathbb{R}^{p-md} \mapsto \mathbb{R}$.

Consider $G_{\mathcal{X}} = G_{\mathrm{ort}}(d)$ and

$$Q : U \to \mathrm{diag}\{U, U, \ldots, U, I_{p-md}\}. \tag{94}$$

Let $G_\Theta := \{Q(U) : U \in G_{\mathcal{X}}\}$. Then, it is not hard to verify that $G_\Theta$ is a group under matrix product and $Q : G_{\mathcal{X}} \mapsto G_\Theta$ is a group isomorphism.

First, it is not hard to verify that the Gaussian initialization $P_{\mathrm{init}}$ is invariant under $G_\Theta$. To verify (91) for SGD, we can simply consider the case where $n = 1$ and $S_n = \{(\mathbf{x}, y)\}$ without loss of generality. First, we can show that there exist functions $\{s_i(\cdot)\}_{i=1}^m$ and $O(\cdot)$ such that

$$F_{t+1}(\theta_t, \ldots, \theta_0, S_n, \xi_{t+1}) = \theta_t - \eta_t \nabla_\theta \ell(h_{\theta_t}^{\mathrm{fcn}}(\mathbf{x}), y) = \theta_t - \eta_t \begin{pmatrix} s_1(W^\top \mathbf{x}, \bar\theta_t)\mathbf{x} \\ s_2(W^\top \mathbf{x}, \bar\theta_t)\mathbf{x} \\ \vdots \\ s_m(W^\top \mathbf{x}, \bar\theta_t)\mathbf{x} \\ O(W^\top \mathbf{x}, \bar\theta_t) \end{pmatrix},$$

where $W = (\mathbf{w}_1, \mathbf{w}_2, \ldots, \mathbf{w}_m) \in \mathbb{R}^{d \times m}$. It is not hard to verify that (91) holds for the above update map. Then, Lemma 5.6 follows trivially from Proposition F.1.  $\square$

**LCNs: Proof of Lemma 5.3**  Here we only prove Lemma 5.3 for the Adam optimizer since the SGD case is similar to the proof above.

WLOG, we let $C_1 = 1$ in the LCN model. Denote by $\bar\theta = W_{1,1,:}^{(1)}$ the the filter weights of the first layer and $\bar\theta^c$ all the other parameters. Then, $\theta = \{\bar\theta, \bar\theta^c\}$ and $W_{1,1,:}^{(1)} \in \mathbb{R}^{D_0}$. Let $G_{\mathcal{X}} = G_{\mathrm{loc}}$ and define

$$Q : U \to \mathrm{diag}\{U, I\}. \tag{95}$$

Let $G_\Theta = \{Q(U) : U \in G_{\mathcal{X}}\}$. It is not hard to verify that $G_\Theta$ is a group under the matrix multiplication and $Q$ is the group isomorphism.

First, it is obvious that under Assumption 1, $P_{\mathrm{init}}$ is $G_{\mathrm{loc}}$-invariant. To verify (91) for Adam, we can simply consider the case where $r(\cdot) = 0$, $n = 1$ and $S_n = \{(\mathbf{x}, y)\}$ without loss of generality. In this case, for the first step, we have

$$F_1(\theta_0, S_n, \xi_1) = \theta_0 - c_{\alpha,\beta}\eta_0 \frac{\nabla_\theta \ell(h_{\theta_0}(\mathbf{x}), y)}{\sqrt{|\nabla_\theta \ell(h_{\theta_0}(\mathbf{x}), y)|^2 + \epsilon \mathbf{1}}} = \theta_0 - \eta_0 \begin{pmatrix} s_1(\bar\theta_0 \star_s \mathbf{x}, \bar\theta_0^c)\mathbf{x}_{I_1} \\ s_2(\bar\theta_0 \star_s \mathbf{x}, \bar\theta_0^c)\mathbf{x}_{I_2} \\ \vdots \\ s_{D_1}(\bar\theta_0 \star_s \mathbf{x}, \bar\theta_0^c)\mathbf{x}_{I_{D_1}} \\ O(\bar\theta_0 \star_s \mathbf{x}, \bar\theta_0^c) \end{pmatrix} \tag{96}$$

for some function $\{s_i(\cdot)\}_{i=1}^{D_1}$ and $O(\cdot)$. Here we recall that in our paper $s = 2$ and the local linear operater $\star_s : \mathbb{R}^{ks} \times \mathbb{R}^{ks} \mapsto \mathbb{R}^k$ is defined by $\mathbf{v} \star_s \mathbf{w} = (\mathbf{v}_{I_1}^\top \mathbf{w}_{I_1}, \mathbf{v}_{I_2}^\top \mathbf{w}_{I_2}, \ldots, \mathbf{v}_{I_k}^\top \mathbf{w}_{I_k})$, where $I_j = [(j-1)s + 1, js]$ denotes the indices of $j$-th patch.

Noting $(U\bar\theta_0) \star_s (U\mathbf{x}) = \bar\theta_0 \star_s \mathbf{x}$ for any $U \in G_{\mathrm{loc}}$, it is therefore not hard to verify that (91) holds for the update map (96). Thus we show the $G_{\mathrm{loc}}$-equivariance of Adam for training LCNs with $T = 1$. The case of $T > 1$ can be shown in the same way.  $\square$

**Remark F.2.** *Note that the case of FCNs was established in (Li et al., 2020, Corollary C.2) but the proof there is not fully rigorous. We here provide a completely rigorous proof. The case of LCNs under Gaussian initialization was studied in Xiao and Pennington (2022), where the equivariance group is $G_{ort}(2) \otimes I_{2d}$. We instead show that under much milder condition on initialization, the equivariance group becomes the local permutation group (10). It should be stressed that the major contribution of this section is providing a rigorous/unified framework to verify these group equivariance instead of yielding new insights.*

# G CNNs vs. LCNs: Proofs of Section 5.2

## G.1 Proof of Theorem 5.4

By Lemma 5.1, we only need to lower bound the minimax error of learning the enlarged class $\{\bar{h}^*\} \circ G_{\mathrm{loc}}$. To this end, we need to find a proper packing of this enlarged class and then apply the Fano's method, i.e., Proposition B.20. To this end, we will prove

**Lemma G.1.** *There exist absolute positive constants $C, c_1, c_2 > 0, c_3 > 1$ such that if $d \geqslant C$ and $A_0 \geqslant C$, there exists $h_1, h_2, \ldots, h_M \in \{\bar{h}^*\} \circ G_{\mathrm{loc}}$ satisfying that*

$$\sup_{i,j} \|h_i - h_j\|^2_{L^2(P)} \leqslant c_2, \ \inf_{i \neq j} \|h_i - h_j\|^2_{L^2(P)} \geqslant \frac{1}{4} c_1, \ \text{ and } M \geqslant c_3^d.$$

Assuming Lemma G.1 is right for now and applying the above packing to Fano's inequality (Proposition B.20), we have

$$\inf_{\mathbb{A} \in \mathcal{A}} \sup_{h \in \{\bar{h}^*\} \circ G_{\mathrm{loc}}} \mathbb{E}\|h^{\mathrm{lcn}}_{\mathbb{A}(S_n)} - h\|^2_{L^2(P)} \geqslant A\left(1 - \frac{n}{2\sigma^2 M^2 \log M}\sum_{j,j'=1}^M \|h_j - h_{j'}\|^2_{L^2(P)} - \frac{\log 2}{\log M}\right)$$

$$\geqslant \frac{1}{16} c_1 \left(1 - \frac{c_2 n}{2\sigma^2 d \log c_3} - \frac{\log 2}{d \log c_3}\right)$$

(97)

Taking $n$ such that the RHS of the above inequality satisfies

$$\frac{1}{16} c_1 \left(1 - \frac{c_2 n}{2\sigma^2 d \log c_3} - \frac{\log 2}{d \log c_3}\right) \leqslant \frac{c_1}{32} =: \epsilon_0 \tag{98}$$

gives $n = \Omega(\sigma^2 d)$. Thus, we can conclude that $\mathcal{C}(\{\bar{h}^*\} \circ G_{\mathrm{loc}}, \epsilon_0) = \Omega(\sigma^2 d)$. Thus, we complete the proof of Theorem 5.4. $\square$

### G.1.1 Proof of Lemma G.1

We now show how to construct a packing of $\{\bar{h}^*\} \circ G_{\mathrm{loc}}$ to satisfy the condition in Lemma G.1. The key idea is to reduce the problem to pack the equivariance group $G_{\mathrm{loc}}$. For $U = \mathrm{diag}\,(U_1, U_2, \ldots, U_{2d}) \in G_{\mathrm{loc}}$, denote by $U_i \in \mathbb{R}^{2 \times 2}$ the $i$-th block matrix. Define a metric over $G_{\mathrm{loc}}$: for any $U, U' \in G_{\mathrm{loc}}$,

$$\rho_{\mathrm{loc}}(U, U') := \sharp\{i : U_i \neq U'_i\}. \tag{99}$$

In addition, we consider a subgroup of $G_{\mathrm{loc}}$:

$$G_{\mathrm{semiloc}} = \{\mathrm{diag}\,(U_1, U_2, \ldots, U_{2d}) \in G_{\mathrm{loc}} : U_i = I_2 \text{ for } i > d\} \tag{100}$$

Clearly, a packing of $\{\bar{h}^*\} \circ G_{\mathrm{semiloc}}$ is also a packing of $\{\bar{h}^*\} \circ G_{\mathrm{loc}}$. For $G_{\mathrm{semiloc}}$, we have

**Lemma G.2.** *There exist absolute constants $C, c_1, c_2 > 0$ such that if $d, A_0 \geqslant C$, for any $\tau, \tau' \in G_{\mathrm{semiloc}}$, we have*

$$c_1 \frac{\rho_{\mathrm{loc}}(\tau, \tau')}{d} \leqslant \left\|\bar{h}^* \circ \tau - \bar{h}^* \circ \tau'\right\|^2_{L^2(P)} \leqslant c_2 \frac{\rho_{\mathrm{loc}}(\tau, \tau')}{d}. \tag{101}$$

**Proof idea.** The proof of this lemma is quite technically lengthy and is thus deferred to Appendix I.1. Recall that $\bar{h}^* = \pi_{A_0} \circ h^*$. If there is no truncation, i.e., $A_0 = \infty$, the proof is straightforward in the sense that a direct calculation gives $\|h^* \circ \tau - h^* \circ \tau'\|^2_{L^2(P)} = c\rho_{\mathrm{loc}}(\tau, \tau')/d$ for some absolute constant $c$. When $A_0$ is finite but large enough, by concentration inequalities, we can expect that the truncated target function works in a way similar to the orginal one. Indeed the lengthy part in the proof is to estimate the influence of truncation.

In the following, we show that $G_{\mathrm{semiloc}}$ is isometric to the Hamming space:

**Definition G.3** (Hamming Space). The Hamming cube $\{0, 1\}^n$ consists of all binary strings of length $n$. The Hamming distance $d_H(x, y)$ between two binary strings is defined as the number of bits where $x$ and $y$ disagree, i.e.

$$d_H(x, y) := \#\{i : x(i) \neq y(i)\}, \quad x, y \in \{0, 1\}^n.$$

We call $(\{0, 1\}^n, d_H)$ the Hamming space.

Specifically, it is obvious that

- $(G_{\text{loc}}, \rho_{\text{loc}})$ and $(\{0,1\}^{2d}, d_H)$ are isometric.
- $(G_{\text{semiloc}}, \rho_{\text{loc}})$ and $(\{0,1\}^d, d_H)$ are isometric.

For the Hamming space, we have the following bounds for packing and covering numbers.

**Lemma G.4.** *(Vershynin, 2018, Excercise 4.2.16) Let $K = \{0,1\}^n$. For every $0 < m \leqslant n$, we have*

$$\mathcal{P}(K, d_H, m) \geqslant \mathcal{N}(K, d_H, m) \geqslant \frac{2^n}{\sum_{k=0}^{\lfloor m \rfloor} \binom{n}{k}}$$

*Proof.* Without loss of generality we set $m$ as an integer. The first inequality follows trivially from the definition and thus, we only need to prove the second one.

Set $B_{m,x} = \{y \in K : d_H(x,y) \leqslant m\}$. For any $x \in K$, any $k \leqslant m$, there are $\binom{n}{k}$ elements satisfing $d_H(x,y) = k$. This is because the ways of choosing $k$ elements from $n$ elements are $\binom{n}{k}$, and $d_H(x,y) = k$ if and only if $x$ and $y$ have $k$ different coordinates. By taking a union bound we have $|B_{m,x}| \leqslant \sum_{k=0}^{m} \binom{n}{k}$, where $|\cdot|$ denotes the cardinality. The ball of radius $m$ and center $x$ can only cover at most $|B_{m,x}|$ elements, However there are $2^n$ elements in $K$. So we have

$$\mathcal{N}(K, d_H, m) \geqslant \frac{2^n}{\sup_x |B_{m,x}|} \geqslant \frac{2^n}{\sum_{k=0}^{m} \binom{n}{k}} \tag{102}$$

$\square$

By the isometries and Lemma G.4, we trivially have

**Lemma G.5.** *For any $0 < m \leqslant d$,*

$$\mathcal{P}(G_{\text{loc}}, \rho_{\text{loc}}, m) \geqslant \mathcal{N}(G_{\text{loc}}, \rho_{\text{loc}}, m) \geqslant \frac{4^d}{\sum_{k=0}^{\lfloor m \rfloor} \binom{2d}{k}}$$

$$\mathcal{P}(G_{\text{semiloc}}, \rho_{\text{loc}}, m) \geqslant \mathcal{N}(G_{\text{semiloc}}, \rho_{\text{loc}}, m) \geqslant \frac{2^d}{\sum_{k=0}^{\lfloor m \rfloor} \binom{d}{k}}$$

To further simplify the lower bounds, we need

**Lemma G.6.** *(Vershynin, 2018, Exercise 0.0.5) Given $m \in [n]$, $\sum_{k=0}^{m} \binom{n}{k} \leqslant \left(\frac{en}{m}\right)^m$.*

*Proof.* To prove this we only need to prove $\left(\frac{m}{n}\right)^m \sum_{k=0}^{m} \binom{n}{k} \leqslant e^m$, and we observe that

$$\left(\frac{m}{n}\right)^m \sum_{k=0}^{m} \binom{n}{k} \leqslant \sum_{k=0}^{m} \left(\frac{m}{n}\right)^k \binom{n}{k} \leqslant \sum_{k=0}^{m} \frac{m^k}{k!} \leqslant \sum_{k=0}^{+\infty} \frac{m^k}{k!} = e^m \tag{103}$$

$\square$

*Proof of Lemma G.1.* Note that for any $\tau, \tau' \in G_{\text{semiloc}}$, $\rho_{\text{loc}}(\tau, \tau') \leqslant d$. Then, applying Lemma G.2, we have for any $h = \bar{h}^* \circ \tau, h' = \bar{h}^* \circ \tau' \in \{\bar{h}^*\} \circ G_{\text{semiloc}}$ that

$$\|h - h'\|_{L^2(P)}^2 \leqslant \frac{c_2 \rho_{\text{loc}}(\tau, \tau')}{d} \leqslant c_2. \tag{104}$$

In addition, Lemma G.2 implies $\|h - h'\|_{L^2(P)}^2 \geqslant \frac{c_1}{d} \rho_{\text{loc}}(\tau, \tau')$. Thus, for $0 < \delta \leqslant \frac{1}{2}\sqrt{c_1}$:

$$\mathcal{P}(\{\bar{h}^*\} \circ G_{semiloc}, L^2(P), \delta) \geqslant \mathcal{N}(\{\bar{h}^*\} \circ G_{\text{semiloc}}, L^2(P), \delta)$$

$$\geqslant \mathcal{N}(G_{\text{semiloc}}, \rho_{\text{loc}}, d\delta^2/c_1)$$

$$\geqslant \frac{2^d}{\sum_{m=0}^{\lfloor \frac{\delta^2 d}{c_1} \rfloor} \binom{d}{m}},$$

where the last step follows from Lemma G.5. By Lemma G.6 and taking $\delta^* = \frac{1}{2}\sqrt{c_1}$, we have

$$\mathcal{P}(\{\bar{h}^*\} \circ G_{\text{semiloc}}, L^2(P), \frac{1}{2}\sqrt{c_1}) \geqslant \frac{2^d}{\sum_{m=0}^{\lfloor d/4 \rfloor} \binom{d}{m}} \geqslant \left(\frac{2}{(5e)^{1/4}}\right)^d. \tag{105}$$

Let $c_3 = \frac{2}{(5e)^{1/4}}$, which is greater than 1.

Combining equation (104) and (105), we complete the proof. $\quad\square$

## G.2 Proof of Theorem 5.2

Let $\sigma(z) := \mathrm{ReLU}(z)$ in this subsection for notation simplicity. We first recall the architecture of the CNN model: $L = \log_2(4d)$, channel number $C_1 = C_L = 4$, $C_l = 2$ for $l = 2, \ldots, L-1$, and activation functions $\sigma_1(z) = \sigma_L(z) = \sigma^2(z), \sigma_l(z) = \sigma(z)$ for $l = 2, \ldots, L-1$.

*Step I: Representing the target using CNNs.*
**Lemma G.7.** *There is a parametrization $\theta^*$ such that $h_{\theta^*}^{\text{cnn}} = h^*$ and $\|\theta^*\|_{\mathcal{P}} \lesssim \log d$.*

*Proof.* We provide a constructive proof of this lemma. We fist set all the bias to zeros. Then we set the parameter of the filters $W^{(l)} \in \mathbb{R}^{C_l \times C_{l-1} \times 2}$ as follows.

- For $l = 1$, $W_{1,1,:}^{(1)} = -W_{2,1,:}^{(1)} = (1, 0)$, and $W_{3,1,:}^{(1)} = -W_{4,1,:}^{(1)} = (0, 1)$.

- For $l = 2$, $W_{1,1,:}^{(2)} = -W_{1,2,:}^{(2)} = W_{1,3,:}^{(2)} = -W_{1,4,:}^{(2)} = (1, 1)$ and $W_{2,i,:}^{(2)} = -W_{1,i,:}^{(2)}$ for $i = 1, 2, 3, 4$.

- For $l = 3, \ldots, L-1$, $W_{1,1,:}^{(l)} = -W_{1,2,:}^{(l)} = (1, 1)$ and $W_{2,i,:}^{(l)} = -W_{1,i,:}^{(l)}$ for $i = 1, 2$.

- For $l = L$, $W_{1,1,:}^{(L)} = -W_{2,1,:}^{(L)} = (1, 1)$, $W_{3,1,:}^{(L)} = -W_{4,1,:}^{(L)} = (1, -1)$ and $W_{j,2,:}^{(L)} = -W_{j,1,:}^{(L)}$ for $j = 1, 2, 3, 4$.

- For the final linear layer, $W_o = \frac{1}{4d}(1, 1, -1, -1)$.

First, it is easy to check the output of the first two layer by direct calculation. Concretely, we have

$$(\mathbf{z}^{(1)})^\top = \begin{pmatrix} \sigma^2(x_1) & \sigma^2(x_3) & \cdots & \sigma^2(x_{4d-1}) \\ \sigma^2(-x_1) & \sigma^2(-x_3) & \cdots & \sigma^2(-x_{4d-1}) \\ \sigma^2(x_2) & \sigma^2(x_4) & \cdots & \sigma^2(x_{4d}) \\ \sigma^2(-x_2) & \sigma^2(-x_4) & \cdots & \sigma^2(-x_{4d})) \end{pmatrix} \in \mathbb{R}^{4 \times 2d} \tag{106}$$

and

$$(\mathbf{z}^{(2)})^\top = \begin{pmatrix} \sigma\left(x_1^2 - x_2^2 + x_3^2 - x_4^2\right) & \cdots & \sigma\left(x_{4d-3}^2 - x_{4d-2}^2 + x_{4d-1}^2 - x_{4d}^2\right) \\ \sigma\left(-x_1^2 + x_2^2 - x_3^2 + x_4^2\right) & \cdots & \sigma\left(-x_{4d-3}^2 + x_{4d-2}^2 - x_{4d-1}^2 + x_{4d}^2\right) \end{pmatrix} \in \mathbb{R}^{2 \times d} \tag{107}$$

Next, we are going to prove the following conclusion for $l = 2, \ldots, L-1$ by induction

$$(\mathbf{z}^{(l)})^\top = \begin{pmatrix} \sigma\left(\sum_{k=1}^{2^{l-1}}(x_{2k-1}^2 - x_{2k}^2)\right) & \cdots & \sigma\left(\sum_{k=1}^{2^{l-1}}(x_{4d-2k+3}^2 - x_{4d-2k+2}^2)\right) \\ \sigma\left(-\sum_{k=1}^{2^{l-1}}(x_{2k-1}^2 - x_{2k}^2)\right) & \cdots & \sigma\left(-\sum_{k=1}^{2^{l-1}}(x_{4d-2k+3}^2 - x_{4d-2k+2}^2)\right) \end{pmatrix} \in \mathbb{R}^{2 \times d/2^{l-2}} \tag{108}$$

First, by (107), it holds for $l = 2$. Next we assume (108) holds for the case of $l-1$ and verify it holds also for the case of $l$.

By the definition of our CNN model, we have

$$(\mathcal{T}_l \mathbf{z}^{(l-1)})_{i,j} = \mathbf{z}_{2i-1,1}^{(l-1)} W_{j,1,1}^{(l)} + \mathbf{z}_{2i-1,2}^{(l-1)} W_{j,2,1}^{(l)} + \mathbf{z}_{2i,1}^{(l-1)} W_{j,1,2}^{(l)} + \mathbf{z}_{2i,2}^{(l-1)} W_{j,2,2}^{(l)} \tag{109}$$

when $j = 1$,

$$(\mathcal{T}_l \mathbf{z}^{(l-1)})_{i,1} = \mathbf{z}^{(l-1)}_{2i-1,1} - \mathbf{z}^{(l-1)}_{2i-1,2} + \mathbf{z}^{(l-1)}_{2i,1} - \mathbf{z}^{(l-1)}_{2i,2}$$

$$= \sum_{k=1}^{2^{l-2}} \left( x^2_{2k-1+(2i-2)2^{l-1}} - x^2_{2k+(2i-2)2^{l-1}} \right) + \sum_{k=1}^{2^{l-2}} \left( x^2_{2k-1+(2i-1)2^{l-1}} - x^2_{2k+(2i-1)2^{l-1}} \right)$$

$$= \sum_{k=1}^{2^{l-1}} \left( x^2_{2k-1+(i-1)2^l} - x^2_{2k+(i-1)2^l} \right),$$

which verifies (108) for $j = 1$. The case of $j = 2$ can be verified in the same way.

So by now we have verified that the output of the $L - 1$ layer is

$$(\mathbf{z}^{(L-1)})^\top = \begin{pmatrix} \sigma\left(\sum_{i=1}^d (x^2_{2i-1} - x^2_{2i})\right) & \sigma\left(\sum_{i=1}^d (x^2_{2d+2i-1} - x^2_{2d+2i})\right) \\ \sigma\left(-\sum_{i=1}^d (x^2_{2i-1} - x^2_{2i})\right) & \sigma\left(-\sum_{i=1}^d (x^2_{2d+2i-1} - x^2_{2d+2i})\right) \end{pmatrix} \in \mathbb{R}^{2\times 2} \quad (110)$$

Noting $\sigma(x) - \sigma(-x) = x$, we can recover $\sum_{i=1}^d (x^2_{2i-1} - x^2_{2i})$ and $\sum_{i=1}^d (x^2_{2i+2d-1} - x^2_{2i+2d})$ by using the feature stored in $\mathbf{z}^{(L-1)}$.

Recall that in our last layer, our activation function is $\sigma_L(x) = \sigma^2(x)$, and we have $\sigma^2(x) + \sigma^2(-x) = x^2$. Noting that $4\alpha\beta = (\alpha + \beta)^2 - (\alpha - \beta)^2$ and viewing $\sum_{i=1}^d (x^2_{2i-1} - x^2_{2i})$, $\sum_{i=1}^d (x^2_{2i+2d-1} - x^2_{2i+2d})$ as $\alpha$ and $\beta$, respectively, a simple calculation would lead to $h^{\mathrm{cnn}}_{\theta^*}(\mathbf{x}) = h^*(\mathbf{x})$.

Note that for all $l \in [L]$, $\mathbf{b}^{(l)} = 0$, $C_l \leqslant 4$. Then, it is obvious that there exists an absolute constant $C > 0$ such that $\|W^{(l)}\|_F \leqslant C$ for any $l \in [L]$ and $\|W_o\|_2 \leqslant C$. Thus, we have $\|\theta^*\|_{\mathcal{P}} = \|W_o\|_2 + \sum_{l=1}^L \|W^{(l)}\|_F \lesssim L \lesssim \log d$. $\qquad\square$

*Step II: Estimating the sample complexity.*

Recall the input distribution $P = \mathcal{N}(0, I_{4d})$. Let $\mathcal{H}^{\mathrm{CNN}}_J = \{h_\theta : \|\theta\| \leqslant J\}$, where we recall $\|\cdot\|$ is defined in Appendix C. By Proposition B.14, the covering number satisfies

$$\mathcal{N}(\mathcal{H}^{\mathrm{CNN}}_J, \hat{\rho}_n, t) \leqslant \left( \frac{3\hat{\gamma}_n(J)}{t} \right)^{N_{\mathrm{cnn}}} \quad (111)$$

where $\hat{\gamma}_n(J) = \hat{M}_n J(1 + J)^L$, $\hat{M}_n = \sqrt{\frac{1}{n}\sum_{i=1}^n (\bar{Q}_\sigma(\mathbf{x}_i))^2 (\|\mathbf{x}_i\|_2 + 1)^2}$ and $N_{\mathrm{cnn}} = \mathcal{O}(\log d)$. We will use the fact that $L = \log_2 d + 2$.

First, we turn to the estimate of $\bar{Q}_\sigma(\mathbf{x}_i)$. To this end, we give a simple lemma to estimate the local Lipschitz constant of our activation functions.

**Lemma G.8.** *If for $|x|, |y| \leqslant M$, we have $|\beta(x) - \beta(y)| \leqslant K|x - y|$, then we will have*

$$\|\beta(\mathbf{x}) - \beta(\mathbf{y})\|_2 \leqslant K\|\mathbf{x} - \mathbf{y}\|_2$$

*for any $\|\mathbf{x}\|_2, \|\mathbf{y}\|_2 \leqslant M$.*

*Proof.* $\|\beta(\mathbf{x}) - \beta(\mathbf{y})\|_2^2 = \sum_i (\beta(x_i) - \beta(y_i))^2 \leqslant \sum_i K^2 (x_i - y_i)^2 = K^2 \|\mathbf{x} - \mathbf{y}\|_2^2$ $\qquad\square$

Our $\sigma_1(\cdot) = \sigma_L(\cdot) = \sigma^2(\cdot)$ activation functions are $2K$ Lipschitz continuous when restricted in $[-K, K]$. So for $Q_1(\mathbf{x}_i)$, since $\|\mathcal{T}_1(\mathbf{x}_i)\|_2 \leqslant \|\mathcal{T}_1\|(\|\mathbf{x}_i\|_2 + 1) \leqslant J(\|\mathbf{x}_i\|_2 + 1)$, we have $Q_1(\mathbf{x}_i) \leqslant 2J(\|\mathbf{x}_i\|_2 + 1)$. For $Q_L(\mathbf{x}_i)$, we similarly have

$$\|\mathcal{T}_L \circ \cdots \circ \sigma_1 \circ \mathcal{T}_1(\mathbf{x}_i)\|_2 \leqslant (1 + J)^{L+1}(\|\mathbf{x}_i\|_2 + 1)^2,$$

so $Q_L(\mathbf{x}_i) \leqslant 2(1+J)^{L+1}(\|\mathbf{x}_i\|_2 + 1)^2$ is straightforward. Thus, $\bar{Q}_\sigma(\mathbf{x}_i) \leqslant 2^{L+2}(1+J)^{L+2}(\|\mathbf{x}_i\|_2 + 1)^3$. Therefore, we will have

$$\gamma(J) = \mathbb{E}[\hat{\gamma}_n(J)] \lesssim J(1+J)^L \mathbb{E}[\hat{M}_n]$$

$$\lesssim J(1+J)^{2L+2} 2^{L+2} \sqrt{\mathbb{E}[(\|\mathbf{x}\|_2 + 1)^8]} \lesssim d^2 J(1+J)^{2L+2} 2^{L+2} \quad (112)$$

where the third step follows from the Jensen's inequality and the fact that $z \mapsto \sqrt{z}$ is concave; the last step uses the higher-order moment bound of sub-exponential random variables (Vershynin, 2018, Proposition 2.7.1).

Recalling we have $\alpha_{\mathcal{H}} = 1$ from equation (53) and applying Proposition B.14, we have *w.p.* at least $1 - 2\delta$

$$\left\| \pi_A \circ h_{\hat{\theta}_n} - h^* \right\|_{L^2(P)}^2 - \epsilon_* \lesssim \frac{\sigma^3 B}{(B - 2A)^2} e^{-\frac{(B-2A)^2}{2\sigma^2}} + \lambda M_* + B^2 \sqrt{\frac{\log(4/\delta)}{n}} + B^2 \sqrt{\frac{N_{\text{cnn}} \log(B\gamma(U_\lambda))}{n}},$$

where

$$\epsilon_* = 0, \qquad M_* \lesssim \log d$$

Taking $A = A_0$, $B = 2A + \sigma\sqrt{\log n}$, $\lambda = 1/\sqrt{n}$ and applying Lemma B.13, we have

$$\left\| \pi_{A_0} \circ h_{\hat{\theta}_n} - h^* \right\|_{L^2(P)}^2 \leqslant \text{poly}(A_0) \left( \frac{\sigma^2}{\sqrt{n}} + \frac{\log d}{\sqrt{n}} + B^2 \sqrt{\frac{\log(1/\delta)}{n}} + B^2 \sqrt{\frac{\log d(\log(B) + \log(\gamma(U_\lambda)))}{n}} \right)$$

where

$$\log(\gamma(U_\lambda)) \lesssim \log(d) \log\left( \frac{1}{2\lambda} \left( \sigma^2 + B^2 \sqrt{\frac{2\log(2/\delta)}{n}} \right) + \log d \right).$$

Simplify the inequality, we have

$$\left\| \pi_{A_0} \circ h_{\hat{\theta}_n} - h^* \right\|_{L^2(P)}^2 \leqslant \text{poly}(A_0, \sigma, \log(1/\delta), \log\log n) \frac{\log n \log d \sqrt{\log n + \log\log d}}{\sqrt{n}}$$

This implies that, for a target accuracy $\epsilon$ and failure probability $\delta$,

$$n \geqslant \text{poly}(\sigma, \log\frac{1}{\delta}, \log\frac{1}{\epsilon}) \epsilon^{-2} \log^2 d (\log\log d)^3$$

is enough. $\quad\square$

# H    LCNs vs. FCNs: Proofs of Section 5.3

## H.1    Proof of Theorem 5.7

Under the assumptions in Theorem 5.7, Lemma 5.6 implies that $\mathbb{A}_T^{\text{fcn}}$ is $G_{\text{ort}}(4d)$-equivalent. Then, by Lemma 5.1, we have

$$\mathcal{C}(\mathbb{A}_T^{\text{fcn}}, \{\bar{h}^*\}, \epsilon) \geqslant \bar{\mathcal{C}}(\{\bar{h}^*\} \circ G_{\text{ort}}(4d), \epsilon).$$

We next use the Fano's method (Proposition B.20) to bound the right hand side, by which the key to obtaining a tight bound is to construct a packing of $\{\bar{h}^*\} \circ G_{\text{ort}}(4d)$ as large as possible. Specifically, we will prove the following lemma.

**Lemma H.1.** *There exist absolute positive constants* $C, c_5, c_6 > 0$ *such that if* $d, A_0 \geqslant C$, *there exists* $h_1, h_2, \ldots, h_M \in \{\bar{h}^*\} \circ O(4d)$ *satisfying that*

$$\sup_{i,j} \|h_i - h_j\|_{L^2(P)}^2 \leqslant c_6, \quad \inf_{i \neq j} \|h_i - h_j\|_{L^2(P)}^2 \geqslant \frac{1}{4} c_5, \quad \text{and} \quad M \geqslant 2^{d(d-1)/2}.$$

Assuming Lemma H.1 is right for now and applying the above packing to Fano's inequality (Proposition B.20), we have

$$\inf_{\mathbb{A} \in \mathcal{A}} \sup_{h^* \in \bar{h}^* \circ O(4d)} \mathbb{E}\left[ \|h_{\mathbb{A}(S_n)} - h^*\|_{L^2(P)}^2 \right] \geqslant A \left( 1 - \frac{n}{2\sigma^2 M^2 \log M} \sum_{i,j=1}^M \|h_i - h_j\|_{L^2(P)}^2 - \frac{\log 2}{\log M} \right)$$

$$\geqslant \frac{1}{4} c_5 \left( 1 - \frac{c_6 n}{\sigma^2 d(d-1) \log 2} - \frac{2}{d(d-1)} \right)$$

Taking $n$ such that the right hand side to satisfy

$$\frac{1}{4} c_5 \left( 1 - \frac{c_6 n}{\sigma^2 d(d-1) \log 2} - \frac{2}{d(d-1)} \right) \leqslant \frac{c_5}{8} =: \epsilon_0$$

gives $n = \Omega(\sigma^2 d^2)$. Thus, we can conclude that $\bar{\mathcal{C}}(\{\bar{h}^*\} \circ G_{\text{ort}}(4d), \epsilon_0) = \Omega(\sigma^2 d^2)$. Otherwise, the minimax error becomes larger than $\epsilon_0$. $\quad\square$

### H.1.1 Proof of Lemma H.1

We now turn to construct a proper packing for $\{\bar{h}^*\} \circ G_{\text{ort}}(4d)$ to satisfy the condition in Lemma H.1. To simplify the statement, define $q : \mathbb{R}^{2d} \mapsto \mathbb{R}$ and $g_U : \mathbb{R}^{2d} \mapsto \mathbb{R}$ for $U \in \mathbb{R}^{d \times d}$ by

$$q(\mathbf{x}) = \sum_{i=1}^{d}(x_{2i-1}^2 - x_{2i}^2) \quad g_U(\mathbf{x}) = \mathbf{x}_{1:d}^T U \mathbf{x}_{d+1:2d}.$$

**Lemma H.2.** *(A restatement of (Li et al., 2020, Lemma D.2)) For any $U \in G_{\text{ort}}(d)$, there exist $\tau_U \in G_{\text{ort}}(2d)$ such that*

$$g_U(\mathbf{x}) = q \circ \tau_U(\mathbf{x}).$$

This lemma implies that for any $U \in G_{\text{ort}}(d)$,

$$\frac{1}{d} \left( \mathbf{x}_{1:d}^T U \mathbf{x}_{d+1:2d} \right) \left( \sum_{i=1}^{d}(x_{2i+2d-1}^2 - x_{2i+2d}^2) \right) = d^{-1}q \circ \tau_U(\mathbf{x}_{1:2d})q(\mathbf{x}_{2d+1:4d})$$

$$= h^* \circ \begin{pmatrix} \tau_U & 0 \\ 0 & I_{2d} \end{pmatrix}(\mathbf{x}) \in \{h^*\} \circ G_{\text{ort}}(4d), \tag{113}$$

where we slightly abuse the notation: using $f(\mathbf{x})$ to denote the function $f(\cdot)$.

Let

$$\mathcal{U}_{A_0} = \left\{ \pi_{A_0} \circ f_U : f_U(\mathbf{x}) = \frac{1}{d} \left( \mathbf{x}_{1:d}^T U \mathbf{x}_{d+1:2d} \right) \left( \sum_{i=1}^{d}(x_{2i+2d-1}^2 - x_{2i+2d}^2) \right), U \in G_{\text{ort}}(d) \right\}. \tag{114}$$

Note that equation (113) implies

$$\mathcal{U}_{A_0} \subset \{\bar{h}^*\} \circ G_{\text{ort}}(4d).$$

Next, the following lemma shows that when the target truncation threshold $A_0$ is large enough, finding a packing of $\left( \mathcal{U}_{A_0}, \| \cdot \|_{L^2(P)} \right)$, thus a packing of $\left( \{\bar{h}^*\} \circ G_{\text{ort}}(4d), \|\cdot\|_{L^2(P)} \right)$, can be reduced to pack the equivariance group $\left( G_{\text{ort}}(d), \| \cdot \|_F / \sqrt{d} \right)$.

**Lemma H.3.** *There exist absolute constants $C, c_3, c_4 > 0$ such that when $d, A_0 \geqslant C$, we have for any $U, U' \in G_{\text{ort}}(d)$ that*

$$\frac{c_3}{d} \|U - U'\|_F^2 \leqslant \|\pi_{A_0} \circ f_U - \pi_{A_0} \circ f_{U'}\|_{L^2(P)}^2 \leqslant \frac{c_4}{d} \|U - U'\|_F^2$$

**Proof idea.** The proof of this lemma is technically lengthy and deferred to Appendix I.2. Note that when $A_0 = +\infty$, i.e., there is no truncation in the target function, a simple calculation leads to

$$\mathbb{E}\left[ (f_U(X) - f_{U'}(X))^2 \right] = \frac{4\|U - U'\|_F^2}{d}$$

When $A_0$ is finite but large enough, by concentration inequalities, we can show that the influence of truncation is negligible, although the proof is lengthy.

**Lemma H.4.** *(Li et al., 2020, Lemma D.4.) For any positive integer $d \geqslant 2$ and any positive $\epsilon \leqslant c_2/2$, we have*

$$\mathcal{P}(G_{\text{ort}}(d), c_1 \|\cdot\|_F / \sqrt{d}, \epsilon) \geqslant \left( \frac{c_2}{\epsilon} \right)^{\frac{d(d-1)}{2}}$$

*where $c_1$ and $c_2$ are two positive absolute constants.*

The above lemma provides a lower bound of the packing number of $\left( G_{\text{ort}}(d), \| \cdot \|_F / \sqrt{d} \right)$. Intuitively speaking, this lemma is true due to the fact that $G_{\text{ort}}(d)$ can be viewed as a $\frac{d(d-1)}{2}$ dimensional compact manifold from a Lie group perspective.

*Proof of Lemma H.1.* First, for any $U, U' \in G_{\text{ort}}(d)$, we have

$$\|U - U'\|_F^2 \leqslant 2(\|U\|_F^2 + \|U'\|_F^2) = 4d.$$

By Lemma H.3, we have

$$\|\pi_{A_0} \circ f_U - \pi_{A_0} \circ f_{U'}\|^2_{L^2(P)} \leqslant \frac{c_4}{d}\|U - U'\|^2_F \leqslant 4c_4. \tag{115}$$

By Lemma H.4 and taking $\epsilon_0 = c_2/2$, we there exist $U_1, \ldots, U_M \in G_{\text{ort}}(d)$ with $M \geqslant 2^{\frac{d(d-1)}{2}}$ such that $\|U_i - U_j\|_F \geqslant \frac{c_2\sqrt{d}}{2c_1}$ for any $i \neq j$. Thus, by Lemma H.3, we have for any $i \neq j$ that

$$\|\pi_{A_0} \circ f_{U_i} - \pi_{A_0} \circ f_{U_j}\|^2_{L^2(P)} \geqslant \frac{c_3}{d}\frac{c_2^2 d}{4c_1^2} =: c_5. \tag{116}$$

Combining (115) and (116), we can conclude that $\{\pi_{A_0} \circ f_{U_j}\}_{j=1}^m$ are a packing that satisfy the condition in Lemma H.1. Thus, we complete the proof. $\square$

## H.2 Proof of Theorem 5.5

Let $\sigma(z) := \text{ReLU}(z)$ in this subsection for notation simplicity. We first recall the architecture of the LCN model: $L = \log_2(4d)$, channel number $C_1 = C_L = 4$, $C_l = 2$ for $l = 2, \ldots, L-1$, and activation functions $\sigma_1(z) = \sigma_L(z) = \sigma^2(z), \sigma_l(z) = \sigma(z)$ for $l = 2, \ldots, L-1$.

*Step I: Representing the target using LCNs.*

**Lemma H.5.** *There is a parametrization $\theta^*$ such that $h^{\text{lcn}}_{\theta^*} = h^*$ and $\|\theta^*\|_{\mathcal{P}} \lesssim d$.*

**Proof.** With the same depth, activation functions, and channel numbers, a LCN can simulate its CNN counterpart. By the proof of Lemma G.7, the exact representation is obvious. The parameter norm bound is due to

$$\|\theta\|_{\mathcal{P}} = \|W_o\| + \sum_{l=1}^{L}(\|W^{(l)}\|_F + \|\mathbf{b}^{(l)}\|_F)$$

$$\lesssim 1 + \sum_{l=1}^{L}(D_l + 0) = 1 + \sum_{l=1}^{\log d+2} \frac{4d}{2^l} \lesssim d.$$

$\square$

*Step II: Estimating the sample complexity.*

Recall the input distribution $P = \mathcal{N}(0, I_{4d})$. Let $\mathcal{H}^{\text{LCN}}_J = \{h_\theta : \|\theta\| \leqslant J\}$, where we recall $\|\cdot\|$ is defined in Appendix C. By Proposition B.14, the covering number satisfies

$$\mathcal{N}(\mathcal{H}^{\text{LCN}}_J, \hat{\rho}_n, t) \leqslant \left(\frac{3\hat{\gamma}_n(J)}{t}\right)^{N_{\text{lcn}}} \tag{117}$$

where $\hat{\gamma}_n(J) = \hat{M}_n J(1+J)^L$, $\hat{M}_n = \sqrt{\frac{1}{n}\sum_{i=1}^{n}(\bar{Q}_\sigma(\mathbf{x}_i))^2(\|\mathbf{x}_i\|_2 + 1)^2}$ and $N_{\text{lcn}} = \mathcal{O}(d)$. Here we use the fact that $L = \log_2 d + 2$. Following the same argument as in Appendix G.2 we have $\bar{Q}_\sigma(\mathbf{x}_i) \leqslant 2^{L+2}(1+J)^{L+2}(\|\mathbf{x}_i\|_2 + 1)^3$. Therefore, we will have

$$\gamma(J) = \mathbb{E}[\hat{\gamma}_n(J)] \lesssim J(1+J)^L \mathbb{E}[\hat{M}_n]$$

$$\overset{(a)}{\lesssim} J(1+J)^{2L+2}2^{L+2}\sqrt{\mathbb{E}[(\|\mathbf{x}\|_2 + 1)^8]} \overset{(b)}{\lesssim} d^2 J(1+J)^{2L+2}2^{L+2} \tag{118}$$

where $(a)$ uses the Jensen's inequality and $(b)$ follows from the higher-order moment bound of sub-exponential random variables (Vershynin, 2018, Proposition 2.7.1).

Recalling we have $\alpha_\mathcal{H} = 1$ from equation (61) and applying Proposition B.14, we have with probability at least $1 - 2\delta$

$$\left\|\pi_A \circ h_{\hat{\theta}_n} - h^*\right\|^2_{L^2(P)} - \epsilon_* \lesssim \frac{\sigma^3 B}{(B-2A)^2}e^{-\frac{(B-2A)^2}{2\sigma^2}} + \lambda M_* + B^2\sqrt{\frac{\log(4/\delta)}{n}} + B^2\sqrt{\frac{N_{\text{lcn}}\log(B\gamma(U_\lambda))}{n}},$$

where

$$\epsilon_* = 0, \qquad M_* \lesssim d$$

Taking $A = A_0$, $B = 2A + \sigma\sqrt{\log n}$, $\lambda = 1/\sqrt{n}$ and applying Lemma B.13, we have

$$\left\|\pi_{A_0} \circ h_{\hat{\theta}_n} - h^*\right\|^2_{L^2(P)} \leqslant \operatorname{poly}(A_0)\left(\frac{\sigma^2}{\sqrt{n}} + \frac{\log d}{\sqrt{n}} + B^2\sqrt{\frac{\log(1/\delta)}{n}} + B^2\sqrt{\frac{d(\log(B) + \log(\gamma(U_\lambda)))}{n}}\right)$$

where

$$\log(\gamma(U_\lambda)) \lesssim \log(d)\log\left(\frac{1}{2\lambda}\left(\sigma^2 + B^2\sqrt{\frac{2\log(2/\delta)}{n}}\right) + d\right).$$

Simplify the inequality, we have

$$\left\|\pi_{A_0} \circ h_{\hat{\theta}_n} - h^*\right\|^2_{L^2(P)} \leqslant \operatorname{poly}(A_0, \sigma, \log(1/\delta), \log\log n)\frac{\log n\sqrt{d\log d}\sqrt{\log n + \log d}}{\sqrt{n}}$$

This implies that, for a target accuracy $\epsilon$ and failure probability $\delta$,

$$n \geqslant \operatorname{poly}(A_0, \sigma, \log\frac{1}{\delta}, \log\frac{1}{\epsilon})\epsilon^{-2}d\log^4 d$$

is enough. $\qquad\square$

# I  Auxiliary Lemmas for Appendix G and H

Recall our input distribution $P = \mathcal{N}(0, I_{4d})$. Recall that target function $\bar{h}^* = \pi_{A_0} \circ h^*$ where

$$h^*(\mathbf{x}) = \frac{1}{d}\left(\sum_{i=1}^{d}(x_{2i-1}^2 - x_{2i}^2)\right)\left(\sum_{i=1}^{d}(x_{2d+2i-1}^2 - x_{2d+2i}^2)\right)$$

In the following, we show that the truncation operator $\pi_{A_0}$ on $h^* \circ U$ for any $U \in G_{\text{ort}}(4d)$ will not matter a lot when $A_0$ is moderately large.

**Lemma I.1.** *Let $X \sim P$. For any $U \in G_{ort}(4d)$, there exists one absolute constant $c_1 > 0$ such that for any $\delta \in (0, 1)$, if $A_0 \geqslant \max\left(\frac{\log(4/\delta)}{c_1}, \frac{\log^2(4/\delta)}{c_1^2 d}\right)$, it holds that*

$$\mathbb{P}\{\pi_{A_0} \circ h^* \circ U(X) = h^* \circ U(X)\} \geqslant 1 - \delta$$

*Proof.* Without loss of generality, we can set $U = I_{4d}$ since $P$ is invariant under $G_{\text{ort}}(4d)$. For simplicity, let $Y_i = X_{2i-1}^2 - X_{2i}^2$ for $i \in [2d]$ in this proof. It is easy to check that $Y_i$ are i.i.d. random variables with $\|Y_i\|_{\psi_1} \lesssim 1$. Then, using the Bernstein's inequality (Lemma B.3) gives

$$\mathbb{P}\left\{\left|\sum_{i=1}^{d}Y_i\right| \geqslant \sqrt{d}t\right\} = \mathbb{P}\left\{\left|\frac{1}{d}\sum_{i=1}^{d}Y_i\right| \geqslant \frac{t}{\sqrt{d}}\right\} \leqslant 2e^{-c_1\min(\sqrt{d}t, t^2)}$$

where $c_1 > 0$ is one absolute constant. Setting the failure probability to be smaller than $\delta/2$: $2e^{-c_1\min(t^2, \sqrt{d}t)} \leqslant \delta/2$ gives $t \geqslant \max\left(\sqrt{\frac{\log(4/\delta)}{c_1}}, \frac{\log(4/\delta)}{c_1\sqrt{d}}\right) =: C_\delta$. This implies that it holds *w.p.* at least $1 - \delta/2$ that

$$\left|\sum_{i=1}^{d}Y_i\right| \leqslant C_\delta\sqrt{d}.$$

Similarly, it holds *w.p.* at least $1 - \delta/2$ that

$$\left|\sum_{i=d+1}^{2d}Y_i\right| \leqslant C_\delta\sqrt{d}.$$

Combining them leads to

$$\mathbb{P}\{\pi_{A_0} \circ h^*(X) = h^*(X)\} = \mathbb{P}\{h^*(X) \leqslant C_\delta^2\} \geqslant 1 - \delta.$$

$\qquad\square$

## I.1 Proof of Lemma G.2

For any $\tau = \text{diag}\left(U_1, U_2, \ldots, U_{2d}\right), \tau' = \text{diag}\left(U_1, U_2', \ldots, U_{2d}'\right) \in G_{\text{semiloc}}$, let

$$\mathcal{I} = \{i \in [2d] : U_i \neq U_i'\}$$

be the set of indices where the local permutations are different for $\tau$ and $\tau'$. Let $s = |\mathcal{I}|$. By the definition of $G_{\text{semiloc}}$, we must have $j \leqslant d$ for any $j \in \mathcal{I}$. Thus, we have

$$h^* \circ \tau(\mathbf{x}) - h^* \circ \tau'(\mathbf{x}) = \frac{2}{d} \left( \sum_{j \in \mathcal{I}} (-1)^{o_j} (x_{2j-1}^2 - x_{2j}^2) \right) \left( \sum_{i=1}^{d} (x_{2d+2i-1}^2 - x_{2d+2i}^2) \right), \quad (119)$$

where $o_j \in \{+1, -1\}$ for $j \in \mathcal{I}$. Because of the symmetry of $P$, we can assume $o_j = 1$ for all $j \in \mathcal{I}$ without loss of generality.

**Upper bound.** First, recall that $X \sim \mathcal{N}(0, I_{4d})$, we have

$$
\begin{aligned}
\left\| \bar{h}^* \circ \tau - \bar{h}^* \circ \tau' \right\|_{L^2(P)}^2 &= \mathbb{E} \left| \pi_{A_0} \circ h^* \circ \tau(X) - \pi_{A_0} \circ h^* \circ \tau'(X) \right|^2 \\
&\leqslant \mathbb{E} \left| h^* \circ \tau(X) - h^* \circ \tau'(X) \right|^2 \\
&= \mathbb{E} \left[ \frac{4}{d^2} \left( \sum_{j \in \mathcal{I}} \left( X_{2j-1}^2 - X_{2j}^2 \right) \right)^2 \left( \sum_{i=1}^{d} \left( X_{2d+2i-1}^2 - X_{2d+2i}^2 \right) \right)^2 \right] \\
&= \frac{4}{d^2} \mathbb{E} \left[ \left( \sum_{j \in \mathcal{I}} (X_{2j-1}^2 - X_{2j}^2) \right)^2 \right] \mathbb{E} \left[ \left( \sum_{i=1}^{d} (X_{2d+2i-1}^2 - X_{2d+2i}^2) \right)^2 \right] \\
&= \frac{4}{d^2} \text{Var} \left( \sum_{j \in \mathcal{I}} (X_{2j-1}^2 - X_{2j}^2) \right) \text{Var} \left( \sum_{i=1}^{d} (X_{2d+2i-1}^2 - X_{2d+2i}^2) \right) \\
&= \frac{4}{d^2} \cdot sd \cdot \left( \text{Var} \left( Y^2 - Z^2 \right) \right)^2 \\
&= \frac{64s}{d} \quad (120)
\end{aligned}
$$

where $Y, Z \sim \mathcal{N}(0, 1)$ are independent random variables.

**Lower Bound.** Denote the events

$$
\begin{aligned}
E_1 &= \{\mathbf{x} : \pi_{A_0} \circ h^* \circ \tau(\mathbf{x}) = h^* \circ \tau(\mathbf{x})\} \\
E_2 &= \{\mathbf{x} : \pi_{A_0} \circ h^* \circ \tau'(\mathbf{x}) = h^* \circ \tau'(\mathbf{x})\}
\end{aligned}
$$

for the lower bound, we have

$$
\begin{aligned}
\|\bar{h}^* \circ \tau - \bar{h}^* \circ \tau'\|_{L^2(P)}^2 &\geqslant \mathbb{E} \left[ 1_{[E_1 \cap E_2]}(X)(h^* \circ \tau(X) - h^* \circ \tau'(X))^2 \right] \\
&= \mathbb{E} \left[ (h^* \circ \tau(X) - h^* \circ \tau'(X))^2 \right] - \mathbb{E} \left[ 1_{(E_1 \cap E_2)^c} (h^* \circ \tau(X) - h^* \circ \tau'(X))^2 \right] \\
&\geqslant \mathbb{E} \left[ (h^* \circ \tau(X) - h^* \circ \tau'(X))^2 \right] \\
&\quad - \sqrt{(1 - \mathbb{P}(E_1 \cap E_2)) \mathbb{E} \left[ (h^* \circ \tau(X) - h^* \circ \tau'(X))^4 \right]} \quad (121)
\end{aligned}
$$

where the last inequality follows from the Cauchy-Schwarz inequality: $(\mathbb{E} \, YZ)^2 \leqslant \mathbb{E} \, Y^2 \, \mathbb{E} \, Z^2$.

For the first term on the RHS of (121), we have from (120) that

$$\mathbb{E} \left[ (h^* \circ \tau(X) - h^* \circ \tau'(X))^2 \right] = \frac{64s}{d} \quad (122)$$

For the second term on the RHS of (121), we have

$$\mathbb{E}\left[(h^* \circ \tau(X) - h^* \circ \tau'(X))^4\right] = \frac{16}{d^4}\mathbb{E}\left[\left(\sum_{j\in\mathcal{I}}(X^2_{2j-1} - X^2_{2j})\right)^4 \left(\sum_{i=1}^{d}(X^2_{2i+2d-1} - X^2_{2i+2d})\right)^4\right]$$

$$= \frac{16}{d^4}\mathbb{E}\left[\left(\sum_{j\in\mathcal{I}}(X^2_{2j-1} - X^2_{2j})\right)^4\right]\mathbb{E}\left[\left(\sum_{i=1}^{d}(X^2_{2i+2d-1} - X^2_{2i+2d})\right)^4\right]$$

(123)

We first compute the first term by expanding it.

$$\mathbb{E}\left[\left(\sum_{j\in\mathcal{I}}(X^2_{2j-1} - X^2_{2j})\right)^4\right] = \sum_{i,j\in\mathcal{I}}\mathbb{E}\left[(X^2_{2j-1} - X^2_{2j})^2(X^2_{2j-1} - X^2_{2i})^2\right]$$

$$\lesssim s^2,$$

where we use the fact: $\mathbb{E}\left[(X^2_{2j-1} - X^2_{2j})^2(X^2_{2i-1} - X^2_{2i})^2\right] = \mathcal{O}(1)$ for all $i, j$. By the same argument, we have

$$\mathbb{E}\left[\left(\sum_{i=1}^{d}(X^2_{2i+2d-1} - X^2_{2i+2d})\right)^4\right] \lesssim d^2$$

(124)

That leads to the following estimate: there exists an absolute constant $c_3 > 0$ such that

$$\mathbb{E}\left[(h^* \circ \tau(X) - h^* \circ \tau'(X))^4\right] \leqslant c_3 \frac{s^2}{d^2}$$

(125)

By Lemma I.1, there is an absolute constant $C$ such that when $A_0 \geqslant C$, we have

$$1 - \mathbb{P}(E_1 \cap E_2) \leqslant \frac{1}{c_3}$$

(126)

Plugging (122), (125), and (126) into (121) gives

$$\left\|\bar{h}^* \circ \tau - \bar{h}^* \circ \tau'\right\|^2_{L^2(P)} \geqslant \frac{64s}{d} - \frac{s}{d} = \frac{63s}{d}$$

(127)

Combining the upper bound and the lower bound, we complete the proof. $\quad\square$

## I.2 Proof of Lemma H.3

For any $U \in O(d)$, let $h_U = \pi_{A_0} \circ f_U$ with

$$f_U(\mathbf{x}) = \frac{1}{d}\left(\mathbf{x}_{1:d}^\top U \mathbf{x}_{d+1:2d}\right)\left(\sum_{i=1}^{d}(x^2_{2d+2i-1} - x^2_{2d+2i})\right).$$

Thus, we have $h_U \in \{\bar{h}^*\} \circ G_{\mathrm{ort}}(4d)$ due to Lemma H.2.

**Upper Bound.** We have

$$\|h_U - h_{U'}\|^2_{L^2(P)} = \mathbb{E}\left[(\pi_{A_0} \circ f_U(X) - \pi_{A_0} \circ f_{U'}(X))^2\right]$$

$$\leqslant \mathbb{E}\left[(f_U(X) - f_{U'}(X))^2\right]$$

$$= \frac{1}{d^2}\mathbb{E}\left[\left(X_{1:d}^\top(U - U')X_{d+1:2d}\right)^2\left(\sum_{i=1}^{d}(X^2_{2d+2i-1} - X^2_{2d+2i})\right)^2\right]$$

$$= \frac{1}{d^2}\mathbb{E}\left[\left(X_{1:d}^\top(U - U')X_{d+1:2d}\right)^2\right]\mathbb{E}\left[\left(\sum_{i=1}^{d}(X^2_{2d+2i-1} - X^2_{2d+2i})\right)^2\right],$$

(128)

where the last step follows from the independence between $X_{1:2d}$ and $X_{2d+1:4d}$.

Note that by equation (120),

$$\mathbb{E}\left[\left(\sum_{i=1}^{d}(X_{2d+2i-1}^2 - X_{2d+2i}^2)\right)^2\right] = 4d \tag{129}$$

And, for the first term, we have

$$
\begin{aligned}
\mathbb{E}\left[\left(X_{1:d}^\top(U - U')X_{d+1:2d}\right)^2\right] &= \mathbb{E}\left[\mathbb{E}\left[\left(X_{1:d}^\top(U - U')X_{d+1:2d}\right)^2 | X_{d+1:2d}\right]\right] \\
&= \mathbb{E}\left[\mathbb{E}\left[X_{d+1:2d}^\top(U - U')^\top X_{1:d}X_{1:d}^\top(U - U')X_{d+1:2d}|X_{d+1:2d}\right]\right] \\
&= \mathbb{E}\left[X_{d+1:2d}^\top(U - U')^\top(U - U')X_{d+1:2d}\right] \\
&= \mathrm{Tr}\left((U - U')^\top(U - U')\right) \\
&= \|U - U'\|_F^2
\end{aligned}
\tag{130}
$$

Plugging (129) and (130) into (128) gives

$$\|h_U - h_{U'}\|_{L^2(P)}^2 \leqslant \frac{4\|U - U'\|_F^2}{d}$$

**Lower Bound.** Define events

$$
\begin{aligned}
E_1 &= \{\mathbf{x} : \pi_{A_0} \circ f_U(\mathbf{x}) = f_U(\mathbf{x})\} \\
E_2 &= \{\mathbf{x} : \pi_{A_0} \circ f_{U'}(\mathbf{x}) = f_{U'}(\mathbf{x})\}.
\end{aligned}
$$

Then, we have

$$
\begin{aligned}
\|h_U - h_{U'}\|_{L^2(P)}^2 &\geqslant \mathbb{E}\left[1_{[E_1 \cap E_2]}(f_U(X) - f_{U'}(X))^2\right] \\
&= \mathbb{E}\left[(f_U(X) - f_{U'}(X))^2\right] - \mathbb{E}\left[1_{(E_1 \cap E_2)^c}(f_U(X) - f_{U'}(X))^2\right] \\
&\geqslant \mathbb{E}\left[(f_U(X) - f_{U'}(X))^2\right] - \sqrt{(1 - \mathbb{P}(E_1 \cap E_2))\mathbb{E}\left[(f_U(X) - f_{U'}(X))^4\right]}
\end{aligned}
\tag{131}
$$

where the last step uses the Cauchy-Schwartz inequality: $(\mathbb{E}[YZ])^2 \leqslant \mathbb{E}Y^2 \mathbb{E}Z^2$. We next bound the two terms of the right hand side separately.

For the second term, we observe that

$$
\begin{aligned}
\mathbb{E}\left[(f_U(X) - f_{U'}(X))^4\right] &= \frac{1}{d^4}\mathbb{E}\left[\left(X_{1:d}^\top(U - U')X_{d+1:2d}\right)^4\left(\sum_{i=1}^{d}(X_{2d+2i-1}^2 - X_{2d+2i}^2)\right)^4\right] \\
&= \frac{1}{d^4}\mathbb{E}\left[\left(X_{1:d}^\top(U - U')X_{d+1:2d}\right)^4\right]\mathbb{E}\left[\left(\sum_{i=1}^{d}(X_{2d+2i-1}^2 - X_{2d}^2)\right)^4\right] \\
&\lesssim \frac{1}{d^2}\mathbb{E}\left[\left(X_{1:d}^\top(U - U')X_{d+1:2d}\right)^4\right]
\end{aligned}
\tag{132}
$$

where the last step follows from (124). Note that conditioned on $X_{d+1:2d}$,

$$X_{1:d}^\top(U - U')X_{d+1:2d} \sim \mathcal{N}(0, \|(U - U')X_{d+1:2d}\|_2^2).$$

Combining with the fact that $EY^4 = 3\sigma^2$ for $Y \sim \mathcal{N}(0, \sigma^2)$, we have

$$\mathbb{E}\left[\left(X_{1:d}^\top(U - U')X_{d+1:2d}\right)^4\right] = 3\mathbb{E}\left[\|(U - U')X_{d+1:2d}\|_2^4\right]$$

Let $B = U - U'$, $Z \sim N(0, I_d)$, and $B^\top B = D$. Then,

$$
\begin{aligned}
\mathbb{E}\left[\|(U - U')X_{d+1:2d}\|_2^4\right] &= \mathbb{E}\left[(Z^\top D Z)^2\right] \\
&= \mathbb{E}\left[\left(\sum_{i,j} D_{i,j} Z_i Z_j\right)^2\right] \\
&= \sum_{i,j} D_{i,j}^2 \, \mathbb{E}\left[Z_i^2 Z_j^2\right] \\
&\lesssim \|D\|_F^2 \leqslant \|B\|_F^4 = \|U - U'\|_F^4
\end{aligned}
\tag{133}
$$

Plugging (133) into (132) gives

$$
\mathbb{E}\left[(f_U(X) - f_{U'}(X))^4\right] \leqslant c_4 \frac{\|U - U'\|_F^4}{d^2}
\tag{134}
$$

for some absolute constant $c_4 > 0$. By Lemma I.1, there is an absolute constant $C$ such that when $A_0 \geqslant C$, we have

$$
1 - \mathbb{P}(E_1 \cap E_2) \leqslant \frac{1}{c_4}
\tag{135}
$$

Continuing the equation (131), we have

$$
\begin{aligned}
\|h_U - h_{U'}\|_{L^2(P)}^2 &\geqslant \mathbb{E}\left[(f_U(X) - f_{U'}(X))^2\right] - \sqrt{(1 - \mathbb{P}(E_1 \cap E_2))\,\mathbb{E}\left[(f_U(X) - f_{U'}(X))^4\right]} \\
&\geqslant \mathbb{E}\left[(f_U(X) - f_{U'}(X))^2\right] - \sqrt{\frac{1}{c_4} \cdot c_4 \frac{\|U - U'\|_F^4}{d^2}} \\
&= \frac{4}{d}\|U - U'\|_F^2 - \frac{\|U - U'\|_F^2}{d} \\
&= \frac{3}{d}\|U - U'\|_F^2,
\end{aligned}
\tag{136}
$$

where in the third step we use the result from the above upper bound:

$$
\mathbb{E}\left[(f_U(X) - f_{U'}(X))^2\right] = \frac{4\|U - U'\|_F^2}{d}.
$$

Combining the upper bound and the lower bound, we complete the proof. $\quad\square$

