# OpenReview forum: "Theoretical Analysis of the Inductive Biases in Deep Convolutional Networks"
_NeurIPS.cc/2023/Conference — NeurIPS 2023 poster_

### Official Review · Reviewer_o2NN · 2023-06-29

**Soundness:** 4 excellent
**Presentation:** 4 excellent
**Contribution:** 2 fair
**Rating:** 6
**Confidence:** 3

**Summary:**

The paper theoretically analyses the sample complexity of CNNs as compared to fully connected networks. The authors find that while fully connected networks require quadratic samples with input dimension, the local connections of CNNs reduce the sample complexity to linear, while weight sharing further reduces sample complexity to logarithmic.

**Strengths:**

**Originality**

As the authors point out, separating the impact of local connections vs weight sharing in CNNs is not a new idea; nevertheless, as far as I am aware, no previous works have shown theoretical sample complexity results for CNNs vs LCNs vs FCNs.

**Quality**

Technically speaking, the paper appears solid. The results and proof look correct as far as I can tell. The appendix is quite comprehensive.

**Clarity**

The paper is generally well-written. The notation is appropriately introduced and concepts are adequately explained.

**Significance**

I think this paper may be significant to the specific subfield that studies the sample complexity of specific model architectures.

**Weaknesses:**

Overall, I'm unfortunately not too convinced about the significance of this work given that there is already work analyzing at least empirically the sample complexity benefit of weight sharing vs locality (Xiao & Pennington 2022 as the authors point out, but also Poggio et al. 2016). Certainly, it seems that the theoretical result proved by the authors is new. On the other hand, it would benefit the paper to validate these theoretical results with experimental evidence, or at least point to prior experiments that line up with the authors' theory.


Minor comment: Using similar-looking notation for orthogonal groups and big O is not ideal.

**Questions:**

Is this the first work to provide theoretical sample complexity bounds for CNNs vs LCNs vs FCNs?

Are the asymptotic sample complexities found by the authors practically relevant? Can the authors empirically demonstrate these trends?

**Limitations:**

Limitations are adequately addressed; the authors clearly state the assumptions of their theory. No negative potential societal impacts.

---

> ### Author Rebuttal · Authors · 2023-08-10
>
> We thank the reviewer for the encouraging and constructive comments. In the following, we address the specific questions.
>
> **1. Experimental evidences** [W1, Q2]
>
> First, we have conducted  numerical experiments and the results align well with our theoretical predictions. We refer to the **Global Response** and the attached **one-page PDF** for more details.
>
> Secondly, though there are empirical works analyzing the sample complexity benefits by weight sharing and locality, but rigorous theoretical analysis is lacking. Our work fills this gap to some extent.
>
> In addition, we shall emphasize that the theoretical findings presented in Sections 3 and 4 are also novel and important for understanding deep CNNs:
>  - In Section 3, we originally establish the universality of deep CNNs with a depth of $L=O(\log d)$. Our proof reveals a mechanism of how the synergy between multichanneling and downsampling is crucial for achieving universality.
> -  In Section 4, we demonstrate, **for the first time**, that deep CNNs can efficiently capture long-range sparse interactions. This claim is further corroborated by supplementary experiments provided in the Global Response. Such findings prompt a rethinking of the common assertion that CNNs struggle to capture long-range interactions.
>
>
>
> **2. Novelty.** [Q1]
>
> To the best of our knowledge, our work does provide the first theoretical sample complexity separation result comparing CNNs, LCNs, and FCNs.

---

> > ### Comment · Reviewer_35ph · 2023-08-13
> >
> > I think the authors should have been more precise in their response about the novelty. Although working in the kernel regime, Favero et al. (2021), Misiaciewicz and Mei (2022) and Cagnetta et al. (2023) all provide sample complexity separation results between CNNs, LCNs and FCNs (the kernel result applies to infinitely-wide neural networks too via the NTK equivalence). Another example is Karp et al. *Local Signal Adaptivity: Provable Feature Learning in Neural Networks Beyond Kernels* (2021), there could be more that I am not aware of.

---

> > > ### Author Response · Authors · 2023-08-16
> > > **We will add discussions on these related works**
> > >
> > > Thank you for your comment and indeed, our comparison with existing results was not precise. In particular, we have set aside those studies that examine the separation of CNNs vs. LCNs vs. FCNs in the **kernel regime**, as our analysis is specifically focused on the sample complexity separation **in the adaptive regime**. In retrospect, we recognize that this omission may be perceived as unfair, and we commit to including discussions on these related works in our revised manuscript.

---

### Official Review · Reviewer_eXvS · 2023-07-06

**Soundness:** 3 good
**Presentation:** 3 good
**Contribution:** 3 good
**Rating:** 7
**Confidence:** 5

**Summary:**

The authors study inductive biases in deep CNNs. They analyse approximation abilities of these networks and compare the learning performances of CNNs for regression with those of locally-connected networks (LCNs) and fully-connected networks. They show that CNNs of depth O(\log d) is sufficient for achieving universality. They also present some analysis for CNNs in approximating functions with sparse variable structures.


**Strengths:**

The result that CNNs of depth O(\log d) with exponentially increasing channels is sufficient for achieving universality is very interesting. It shows the power of CNNs and channels in approximating functions. The example presented in Proposition 3.2 about the lower bound for the universality is nice. The comparison between CNNs and LCNs on learning with bounds O(\log^2 d) and \Omega (d) is simulating. Though the methods used by the authors are not new, the obtained results are valuable and insightful and can be used to explain the efficiency of deep learning algorithms induced by structures of deep neural networks.

Analysing structured deep neural networks in approximating and learning functions with sparse variable structures is an important topic in deep learning theory.

**Weaknesses:**

The topic of analysing structured deep neural networks in approximating and learning functions with sparse variable structures has been studied in the community of deep learning theory. the related literature should be mentioned in the paper. The work of Mhaskar and Poggio (Analysis and Applications 2016) with fully connected neural networks is based on a known sparse variable structure while the recent work of Mao, Shi, and Zhou (Analysis and Applications 2023) with CNNs is for unknown sparse variable structures. Another paper of Chui, Lin, Zhang, and Zhou (IEEE NNLS 2020) is about learning of spatially sparse functions.

**Questions:**

The number of channels for CNNs is often large, but it does not increase exponentially. Can the authors derive a universality result with polynomially increasing channels?

**Limitations:**

The CNNs with exponentially increasing channels might be restrictive.

---

> ### Author Rebuttal · Authors · 2023-08-10
>
> We thank the reviewer for the encouraging and constructive comments. In the following, we address the specific questions.
>
> **1. Related Works.** [W1]
>
> We thank you for pointing out these references.  [Mao et al., 2023] considered various kinds of spatially sparse functions. [Chui et al., 2019]  used deep ReLU FCNs, not deep CNNs. [Poggio et al., 2017]   focuses on studying how deep locally-connected networks can learn functions with similar hierarchical structures efficiently, while shallow fully-connected networks cannot. These works are definitely related, though the settings and analyses are different from ours. In the next version of our paper, we will carefully discuss these related works
>
>  **2. Channel Numbers.** [Q1]
>
> Indeed, our current proof requires an exponential growth of channel numbers. However, we do not view this as a limitation since with only $L=\log d$ layers, the number of channels and the total number of parameters of all except the last layer are only $O(d)$, scaling linearly with input dimension.

---

### Official Review · Reviewer_mzq4 · 2023-07-06

**Soundness:** 2 fair
**Presentation:** 2 fair
**Contribution:** 2 fair
**Rating:** 5
**Confidence:** 3

**Summary:**

This paper provides a fundamental analysis of the superior performance of CNNs, specifically focusing on their downsampling and multichanneling capabilities. Moreover, it establishes evidence with theoretical guarantees that CNNs exhibit efficient learning of sparse functions compared to FCNs, attributing this efficiency to the weight sharing and locality properties inherent in CNNs.

**Strengths:**

This paper presents a comprehensive and rigorous theoretical analysis of the intrinsic inductive biases linked to multichanneling, downsampling, weight sharing, and locality within CNNs. It introduces a new theoretical framework that characterizes the crucial role of multichanneling and downsampling in empowering deep CNNs to efficiently capture long-range dependencies within the target function, yielding valuable insights into their operational mechanisms. Furthermore, leveraging these theoretical insights, the paper substantiates the superior performance of CNNs compared to FCNs, providing further support for their efficacy.

**Weaknesses:**

1. This paper primarily concentrates on exploring the expressive power of Convolutional Neural Networks (CNNs) and argues that a CNN with a depth on the order of $\log d$ possesses the capacity to learn spatial information within the data. However, it lacks the convergence and generalization analysis, i.e., whether CNNs with a layer depth in the order of $\log d$ can effectively learn spatial information using SGD-based algorithms remains uncertain.

2. This paper lacks empirical experiments, particularly those using real data, to provide concrete evidence in support of its theoretical findings.

3. Recent observations suggest that Transformers are more efficient in learning spatial information compared to CNNs, and CNNs are not as widely utilized in current practices. In light of these trends, it would be valuable for the paper to offer insights or considerations pertaining to the Transformer model, further exploring its strengths and potential advantages in learning spatial information.

**Questions:**

1. In lines 158-159, the authors assert that "universality can be established by simply showing that ... can simulate any two-layer ReLU networks." It raises the question of whether the authors assume that a two-layer ReLU network can approximate any continuous function.

2. Could you please provide more context or specify the notations in Assumption 1 you are referring to? This will help in understanding Assumption 1 and the difficulties in following the notations.

3.  It would be better to provide a summary of the technical novelty of this paper.

**Limitations:**

Yes

---

> ### Author Rebuttal · Authors · 2023-08-10
>
> We thank the reviewer for the comments and questions. The following is our response.
>
> **1. Weakness.**
>
> -  **Lack generalization analysis.**  We  already provided generalization analysis in Section 4 for learning sparse functions and Section 5 for disentangling the inductive biases of weight sharing and locality.
> - **Lack optimization analysis.** In our view, both approximation and generalization analyses offer significant insights into the understanding of deep CNNs. The analysis of optimization presents greater challenges due to the non-convex nature of neural networks. At present, even for basic two-layer neural networks, the optimization results are notably limited, let alone for deep CNNs.
> - **Lack empirical experiments.** We have carried out experiments to validate our theory, and the results can be found in the attached **one-page PDF** file within the global response. These experimental results align with our theoretical predictions. For more details, please refer to the **Global  Response**.
> - **Transformers.**
> We respectfully disagree with the assertion that ``transformers are more efficient at learning spatial information compared to CNNs and that CNNs are not prevalent in current practices''. Firstly, while vision transformers (ViTs) might outperform CNNs when vast amounts of data are available, CNNs tend to perform better in limited data scenarios. This implies that ViTs are not necessarily more efficient, as they require more data for optimal performance. Secondly, in practical applications, CNNs remain widely adopted because of their efficiency with smaller datasets. Even in situations with extensive data, U-net styled CNNs continue to be the dominant choice in diffusion models for image generalization. However, we concur that comprehending why ViTs can better capture spatial information in the large data regime is a crucial question, which we leave to future work.
>
>
>
>
>
>
>  **2. Questions.**
>
> - (Q1)  That two-layer ReLU network can approximate any continuous functions has been proved in e.g.,  [Leshno et al., 1993]..
>
> -  (Q2) **Notations in Assumption 1.** The notations are defined in Subsection 2.1 and the weight matrix $W\in \mathbb{R}^{C_1\times C_0\times 4d}$, where  $C_1,C_0$ represent the output and input channels, respectively. The term $4d$ denotes the input dimensions. Note that in this paper, $C_0$ is set to be $1$.
> As such,  $W_{j,1,k}$ denotes the weight  connecting the $k$-th coordinate to the $j$-th output channel.
> Our assumption regarding initialization is mild and is satisfied by most commonly-used initializations: Initializing each element in $W$ independently from the same distribution.
>
> -  (Q3) **Technical Novelty.** We will add a paragraph to summarize the technical novelty in the next version of our paper. Briefly speaking, it can be summarized as follows
>    - **Efficient variable selection.** We are the first to demonstrate that deep CNNs are capable of executing variable selection using merely $\log d$ parameters. This  result is derived through a novel *binary decomposition* of indices.
>    - **Tighter covering number bounds.** To the best of our knowledge, our covering number bound for deep CNNs is tighter than previous works like [Long and Sedghi, 2020].
>     - **Fano's method for random estimators.** The lower bounds for LCNs and FCNs are established using  Fano's method from minimax theory [Wainwright, 2019, Section 15]. However, different from the traditional statistical setup where the estimator is deterministic, the estimators produced by stochastic optimizers  are random. To address this issue, *we develop a variant of Fano's method for random estimators*, which might be of independent interest.
> Further details can be found in  Appendix B.3.o original.
>
>
> **Reference**
>
> Leshno, M., Lin, V. Y., Pinkus, A., and Schocken, S. (1993). Multilayer feedforward networks with a nonpolynomial activation function can approximate any function. Neural networks, 6(6):861–867.

---

> > ### Comment · Reviewer_mzq4 · 2023-08-10
> > **Requires additional clarification**
> >
> > Thank the authors for their comprehensive responses. However, I am eager to deepen my understanding of this work, particularly regarding the theoretical contributions. I am under the assumption that the reference to the generalization analysis pertains to Theorem 4.6 (and please correct me if I am mistaken).
> >
> > Initially, I must acknowledge my time constraints, which unfortunately hinder my ability to thoroughly review the proof details. In light of this, I would greatly appreciate a high-level overview of the theoretical proof as provided by the authors. From my current grasp, the crux of the matter involves demonstrating that the acquired weights exhibit some form of boundedness. This, in turn, leads to a bounded covering number, thus making a substantial contribution to achieving bounded generalization.
> >
> > From my perspective, the key to proving Theorem 4.6 seems to rest in Proposition B.14. It appears that subsequent components naturally flow from the foundation laid by Proposition B.14. Nevertheless, I am keen on understanding if the authors encountered additional challenges beyond Proposition B.14 on their journey towards establishing Theorem 4.6.
> >
> > It is unclear for us to find the global optimal solution $\hat{\theta}_n$ for equation (25). This lack of convergence analysis impairs the strength of the generalization analysis, as the absence of a guarantee in obtaining such an optimal solution raises doubts about the generalizability of $\hat{\theta}_n$. Furthermore, I am interested in learning about the difficulties that the authors faced while proving Proposition B.14, under the assumption that $\hat{\theta}_n$ has already been identified. For me, the most formidable obstacle seems to be the process of locating the optimal solution.
> >
> > Lastly, the theoretical sample complexity's practical applicability is brought into question by the absence of a global optimal solution. The claim that only $\log d$ samples are required to achieve the desired accuracy appears almost too good to be true. As far as my knowledge extends, I am unaware of any real-world experiments that have managed to employ $\log d$ samples to attain the stated level of accuracy. I would greatly appreciate the authors elaborating further on this point, especially by providing insights from real data experiments to help clarify this assertion.

---

> > > ### Author Response · Authors · 2023-08-12
> > > **Additional Clarification**
> > >
> > > We are truly grateful for the reivewer's prompt feedback. Below, we response to the specific points raised.
> > >
> > > **Deepen my understanding of this work, particularly regarding the theoretical contributions**
> > >
> > >
> > > **Response:** We would like to use this opportunity to provide a succinct summary of our theoretical contributions:
> > >
> > > -  We prove that deep CNNs of depth only $L=O(\log d)$ are universal.  This part of analysis reveals a mechanism of how the synergy between multichanneling and downsampling is crucial and shed light on the architectural choice in practice: decisions regarding the number of channels during spatial dimension reduction through pooling or stride.
> > > - We prove that deep CNNs can learn sparse functions efficiently regardless if they are short or long range. Such findings prompt a rethinking of the common assertion that CNNs struggle to capture long-range interactions.
> > > - We establish generalization separations for FCNs vs. LCNs vs. CNNs, which provide theoretical evidences on the distinct roles of locality and weight sharing.
> > >
> > > It is worth noting that the architectrure of CNNs we considered align closely with those used in real-world applications. In contrast, most previous works consider CNNs with architecures very different from  practical ones.
> > > We also emphasize that the core value of our study is deepening our theoretical understanding on CNNs---one of the most important architectures in deep learning--instead of developing new proof techniques.
> > >
> > > **I am under the assumption that the reference to the generalization analysis pertains to Theorem 4.6 (and please correct me if I am mistaken)**
> > >
> > > **Response:**
> > > We wish to highlight that Section 5 also delves into  generalization analysis but for learning a particular task. That task is designed to understand the distinct roles of weight sharing and locality within deep CNNs. Specifically, we have established both **upper and lower bounds for the  sample complexity** for CNNs, LCNs, and FCNs:
> > >
> > > - The upper bounds were derived using Proposition B.14, analogous to the approach used for the upper bounds in Theorem 4.6.
> > > - The lower bounds were established through a unique blend of the *symmetry group perspective* and *minimax theory*. To achieve this, we have made rectifications on the classical Fano's method to make it applicable to random estimators, which is common in machine learning. Notably, these lower bounds are tailored for this explicit task and are *not the frequently encountered worst-case lower bounds*.
> > >
> > >
> > > These generalization bounds, both upper and lower,  provide a rigorous formulation of  the insight behind the symmetry group perpsective for explaining the distinction roles between locality and weight sharing on the efficacy of deep CNNs.
> > >
> > >
> > > **Nevertheless, I am keen on understanding if the authors encountered additional challenges beyond Proposition B.14 on their journey towards establishing Theorem 4.6.**
> > >
> > > **Response:**
> > > The proof of Theorem 4.6 consists of three steps:
> > > - *Step 1: Well-controlled Approximation:* We need to show that sparse functions can be well approximated by deep CNNs with only $O(\log d)$ parameters and well-controlled norms. This part of proof is through a novel binary representation of the indices of relevant coordinates.
> > > - *Step 2: Bounding the covering number:* In this part, we provide a tighter bound on the covering numbers than existing ones like \cite{long2019generalization}, which improves the exponential dependence on parameter norms to polynomial dependence, which is critical for obtaining $O(\log^2 d)$ sample complexity. See the Appendix B.3 for a more comprehensive explanation.
> > > - *Step 3:* Plugging the above approximation result and covering number bound into Proposition B.14.
> > >
> > >
> > >
> > >
> > >
> > > We highlight that the first two steps are novel and more important than step 3, which is somehow standard obtained by  tailoring classical Dudely entropy integral bound into our setup.
> > >
> > > **This lack of convergence analysis impairs the strength of the generalization analysis, as the absence of a guarantee in obtaining such an optimal solution raises doubts about the generalizability of $\hat{\theta}_n$**
> > >
> > > **Response:**
> > >
> > > - On the one hand,  our experimental results provided in the Global Response do show that commonly-used optimizers can find $\hat{\theta}_n$ efficiently for **both short- and long-range sparse functions**.
> > > As for the generalization ability of $\hat{\theta}_n$, it is **guaranteed** by Theorem 4.6.
> > >
> > > - On the other hand, we do agree that lacking convergence analysis is a limitation, which is partially mitigated by our experimental results mentioned above. However, we would like to emphasize again that in the setup as general and complex as ours (deep CNNs in adaptive regime), theoretical analysis of convergence and time complexity is exceedingly challenging, as far as our current understanding permits.

---

> > > > ### Comment · Reviewer_mzq4 · 2023-08-20
> > > >
> > > > Thanks for providing the detailed response. While I still believe that the lack of convergence analysis impairs the strength of the generalization analysis, the novelty described in steps 1 & 2 seems to be a solid contribution, although I did not have enough time to check the full proof yet. I suggest the author highlight the technical challenges and the methods for solving such challenges in the main text. Since most of my concerns have been addressed, I am increasing my score from 3 to 5.

---

> > > > > ### Author Response · Authors · 2023-08-21
> > > > > **Thanks for increasing the score**
> > > > >
> > > > > We are grateful to the reviewer for increasing the score. We will certainly incorporate the suggestion in our revised manuscript.
> > > > >
> > > > > Best,
> > > > >
> > > > > The Authors

---

> > > ### Author Response · Authors · 2023-08-12
> > > **Further additional clarification**
> > >
> > >
> > > **I am interested in learning about the difficulties that the authors faced while proving Proposition B.14, under the assumption that $\hat{\theta}_n$  has already been identified.**
> > >
> > > **Response:**
> > >
> > > The primary technical difficulties in proving Theorem 4.6 lie in establishing the approximation result and bounding the covering number for deep CNNs, which constitute the more novel and technical contributions. In contrast, Proposition B.14 involves more standard generalization bounds. Nonetheless, we emphasize again that the main focus of this work is to elucidate the inductive biases of deep CNNs in the adaptive regime, providing insights into their bias and properties. While we do make technical contributions, our primary aim is not to advance the toolboxs, but rather to leverage theoretical tools to obtain insights into deep learning.
> > >
> > >
> > > **The claim that only
> > >  samples $\log d$ are required to achieve the desired accuracy appears almost too good to be true. As far as my knowledge extends, I am unaware of any real-world experiments that have managed to employ $\log d$ samples to attain the stated level of accuracy. I would greatly appreciate the authors elaborating further on this point, especially by providing insights from real data experiments to help clarify this assertion.**
> > >
> > >
> > >
> > > **Response:**
> > >
> > >
> > > - **``Too good to be true''.** In Section 4, the functions under consideration are {\bf sparse}, for which  learning requires only $O(\log d)$ samples---a well-known fact in statistical and machine learning theory. For instance, sparse linear regression problems like Lasso and compressive sensing, the $O(\log d)$ sample bound can be achieved through {\it explicit sparsity regularization}, e.g., $\ell_1$ norm. In contrast, in our analysis of deep CNNs and the experiments in the global response, there's no explicit sparsity regularization involved. Instead, the sparsity bias emerges inherently from the architectural bias of deep CNNs to help us get $O(\log^2 d)$ sample complexity upper bound.
> > >
> > > - **``unaware of any real-world experiments . . . $\log d$ sample''.**
> > >
> > > Firstly, In real-world scenarios, for complex problems, label functions tend to be much more complicated, often intertwining with various structures. A possible representation might be $f^*=f_1^*+f_2^*+f_3^*+\cdots$, where $f_1^*$ exhibits sparse structure, $f_2^*$ embodies the structures discussed in Section 5, and $f_3^*$ exhibits characteristics we've yet to define. Given this complexity, it's reasonable to expect that in practice, $\operatorname{polylog} (d)$ samples may not always suffice for effective learning, and hence, there isn't any contradiction.
> > >
> > > However, our theoretical results in Section 4  can shed light on the partial efficiency of deep CNNs, especially when the true label functions harbor sparse attributes. This isn't an unfounded assumption. For instance, in image classifications, an object's label predominantly stems from its outline, which represents a minor fraction of the total pixels in an image, signifying sparsity.
> > >
> > > We hope you will increase the score after the clarification. Any further question is welcomed.

---

### Official Review · Reviewer_35ph · 2023-07-06

**Soundness:** 3 good
**Presentation:** 3 good
**Contribution:** 3 good
**Rating:** 7
**Confidence:** 4

**Summary:**

This paper studies the inductive bias of deep convolutional neural networks (CNNs) in three distinct ways. With $d$ the input dimensions, it is shown that (i) depth $O(\log{d})$ is sufficient to achieve universality, i.e. the capacity to approximate any function of $d$ variables, when there is downsampling between layers; (ii) all sparse functions can be learned with $O(\log^2{d})$ samples, close to the optimal $O(\log{d})$; (iii) both weight sharing and locality of the connection are crucial for achieving the nearly-optimal sample complexity.

**Strengths:**

Our theoretical understanding of deep CNNs is still very limited and this paper has the potential to provide a solid contribution to the topic. The explanation of how multichannelling and downsampling conspire to achieve universality with $O(\log{d})$ layers is nice and intuitive. The result of Theorem 4.6 on the sample complexity of learning sparse functions is novel and interesting. The separation results start from a known idea  (using the symmetry of an algorithm to unveil its limitations) but give significantly new insights into the separate roles of weight sharing and locality.

**Weaknesses:**

Overall, I believe this paper would be a valid contribution to the community but there are some issues that I would like to see addressed before recommending publication.

I like section 3 and I believe it is a valuable addition to the literature, but I would downplay the claim that all previous results required a depth of $\log{d}$, as currently written in the abstract. The idea that sufficiently deep CNNs achieve the same expressivity as fully-connected networks (hence they are universal) has been there for a while and the difference between $d$ and $\log{d}$ is solely due to the amount of stride included in the architecture. The case considered in this paper, where the stride equals the filter size, is commonly referred to as *nonoverlapping patches*---the authors should say it explicitly at least in section 2.1---and it has been studied in several other works, including some cited in this paper such as  Cagnetta et al. (2022) and some that are not cited but should be such as Poggio et al. (2016), *'Why and When Can Deep – but Not Shallow – Networks Avoid the Curse of Dimensionality: a Review'*. Both these works imply that deep CNNs with nonoverlapping patches and depth $\log{d}$, so that the post-activations of the last hidden layer look at the whole input, are universal.

There is also something about theorem 4.6 that I think requires further explanation. As far as I understand, theorem 4.6 uses lemma 4.4 together with the generalisation properties of one-hidden layer neural networks (theorem 4 from E et al. (2022)). If this is true, then I do not understand why one needs a deep CNN to achieve sample complexity $O(\log^2{d})$: the result of lemma 4.4, i.e. that there exist a set of parameters such that the $L$-th layer outputs the right-hand side of Eq. (4), is also applicable to the post-activations of a  one-hidden-layer fully-connected network. I should remark that my concern might be due to a misunderstanding of the proof idea, but if that is the case I will be happy to raise mi mark after rebuttal.

Finally, the symmetry-based lowe bounds like those of theorems 5.4 and 5.7 can usually be explained very well at an intuitive level. I think not having such intuitive explanations is a weakness of this manuscript.

**Questions:**

1. Regarding the difference between CNNs and LCNs, Bietti et al. (2021) provides a very simple explanation of why there should be a factor $d$ of difference between the sample complexity of a LCN and that of a CNN. Is their argument valid also in the setting of section 5?

2. Is there a simple explanation based on symmetry as to why the sample complexity of FCNs presents another factor of $d$ with respect to that of LCNs?

The following are not questions but suggestions:

3. There is a typo in line 12 of the abstract (separete instead of separate).

4. In the same line of the abstract I would use a different word than `introduce', since locally-connected networks have been introduced before.

5. Could you please quickly recall the meaning of the various big-O notations in section 1.2?

6. State explicitly that this work considers CNNs with nonoverlapping patches at all layers. e.g. in section 2.1.

7. The related works section needs further references, for instance, the aforementioned Poggio et al. (2016), *'Why and When Can Deep – but Not Shallow – Networks Avoid the Curse of Dimensionality: a Review'* and other works of the same group. Schmidt-Hierber (2021), *'Nonparametric regression using deep neural networks with ReLU activation function'* is also relevant. Furthermore, Bietti (2021) and Favero et al. (2021) *'Locality defeats the curse of dimensionality in convolutional teacher-student scenarios'* studied the benefits of locality before Misiaciewicz and Mei (2022), and Xiao (2022) *'Eigenspace restructuring: a principle of space and frequency in neural networks'* also studied the inductive bias of deep CNNs in the kernel regime.

**Limitations:**

The authors have adequately addressed the limitations and potential negative societal impact of their work.

---

> ### Author Rebuttal · Authors · 2023-08-10
>
> We thank the reviewer for the valuable comments and questions and we will revise our manuscript accordingly.  In the following, we response to the specific weakness and questions.
>
> (1) **CNN with non-overlapping pathces and the universality** [Weakness 1; Question 6,7]
>
> First of all, we agree that we should have emphasized in the beginning of the paper that we are consid- ering CNNs with non-overlapping patches and discuss related works, in particular the papers mentioned by the reviewer, of studying CNNs with similar architectures.
> Regarding prior works on universality for CNNs with log d depth, after reviewing the two highlighted
> papers, we observe:
> - Both papers indeed discuss CNNs with non-overlapping patches.
> - Neither paper addresses the universality problem.
> - Their analyses do not directly lead to universality conclusions similar to ours. A significant gap in their studies is the absence of revealing that multichannels can be used to store extra information when reducing spatial dimension with stride/pooling.
>
> In the following, we provide detailed comments for each paper.
> - [Cagnetta et al., 2023] investigates the inductive bias of NTK with deep CNNs that use non-overlapping patches and have infinitely-wide channels. First, [Cagnetta et al., 2023] did not examine the univer- sality and it is generally unclear if the associated kernel is universal. Second, [Cagnetta et al., 2023] focuses on kernel regime, which is very different from CNNs in adaptive regime. For example, in adaptive regime, that the number of channels to be O(d) for all except the last layer suffices for achieving universality.
> -  [Poggio et al., 2017] hightlights that deep locally-connected networks can learn functions with similar hierarchical structures efficiently, while shallow fully-connected networks cannot. [Poggio et al., 2017] emphasizes the role of locality in separating deep CNNs with fully-connected networks. Their study does not touch upon universality and centers on locally-connected networks, not CNNs. They use multichannels in their analysis, but does not reveal its role in maintaining information during spatial dimension reduction.
>
> In summary, while both works are valuable, they do not explore universality as we do. Their contexts are distinct from ours, making direct comparisons challenging. Furthermore, the foundational insights of our study differ notably from theirs.
>
>
> (2) **Learning Sparse Functions.** [Weakness 2]
>
> Lemma 4.4 demonstrates that deep CNNs can achieve the variable selection $
> (x_1,x_2,\dots,x_d)\mapsto (x_{i_1},x_{i_2}, \dots, x_{i_k})
> $,
> reducing the spatial dimension to $O(k)$ by using only $O(\text{log} d)$ parameters, facilitating the efficient learning of sparse functions. In contrast, fully-connected networks can achieve the same variable selection, but they necessitate a minimum of $\Omega(d)$ parameters (a single neuron already requires $\Omega(d)$ parameters). Consequently, fully-connected networks demand at least $\Omega(d)$ samples to learn sparse functions, unless specific sparsity regularizations are imposed.
>
> (3) **Symmetry-based lower bounds and the intuitive explanations.** [Weakness 3; Question 2]
>
> Our intuitive explanations are given as follows.
> Denote by $\mathbb{A}$ the learning algorithm and $G$ the correpsonding symmetry group. The inherent symmetry ensures  that learning $f^*$ is equivalent to learning the $G$-enlarged class $f^*\circ G := ${$x\mapsto f^*(\tau x): \tau \in G$}. Thus, the sample complexity of learning $f^*$ by $\mathbb{A}$ is lower bounded by the packing number of $f^*\circ G$, which, in turn, is further controlled by the packing number of the symmetry group $G$ when  $f^*$ is chosen to satisfy
> \begin{align}
> ||f^*\circ \tau_1-f^*\circ \tau_2||_{L^2(\gamma)}^2 \sim ||\tau_1-\tau_2||^2_G,
> \end{align}
> where $||\cdot||_G$ is a proper norm defined on $G$. Thus, the sample complexity is fundamentally governed by the size of the symmetry group.
>  For SGD algorithms, we have
> - The symmetry group of FCNs is the orthogonal group $G_{\mathrm{orth}}$, whose degree of freedom is $\Theta(d^2)$. This explains the lower bound $\Omega(d^2)$ for FCNs.
> - The symmetry group of LCNs is the local permutation group $G_{\mathrm{loc}}$, whose degree of freedom is $\Theta(d)$. This explains the lower bound $\Omega(d)$ for LCNs.
>
> (4)  **The argument in Bietti et al. (2021)**
>
> If our interpretation is accurate, [Bietti, 2022] contrasts shallow CNNs (encompassing 2-3 layers) with and without global pooling within a kernel regime. From the standpoint of sample complexity, introducing global pooling provides a factor d advantage, provided the target function aligns with this architecture. Therefore, in our humbble opinion, the inductive bias studied in [Bietti, 2022] is distinctly different from inductive biases associated with locality (LCNs vs. FCNs) and weight sharing (CNNs vs. LCNs).

---

> > ### Comment · Reviewer_35ph · 2023-08-13
> >
> > Thanks for your response. Let me start by saying that I am satisfied by answers 2, 3 and 4. For the latter, in particular, I suggest the author to add a comment to the manuscript which explains why their factor of $d$ difference is different from that of Bietti et al. (I seem to understand because the current manuscript does not consider global average pooling). Consequently, I will be happy to raise my mark to 7 **if** the following concerns are addressed:
> >
> > **1. Universality**
> >
> > I disagree with the authors here. I am sure Poggio et al. discuss universality (which they also find by arguing that deep convolutional networks can reproduce the calculations of fully-connected networks). Cagnetta et al. results are limited to the kernel regime, which requires infinite width and small initialisation. But once the width is sufficiently large, initialisation does not matter anymore for approximation property. I understand that the contribution of the authors is still valuable and, as I said in the original review, I believe that section 3 is a nice addition to the literature. However, for the aforementioned reasons, I suggest that the authors downplay the claim that ther work is the first to show that universality requires depth of order log(d).
> >
> > **2. Additional questions/comments**
> >
> > The rebuttal does not mentioned questions 2 to 7 of my original review. Are the authors planning to address those?

---

> > > ### Author Response · Authors · 2023-08-16
> > > **Thanks for the feedback**
> > >
> > > Thank you for your constructive and encouraging feedback. We truly appreciate these thoughtful comments which will significantly strengthen our paper.
> > >
> > > ---
> > > **1. Bietti et al. (2021)**
> > >
> > > We are very happy to add a few sentences to explain why the factor $d$ in Bietti et al. (2021) is different from ours.
> > >
> > > ---
> > > **2. Universality**
> > >
> > > We acknowledge the possibility that our claim may overlook relevant prior works. **We are more than willing to downplay the claim**. Nevertheless, we are encountering difficulty in understanding how the universality was established in the two papers mentioned by the reviewer. We kindly request the reviewer's assistance in clarifying this matter.
> > >   - **[Cagnetta et al., 2023]**  studies the kernel associated to deep CNNs with $\log d$ layers and infinite-many channels.  we do not find evidence within the work of [Cagnetta et al., 2023]  to suggest that universality has been established for the associated deep CNNs:
> > >     -  Firstly, this work does not appear to investigate the universality of deep CNNs.
> > >     - Secondly, proving universality in the kernel regime needs  to verify that the corresponding kernel has no zero eigenvalues. [Cagnetta et al., 2023] does not undertake this verification. Moreover, for the two-layer ReLU networks in NTK/Random feature regime,  the corresponding kernels do has zero eigenvalues as calculated in Appendix D.2 of [Bach 2017, *Breaking the Curse of Dimensionality with Convex Neural Networks*]. Given this, it seems plausible to hypothesize that the deep-CNN kernel studied in [Cagnetta et al., 2023] may also exhibit zero eigenvalues.
> > >
> > >
> > >
> > > -  We also carefully read the content of  **[Poggio et al., 2017]** but were unable to identify a clear statement regarding the universality result. However, we did find several comments that could potentially be interpreted as assertions of universality:
> > >     - In the caption of Figure 1 of [Poggio et al., 2017], the text states: *``In a binary tree with $n$ inputs, there are $\log 2n$ levels and a total of $n-1$ nodes. Similar to the shallow network, a hierarchical network is universal, that is, it can approximate any continuous function''*. Unfortuantely, we did not find how this conclusion is proved by the technique developped by [Poggio et al., 2017].
> > >     - In the second paragraph of Section 4.3, the paper mentions: *``The proofs of our theorems show that linear combinations of compositional functions are universal in the sense that they can approximate any function''*. While we understand that this is true,  it remains unclear  how [Poggio et al., 2017] demonstrate that deep CNNs are capable of implementing this property.
> > >     - In page 13, the paper contains the statement: *``From the previous observations, it follows that a hierarchical
> > > network of order at least 2 can be universal. ''*  We are unsure what specific "previous observations" this statement refers to.
> > >
> > > &nbsp;&nbsp;&nbsp;&nbsp;&nbsp;&nbsp;&nbsp;&nbsp;&nbsp;We would be most grateful if the reviewer could point us to the specific statements or sections in [Poggio et &nbsp;&nbsp;&nbsp;&nbsp;&nbsp;&nbsp;&nbsp;&nbsp;&nbsp;al., 2017] that articulate and support the claim of universality.
> > >
> > > ---
> > >
> > > **3. Other Questions.**
> > >
> > > Regarding Question 2, we have provided the explanation in point (3) of the  rebuttal. In a nutshell, the key difference stems from the different degrees of freedom for the symmetry groups of FCNs versus LCNs. The suggestions in Questions 3-7 are helpful, and we will certainly incorporate them in the forthcoming revision.
> > >
> > > ***
> > >
> > > Lastly, we would like to extend our sincere gratitude to the reviewer again for taking the time to  respond to our rebuttal.
> > >
> > > Best Regards,
> > >
> > > The Authors

---

> > > > ### Comment · Reviewer_35ph · 2023-08-19
> > > >
> > > > Thanks for your response.
> > > >
> > > > Regarding Poggio et al., I stand corrected. I too cannot find any clear demonstration of universality in the paper. I guess since I found it intuitively true I never checked, apologies for that.
> > > >
> > > > Regarding the kernel case (and Cagnetta et al.), it is true that there can be some zero eigenvalues of the kernel preventing universality, but this issue is a technical one which can be resolved in simple ways e.g. inclusion of biases or change of the activation function, as discussed, for instance, in Bietti and Bach *Deep Equals Shallow for ReLU Networks in Kernel Regimes* (2020).
> > > >
> > > > To sum up, I still think that the idea that universality can be achieved with log(d) layers with non-overlapping patches was already out there in different forms. After all, in such architecture, the activations of the penultimate layer are functions of the whole input, thus universality should be easy to achieve with the proper activation function and enough channels. The current writing (line 6, abstract, and lines 45-46) suggests otherwise, hence it should be modified.
> > > >
> > > > Let me restate that I do not think this diminishes in any way the value of the present manuscript: I like the way the authors show universality and I believe that it leads to a valuable insight.

---

> > > > > ### Author Response · Authors · 2023-08-20
> > > > > **Thanks for your further comments**
> > > > >
> > > > > We sincerely appreciate the reviewer's time and thoughtful engagement in this discussion—big thanks! Below are our further clarifications in response to your comments.
> > > > >
> > > > > - Regarding [Cagnetta et al., 2023], we are in agreement with your argument, which suggests that demonstrating universality within the kernel regime is indeed possible. Nonetheless, this doesn't negate our main point: [Cagnetta et al., 2023] did not explicitly prove universality.
> > > > >
> > > > > - With respect to your observation that "*the penultimate layer consists of functions of the entire input, making universality readily achievable with the appropriate activation function and sufficient channels,*" we concur. In our manuscript, we offer a specific construction to implement this intuition.
> > > > >
> > > > >
> > > > >
> > > > > In light of the above  discussion, **we will make the following changes in the revised manuscript**:
> > > > >
> > > > > -  In the abstract and introduction, we will moderate the statement when discussing our universality result. Specifically, we will acknowledge prior efforts, like [Cagnetta et al., 2023] and [Poggio et al., 2017], while making clear that these studies have not **explicitly** established universality.
> > > > >
> > > > > -  In Section 3, we will include a paragraph, thoroughly explaining the difference from  [Cagnetta et al., 2023] and [Poggio et al., 2017] by **incorporating points from our discussion**.
> > > > >
> > > > >
> > > > >
> > > > > Best,
> > > > >
> > > > > The Authors

---

> > > > > > ### Comment · Reviewer_35ph · 2023-08-20
> > > > > >
> > > > > > I'm happy with the proposed revisions! Best regards

---

### Author Rebuttal · Authors · 2023-08-10

**Global Response to All Reviewers**

We thank all reviewers for their constructive feedback. In our forthcoming revision, we will integrate these comments, paying special attention to an enriched discussion on the related works suggsted  by the reviewers, in particular [Poggio et al., 2017].

In addition, we have conducted some numerical experiments and the results can be founded in the attached one-page PDF file:

-  In Figure 1, we empirically show that CNNs with limited channels can learn sparse interactions efficiently regardless of whether they are short- or long-range, without the need of explicit regularization. This is consistent with our theoretical predictions in Section 4.
In contrast, FCNs and linear regression fail in learning the sparse functions.
-  In Figure 2, we plot how the sample complexity scales with input dimension for CNNs and FCNs for learning a quadratic target, similar to what is considered in our theoretical analysis. Again, these empirical results are in line with our theoretical predictions in Section 5.

The above results demonstrate that our theory is empirically relevant in the sense that commonly-used optimizers can find those solutions.

---

> ### Comment · Reviewer_35ph · 2023-08-13
>
> Thanks for the figures! Why do you stop training FCNs so early? In the case of CNNs the test error starts decreasing at 500 (left figure) and 1500 (right figure) iterations, the FCNs should be at least trained for longer than that.

---

> > ### Author Response · Authors · 2023-08-16
> > **Further training cannot improve the generalization for FCNs**
> >
> > Thank you for the question. We would like to clarify that we did not intentially stop training FCNs early. The stop criteria for all models are the same: training was terminated when the training loss falls below 1e-5. We aplogieze for the confusion and we will provide a new figure to fix this issue in the next version of our paper.
> >
> > In fact, we indeed trained FCNs  for much more steps (e.g., $10^4$ steps where the loss becomes smaller than 1e-20) and never obseve improvement in  generalization. We point out that this is also evident from the current Figure 1 in the attached pdf file:
> >
> > -  For FCNs, the training loss keeps dececreasing exponentially, while the test loss plateaus rapidly. This pronounced divergence between the training and test curves is a clear indication of overfitting in the case of FCNs. Furture training will not  be helpful.
> > - In contrast, for training CNNs,  the training and test curves remain closely to each other, though it requires more steps to reduce the training loss. This indicates further training is helpful for improving generalization.

---

### Decision · Program_Chairs · 2023-09-21

**Decision:**

Accept (poster)

**Comment:**

This paper studies the inductive bias of deep convolutional neural networks (CNNs) in three distinct ways by giving sample complexity bounds. All reviewers belive this paper makes valid contributions to the community. The AC agrees and recommends acceptance.